# Mechanical loading of intraluminal pressure mediates wound angiogenesis by regulating the TOCA family of F-BAR proteins

Shinya Yuge[1,12], Koichi Nishiyama[2,3,12 ✉], Yuichiro Arima [2,4], Yasuyuki Hanada[2,5], Eri Oguri-Nakamura[1], Sanshiro Hanada [2], Tomohiro Ishii[1], Yuki Wakayama[6], Urara Hasegawa [7], Kazuya Tsujita [8,9], Ryuji Yokokawa [10], Takashi Miura [11], Toshiki Itoh[8,9], Kenichi Tsujita[4], Naoki Mochizuki [6] & Shigetomo Fukuhara [1 ✉]

Angiogenesis is regulated in coordinated fashion by chemical and mechanical cues acting on endothelial cells (ECs). However, the mechanobiological mechanisms of angiogenesis remain unknown. Herein, we demonstrate a crucial role of blood flow-driven intraluminal pressure (IP) in regulating wound angiogenesis. During wound angiogenesis, blood flow-driven IP loading inhibits elongation of injured blood vessels located at sites upstream from blood flow, while downstream injured vessels actively elongate. In downstream injured vessels, F-BAR proteins, TOCA1 and CIP4, localize at leading edge of ECs to promote N-WASP-dependent Arp2/3 complex-mediated actin polymerization and front-rear polarization for vessel elongation. In contrast, IP loading expands upstream injured vessels and stretches ECs, preventing leading edge localization of TOCA1 and CIP4 to inhibit directed EC migration and vessel elongation. These data indicate that the TOCA family of F-BAR proteins are key actin regulatory proteins required for directed EC migration and sense mechanical cell stretching to regulate wound angiogenesis.

[1] Department of Molecular Pathophysiology, Institute for Advanced Medical Sciences, Nippon Medical School, 1-1-5 Sendagi, Bunkyo-ku, Tokyo 113-8602, Japan. [2] International Research Center for Medical Sciences, Kumamoto University, Kumamoto City, Kumamoto 860-0811, Japan. [3] Laboratory of Vascular and Cellular Dynamics, Department of Medical Sciences, University of Miyazaki, Miyazaki City, Miyazaki 889-1962, Japan. [4] Department of Cardiovascular Medicine, Graduate School of Medical Sciences, Kumamoto University, Kumamoto City, Kumamoto, Japan. [5] Department of Cardiology, Graduate School of Medicine, Nagoya University, Nagoya City, Aichi 466-8550, Japan. [6] Department of Cell Biology, National Cerebral and Cardiovascular Center Research Institute, Suita, Osaka 565-8565, Japan. [7] Department of Materials Science and Engineering, Pennsylvania State University, University Park, PA 16802, USA. [8] Biosignal Research Center, Kobe University, 1-1 Rokkodai-cho, Nada-ku, Kobe, Hyogo 657-8501, Japan. [9] Division of Membrane Biology, Department of Biochemistry and Molecular Biology, Kobe University Graduate School of Medicine, 7-5-1 Kusunoki-cho, Chuo-ku, Kobe, Hyogo 650-0017, Japan. [10] Department of Micro Engineering, Graduate School of Engineering, Kyoto University, Kyoto 615-8540, Japan. [11] Department of Anatomy and Cell Biology, Graduate School of Medical Sciences, Kyushu University, Fukuoka City, Fukuoka 812-8582, Japan. [12] These authors contributed equally: Shinya Yuge, Koichi Nishiyama. ✉email: koichi_nishiyama@med.miyazaki-u.ac.jp; s-fukuhara@nms.ac.jp

Angiogenesis refers to physiological and pathological processes through which new blood vessels form from pre-existing vessels[1]. Not only chemical factors but also mechanical cues act on endothelial cells (ECs) to regulate angiogenesis. Blood flow-driven mechanical forces such as shear stress, hydrostatic pressure, and cyclic stretch play multiple roles in angiogenesis. For instance, fluid shear stress is known to control EC sprouting and their elongation direction during angiogenesis[2–4]. It has also been reported that blood flow induces polarized migration of ECs to induce pruning of excessive blood vessels[5,6]. Furthermore, blood flow reportedly drives lumen formation by inducing the formation of inverse membrane blebs during angiogenesis[7]. ECs also sense the mechanical properties of the extracellular environment and adapt their behavior accordingly during physiological and pathological angiogenesis[8]. However, the dynamics of EC behavior in angiogenesis and especially its regulation by mechanical forces remain poorly understood.

Mechanical regulation of angiogenesis is especially important for successful wound healing, a complex and dynamic process by which tissue repairs itself after injury[9–12]. During wound healing, ECs are stretched and activated by wound contraction to facilitate tissue repair through induction of angiogenesis. The biomechanical force generated by wound contraction also reportedly induces nonangiogenic expansion of pre-existing vessels[13]. Furthermore, applying external mechanical forces to wounded tissues influences vascular morphogenesis and angiogenesis to promote tissue regeneration[10]. However, molecular mechanisms by which mechanical forces affect EC behavior to regulate wound angiogenesis remain largely unclear, because methods of analyzing this highly dynamic process in vivo have been lacking.

During angiogenesis, ECs forming the vessel sprouts establish front–rear polarity and migrate by extending actin-based protrusions such as lamellipodia and filopodia at the leading edge. The actin-related protein 2/3 (Arp2/3) complex and Formin family proteins are key actin nucleators that induce the formation of lamellipodia and filopodia at the leading edge of migrating cells. Previously, we showed that Formin-like 3 regulates the extension of endothelial filopodia to facilitate angiogenic sprouting in zebrafish[14]. Arp2/3 complex-mediated actin polymerization reportedly regulates EC migration and junctional remodeling[15–17]. For instance, Arp2/3 complexes induce the polarized formation of actin-driven junctional-associated intermittent lamellipodia to promote directed EC migration during sprouting angiogenesis[16]. Consistently, mice deficient in Wave2, a gene encoding a nucleation-promoting factor of the Arp2/3 complex which belongs to the Wiskott–Aldrich syndrome protein (WASP) family proteins, exhibit embryonic lethality due to impaired angiogenesis[18]. Functional activities of WASP family proteins including WASP, neuronal-WASP (N-WASP), and WAVE are regulated by Rho family small GTPases, such as Cdc42 and Rac, and Bin-Amphiphysin-Rvs (BAR) domain-containing proteins[19]. Thus, WASP family protein-mediated Arp2/3 complex-dependent actin polymerization might regulate vessel elongation during angiogenesis.

In this study, we uncovered a novel molecular mechanism governing the mechanical regulation of EC behavior during wound angiogenesis. By exploiting a recently developed live-imaging system for adult zebrafish[20], we analyzed the dynamics of angiogenic EC behavior during wound healing and surprisingly discovered that elongation of severed blood vessels is preferentially induced downstream from blood flow, whereas blood flow-driven intraluminal pressure (IP) loading suppresses elongation of upstream injured vessels by inducing EC stretching. We confirmed the inhibitory effect of IP load-induced EC stretching on vessel elongation evoked by directional EC migration during angiogenesis by employing a newly developed reconstitution

assay system with a microfluidic device. Furthermore, by exploring the underlying molecular mechanisms, we showed that Transducer of Cdc42 dependent actin assembly (TOCA) family of Fes/Cdc42 interacting protein 4 (CIP4) homology-BAR (F-BAR) proteins, TOCA-1 (also known as Formin-binding protein 1-like (FNBP1L) and CIP4 (also known as a thyroid hormone receptor interactor 10 (TRIP10)), are key regulators of Arp2/3 complex-mediated actin polymerization and polarized EC migration in angiogenesis and also act as sensors for IP load-induced EC stretching which restricts elongation of upstream injured vessels during wound angiogenesis.

## Results

**Severed blood vessels elongate downstream, not upstream, from blood flow during wound angiogenesis.** To analyze EC behavior during cutaneous wound healing, we introduced wounds onto the flanks of adult *Tg(kdrl:EGFP)* zebrafish expressing EGFP in ECs. Cutaneous wounding immediately induced angiogenesis, during which elongation of severed blood vessels was actively induced at the early stage (1-2 days after injury), while sprouting from pre-existing vessels occurred at a relatively late stage (3-5 days after injury) (Supplementary Fig. 1a and Supplementary Movie 1), as previously reported[20]. Interestingly, we noticed that elongation of severed blood vessels was actively induced at sites downstream from blood flow, whereas vessels located upstream did not elongate efficiently (Fig. 1a and Supplementary Movie 2). This phenomenon was confirmed by analyzing reparative processes of a single injured capillary, indicating approximately 75% of vessel repair to be attributable to elongation occurring downstream (Fig. 1b, c, Supplementary Fig. 1b, and Supplementary Movies 3 and 4). We also observed preferential elongation of injured downstream vessels when arterial and venous intersegmental vessels (ISVs) of *Tg(fli1a:EGFP)* zebrafish larvae at 3 days post fertilization (dpf) were severed by laser ablation (Fig. 1d, e, Supplementary Fig. 1c, d, and Supplementary Movies 5 and 6). These results indicate that severed blood vessels mainly elongate downstream, not upstream, from blood flow during wound angiogenesis.

**Loading of IP inhibits elongation of upstream injured vessels during wound angiogenesis.** We attempted to identify the factor(s) underlying the differences in elongation of injured blood vessels. Tissue hypoxia is a key driver of angiogenesis[21]. However, hypoxic states did not differ significantly between tissues surrounding downstream and upstream injured vessels (Supplementary Fig. 2), indicating the different elongation of injured vessels to not be attributable to differing hypoxic states.

Although injured blood vessels have a luminal structure with a closed end, the heart still pumps blood only into upstream injured vessels, suggesting that IP is constantly applied to these vessels, but not to those downstream. Thus, we assumed that IP acting on injured blood vessels will account for elongation differences. To address this hypothesis, we analyzed hemodynamics in injured arterial ISVs (aISVs) in larvae by intravascularly injecting polyethylene glycol-coated fluorescent microspheres (PEGylated FM) and quantum dots (Qdots). FM was coated with polyethylene glycol to avoid non-specific binding to the blood vessel lumen (Supplementary Fig. 3 and Supplementary Table 1). The particle size of PEGylated FM and Qdots is approximately 0.5 μm and 10 nm, respectively. Qdots are sufficiently small to diffuse freely in the vessels, and thereby visualize the blood vessel lumen. On the other hand, diffusion of PEGylated FM within the narrow injured vessels is expected to be much slower due to the high particle/vessel size ratio compared to Qdots. Because of this, PEGylated FM can rarely enter the injured vessels without blood

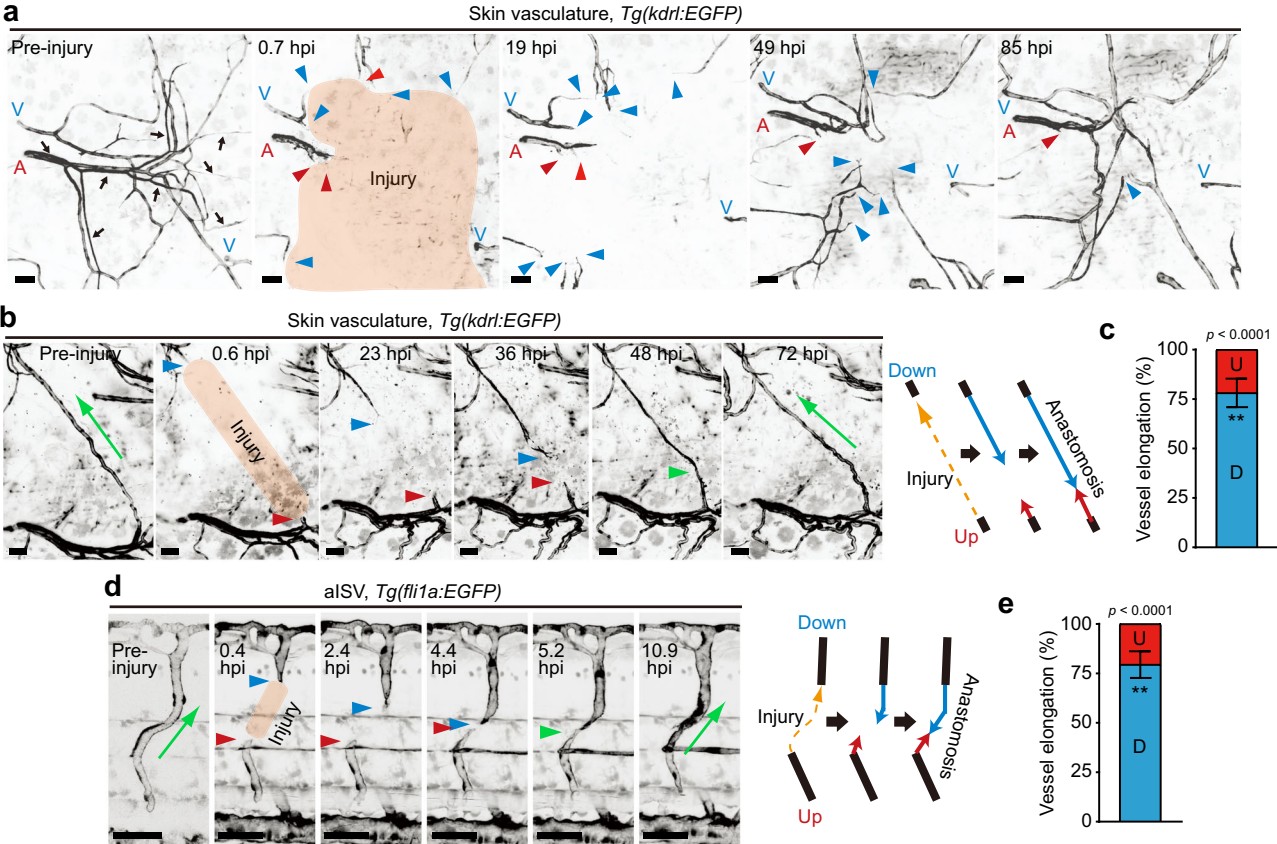

**Fig. 1 Severed blood vessels elongate downstream, not upstream, from blood flow during wound angiogenesis. a, b** Time-lapse confocal images of angiogenesis in injured skin of *Tg(kdrl:EGFP)* adult zebrafish. Confocal *z*-projection images of cutaneous vasculature before injury (Pre-injury) and at the elapsed time following injury (hpi: hours post injury). Note that cutaneous vessel networks in adult zebrafish consist of not only blood vessels but also vessels not containing circulating erythrocytes (arrows)[20]. Herein, we focused on blood vessels. In **b**, a single capillary in the skin (approximately 560 μm in length) was injured. The repair process of the injured vessel is depicted on the right. **c** Amounts of elongation of upstream (red) and downstream (blue) injured vessels as observed in **b** are expressed as percentages of total elongation (measured as length). Data are shown as means ± s.e.m. (*n* = 7 vessels examined over 6 animals). **d** Time-lapse confocal *z*-projection images of the repair process of injured aISV in *Tg(fli1a:EGFP)* larva at 3 dpf are as in **b**. Lateral view, anterior to the left. **e** Amounts of elongation of upstream (red) and downstream (blue) injured aISVs are as in (**c**). Data are shown as means ± s.e.m. (*n* = 6 animals). In **a**, **b**, **d**, injured areas are depicted in orange; A arteries, V veins; red and blue arrowheads, leading edges of injured vessels upstream and downstream from blood flow, respectively; green arrowheads, anastomotic sites of injured vessels; green arrow, the direction of blood flow. \*\**p* < 0.01 by two-sided *t* test (**c**, **e**). Source data are provided as a Source data file. Scale bars: 50 μm.

flow. When aISVs were severed by laser ablation, both PEGylated FM and Qdots entered only the base of downstream injured vessels, barely reaching the tip (Fig. 2a, Supplementary Fig. 4a–c, and Supplementary Movies 7 and 8), indicating that blood did not flow into the lumen. Thus, markedly high IP is not applied to downstream injured vessels. On the other hand, the entire region of upstream injured vessels was filled with Qdots (Fig. 2a, Supplementary Fig. 4b, c, and Supplementary Movie 7). However, PEGylated FM rarely entered injured vessels upstream from the dorsal aorta, despite frequently going into intact aISVs before injury (Fig. 2a, b, Supplementary Fig. 4a, d, and Supplementary Movies 7, 9, and 10). Importantly, some PEGylated FM, which ended up in upstream injured vessels, showed Brownian motion-like movement when entering upstream injured vessels by chance (Fig. 2a, Supplementary Fig. 4a, and Supplementary Movie 8), suggesting the absence of laminar blood flow within upstream injured vessels. Moreover, we confirmed the hemodynamics in injured aISVs by injecting Qdots into the common cardinal vein in larvae with cardiac arrest and subsequently analyzing the fluorescence dynamics in injured aISVs in response to re-starting blood flow (Supplementary Fig. 4e and Supplementary Movie 11). The dorsal aorta was quickly filled with Qdots when blood flow

started. However, the Qdots moved only gradually from the dorsal aorta to the tip of the upstream injured aISV (approximately 0.3 μm/s), suggesting Qdot accumulation in upstream injured aISV via passive diffusion. Collectively, these findings indicate that blood flow in upstream injured vessels is minimal or absent. Therefore, upstream injured vessels are probably exposed mainly to IP rather than shear stress generated by blood flow.

Next, we investigated whether IP restricts the elongation of upstream injured vessels during wound angiogenesis and found that the upstream injured vessels in wounded skin began to elongate when IP was diminished by severing the more upstream site (Fig. 2c, d and Supplementary Movie 12). These results suggest that blood flow-induced IP restricts elongation of upstream injured vessels. To further confirm the inhibitory effect of IP loading on vessel elongation, we originally developed an in vitro angiogenesis system in a microfluidic device (on-chip angiogenesis model), in which hydrostatic pressure can be loaded onto the lumens of elongating vessels formed by human umbilical vein ECs (HUVECs) (Supplementary Fig. 5). This novel system allows us to quantitatively analyze the effects of IP loading on vessel elongation and EC behavior during angiogenesis. We showed, using this system, that an IP load of approximately

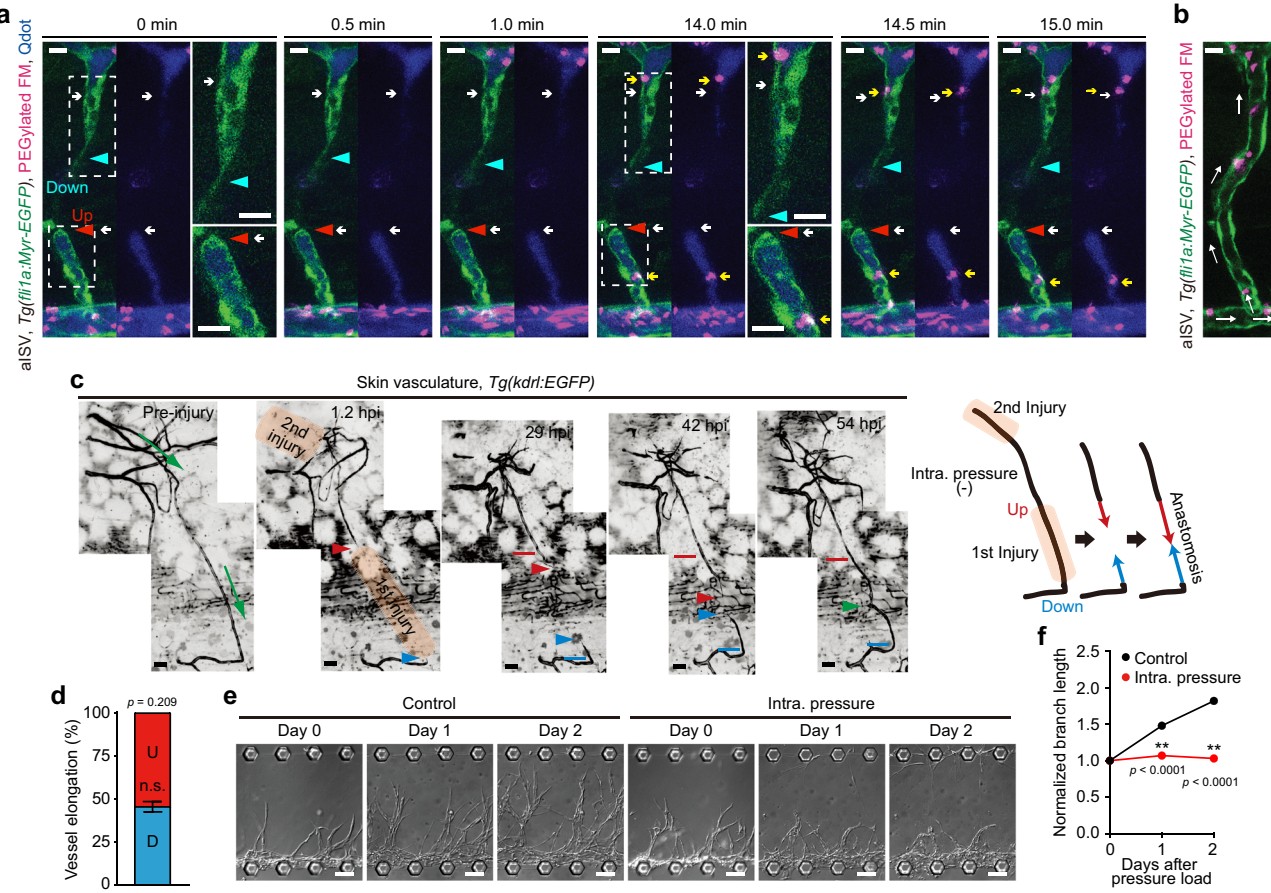

**Fig. 2 Blood flow-driven IP loading restricts elongation of injured blood vessels upstream from blood flow during wound angiogenesis. a** Confocal *z*-projection image of injured aISV in 3 dpf *Tg(fli1a:Myr-EGFP)* larva intravascularly injected with PEGylated FM and Qdots (Qdot 705) and their subsequent timelapse images at the elapsed time indicated at the top. Image acquisition was started at 2.7 hpi and 37 min after the injection. Lateral view, anterior to the left. Left, merged images of EGFP (green), PEGylated FM (magenta), and Qdots (blue); right, merged images of PEGylated FM (magenta) and Qdots (blue). Boxed areas are enlarged on the right. Red and blue arrowheads, leading edges of upstream and downstream injured vessels, respectively; white arrows, tip of upstream and downstream injured vessels filled with Qdots; yellow arrows; PEGylated FM inside the injured vessels. **b** Confocal *z*-projection image of trunk vasculature (aISV) in 3 dpf *Tg(fli1a:Myr-EGFP)* larva intravascularly injected with PEGylated FM. Lateral view, anterior to the left. Merged image of EGFP (green) and PEGylated FM (magenta). White arrows, the direction of blood flow. **c** Time-lapse confocal *z*-projection images of elongation of an injured skin capillary of adult zebrafish in which IP of the upstream injured vessel was relieved by cutting the more upstream site are as in Fig. 1b. Red and blue lines indicate the positions of the tips of upstream and downstream injured vessels at 1.2 hpi, respectively. **d**, Amounts of elongation of upstream (red) and downstream (blue) injured vessels as observed in **c** are as in Fig. 1c. (*n* = 4 animals). n.s., not significant by two-sided *t* test. **e**, **f** Effects of vascular IP loading on branch elongation of on-chip angiogenesis. **e** Serial DIC images showing elongation of angiogenic branches before and after loading without (Control) or with IP (Intra. pressure). **f** Quantification of branch lengths as observed in **e**. Normalized values relative to that before IP load are means ± s.e.m. (*n* = 45 branches examined over 3 independent experiments for each). \*\**p* < 0.01 by two-way ANOVA followed by Sidak's test. Source data are provided as a Source data file. Scale bars: 10 μm (**a**, **b**), 50 μm (**c**), 100 μm (**e**).

1.2 mmHg to lumens of angiogenic branches significantly suppressed their elongation (Fig. 2e, f). Collectively, these findings indicate that IP loading suppresses elongation of upstream injured vessels during wound angiogenesis.

**IP load-induced stretching of ECs inhibits elongation of upstream injured vessels**. To investigate how IP loading suppresses vessel elongation, we analyzed IP load-induced morphological changes in elongating vessels. In an on-chip angiogenesis model, angiogenic branches extended protrusions toward the elongation direction, while expanding and showing rounded morphology of the leading edge after 3 days of IP loading (Fig. 3a). We also analyzed the morphologies of injured skin vessels and severed ISVs in adult and larval zebrafish, respectively (Fig. 3b–e). In both cases, injured vessels located downstream from blood flow extended protrusions at the leading edge. In clear

contrast, the leading edge of upstream injured vessels exhibited expanded and rounded morphology. The diameter of upstream injured ISVs shrank when blood flow was stopped by treatment with either tricaine or 2,3-butanedione monoxime (BDM) and re-expanded in response to re-starting blood flow (Fig. 3f, g and Supplementary Fig. 6). The reduction of blood flow by BDM also resulted in a decreased diameter of upstream injured vessels (Supplementary Fig. 6). However, changes in blood flow did not significantly affect the morphology of downstream injured ISVs (Fig. 3f, g and Supplementary Fig. 6). These results indicate that blood flow-driven IP loading induces expansion of upstream injured vessels.

IP load-induced expansion of injured blood vessels results in stretching of ECs comprising the vessel wall, which might inhibit vessel elongation. In addition, hydrostatic pressure is simultaneously applied to ECs in the upstream injured vessels. To clarify whether IP loading suppresses vessel elongation by stretching the

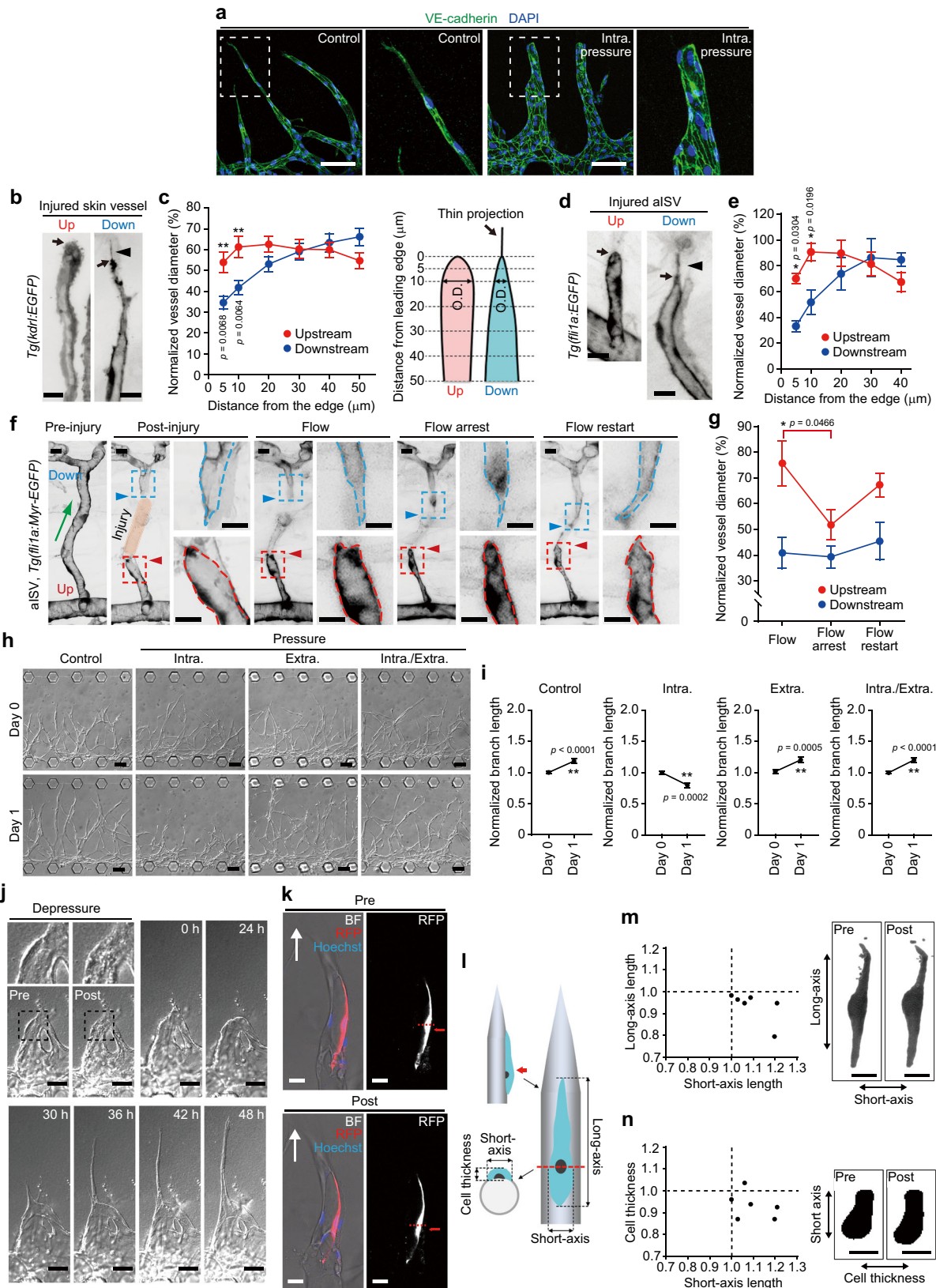

ECs or by applying hydrostatic pressure to them, we examined the effects of IP or extraluminal pressure (EP) loading, alone and in combination, on the elongation of angiogenic sprouts using an on-chip angiogenesis model (Supplementary Fig. 7a). IP loading immediately induced vessel expansion, while loading of EP caused shrinkage of angiogenic branches (Supplementary

Fig. 7b–e and Supplementary Movies 13 and 14). Furthermore, the shrunken vessels loaded with EP were only slightly expanded by additional loading of IP (Supplementary Fig. 7d, e and Supplementary Movie 15), suggesting that ECs in vessels loaded with either EP or both EP and IP were not exposed to stretching, despite hydrostatic pressure application. Thus, using our novel

**Fig. 3 IP loading inhibits elongation of angiogenic sprouts via stretching of ECs. a** Confocal *z*-projection images of on-chip angiogenic branches loaded without (left) or with IP (right) for 3 days. Green, VE-cadherin; blue, DAPI. **b**, **d** Confocal *z*-projection images of upstream (Up) and downstream (Down) injured vessels [**b** skin vessels in adult *Tg(kdrl:EGFP)* zebrafish; **d** aISVs in 3 dpf *Tg(fli1a:EGFP)* larva]. **c**, **e** Outer diameters (O.D.) of the upstream (red) and downstream (blue) injured vessels as in **b**, **d** are shown as percentages relative to that of pre-injured vessels. The *x*-axis shows the distance from the leading edge (arrows in **b**, **d**). Thin projections (arrowheads in **b**, **d**) are excluded from measurement. Data are means ± s.e.m. [**c**, upstream and downstream, $n = 23$ and 19 vessels examined over 10 and 11 animals, except for upstream at 50 μm ($n = 22$) and downstream at 30 μm ($n = 18$); **d**, upstream and downstream, $n = 6$ animals for each, except for upstream at 40 μm ($n = 4$) and downstream at 40 μm ($n = 5$)]. **f** Effects of changes in blood flow on the morphology of injured aISVs. Images of pre- (Pre-injury) and post-injured (Post-injury) aISVs in 3 dpf *Tg(fli1a:Myr-EGFP)* larva and its subsequent images at 3 hpi (Flow), after tricaine-induced blood flow arrest (Flow arrest), and after restarting blood flow (Flow restart). **g** Outer diameters of injured aISVs at 10 μm from the leading edge are as in **c** [upstream and downstream at Flow ($n = 9$ and 10 animals), Flow arrest ($n = 9$ and 10 animals), Flow restart ($n = 8$ and 9 animals)]. **h** DIC images of on-chip angiogenic branches before and 1 day after loading without (Control) and with IP (Intra.), EP (Extra.), or both types of pressure (Intra./Extra.). **i** Quantification of branch lengths as in **h**. Normalized values relative to that before pressure loading are means ± s.e.m. (the number of branches examined over 4 independent experiments: Control, $n = 68$ and 67; IP, $n = 40$ and 38; EP, $n = 50$ and 50; IP/EP, $n = 50$ and 50, for Days 0 and 1). **j** DIC images of IP-loaded angiogenic branch (Pre) and, subsequently, an unloaded vessel (Post) and its subsequent time-lapse images at elapsed times indicated at top. **k** Confocal *z*-projection images merged with bright-field images of an angiogenic branch, with a small population of RFP-expressing ECs, before (Pre) and after (Post) IP loading. Arrows, elongation direction; red, RFP; blue, Hoechst 33342. Right, binary RFP images. **l–n** Changes in Short- and Long-axis lengths of cells (**m**) and those in Short-axis length and Cell thickness of cells (**n**) upon IP loading were measured as in **l**, and expressed as a ratio of the values before and after IP loading. Each dot represents an individual cell. Right, binary images of RFP-expressing cells. Boxed areas are enlarged on the right (**a**, **f**) and top (**j**). Statistical significance was determined by two-way ANOVA followed by Sidak's test (**c**, **e**), Kruskal–Wallis followed by Dunn's test among the changes in the upstream or downstream vessel (**g**), and two-sided *t* test (**i**). *$p < 0.05$, **$p < 0.01$. Source data are provided as a Source data file. Scale bars: 100 μm (**a**, **h**), 10 μm (**b**, **d**, **f**), 50 μm (**j**), 25 μm (**k**, **m**, **n**).

system, we examined whether hydrostatic pressure is involved in IP load-induced inhibition of vessel elongation, and found that loading of either EP or both EP and IP did not inhibit elongation of angiogenic branches (Fig. 3h, i). In contrast, expanded vessels loaded with IP failed to elongate, but they showed immediate shrinkage and began to extend protrusions and re-elongate upon pressure release (Fig. 3h–j and Supplementary Movies 16 and 17). These results suggest EC stretching to be a cause of IP load-mediated inhibition of vessel elongation. Consistently, quantitative analyses of the morphology of endothelial stalk cells in the angiogenic branches revealed that the long-axis length of stalk cells did not change significantly in response to IP loading (from 124.5 ± 11.7 to 116.0 ± 10.9 μm, $p = 0.1091$ by two-sided *t* test, $n = 6$), while the short-axis length increased (from 27.7 ± 2.1 to 30.43 ± 2.7 μm, $p = 0.0465$ by two-sided *t* test, $n = 6$) and the cell thickness decreased (from 11.5 ± 0.7 to 10.7 ± 0.6 μm, $p = 0.0469$ by two-sided *t* test, $n = 6$) with IP loading (mean ± s.e.m.) (Fig. 3k–n). In contrast, loading with both EP and IP did not change significantly the morphology of stalk cells (Supplementary Fig. 7f). These results indicate that endothelial stalk cells which longitudinally aligned along the angiogenic branches were circumferentially stretched by IP loading, but not by loading with both EP and IP. Furthermore, even endothelial tip cells, if forming a lumen together with stalk cells, tended to be circumferentially stretched upon IP loading (Supplementary Fig. 7g). These findings confirm the inhibitory effect of IP load-induced EC stretching on vessel elongation.

**IP loading does not stimulate EC division to inhibit elongation of upstream injured vessels**. Next, we investigated the mechanisms by which IP load-induced EC stretching inhibits vessel elongation. Cell stretching induces EC division, during which they do not migrate efficiently[22,23]. Hence, stretch-induced EC division might inhibit elongation of upstream injured vessels. However, this was not the case, since the number of EC divisions was significantly higher in downstream than in upstream injured vessels in the wounded skin of adult zebrafish (Supplementary Fig. 8). Furthermore, EC division never occurred during the repair processes of injured ISVs in zebrafish larvae, which we analyzed in this study ($n = 28$ animals).

**IP load-induced EC stretching prevents leading edge localization of Arp2/3 complexes to inhibit actin polymerization and front–rear polarization**. Plasma membrane tension reportedly acts as an inhibitor of actin assembly and therefore maintains the front–rear polarity of migrating cells by confining actin polymerization signals to the leading edge[24–28]. Thus, we hypothesized that IP load-induced ectopic stretching of ECs in angiogenic branches may inhibit actin-based protrusion at the leading edge, resulting in disruption of front–rear polarity for directional cell migration and thereby inhibiting vessel elongation. To address this hypothesis, we first examined EC front–rear polarity by analyzing the positions of their Golgi apparatus and nucleus, because directionally migrating ECs position their Golgi apparatus ahead of the nucleus in the direction of migration[29]. In an on-chip angiogenesis model, ECs in vascular sprouts mainly showed Golgi apparatus positioning in front of the nucleus toward the vessel elongation direction, while their positions were random distribution 24 h but not 1 h after IP loading (Fig. 4a, b, Supplementary Fig. 9, and Supplementary Movies 18 and 19). In contrast, loading of either EP or both IP and EP did not affect the polarized positions of the Golgi apparatus (Fig. 4b). These results show that IP load-induced EC stretching disrupts EC front–rear polarity in angiogenic branches. We also investigated the effects of IP loading on actin polymerization in ECs during elongation of angiogenic branches. In the endothelial tip cells constituting elongating vessels, actin polymerization was actively induced at the leading edge where the Arp2/3 complex, a key regulator of actin filament nucleation[30], localized (Fig. 4c–f). However, IP loading significantly inhibited actin polymerization and membrane protrusion at the leading edge (Fig. 4c–e and Supplementary Fig. 10a, b). Furthermore, Arp2/3 complexes disappeared from the leading edge immediately after loading of IP, but not of either EP alone or both IP and EP (Fig. 4c–f and Supplementary Fig. 10c, d). IP load-induced disappearance of Arp2/3 complexes from the leading edge was also observed in endothelial stalk cells (Supplementary Fig. 10e–g). Importantly, the disappearance of Arp2/3 complexes from the leading edge preceded IP load-induced reduction of actin filaments, suggesting that IP loading inhibits Arp2/3 complex-mediated actin polymerization. We further showed blood vessel elongation to be significantly inhibited by treatment with CK-666, an Arp2/3 complex inhibitor

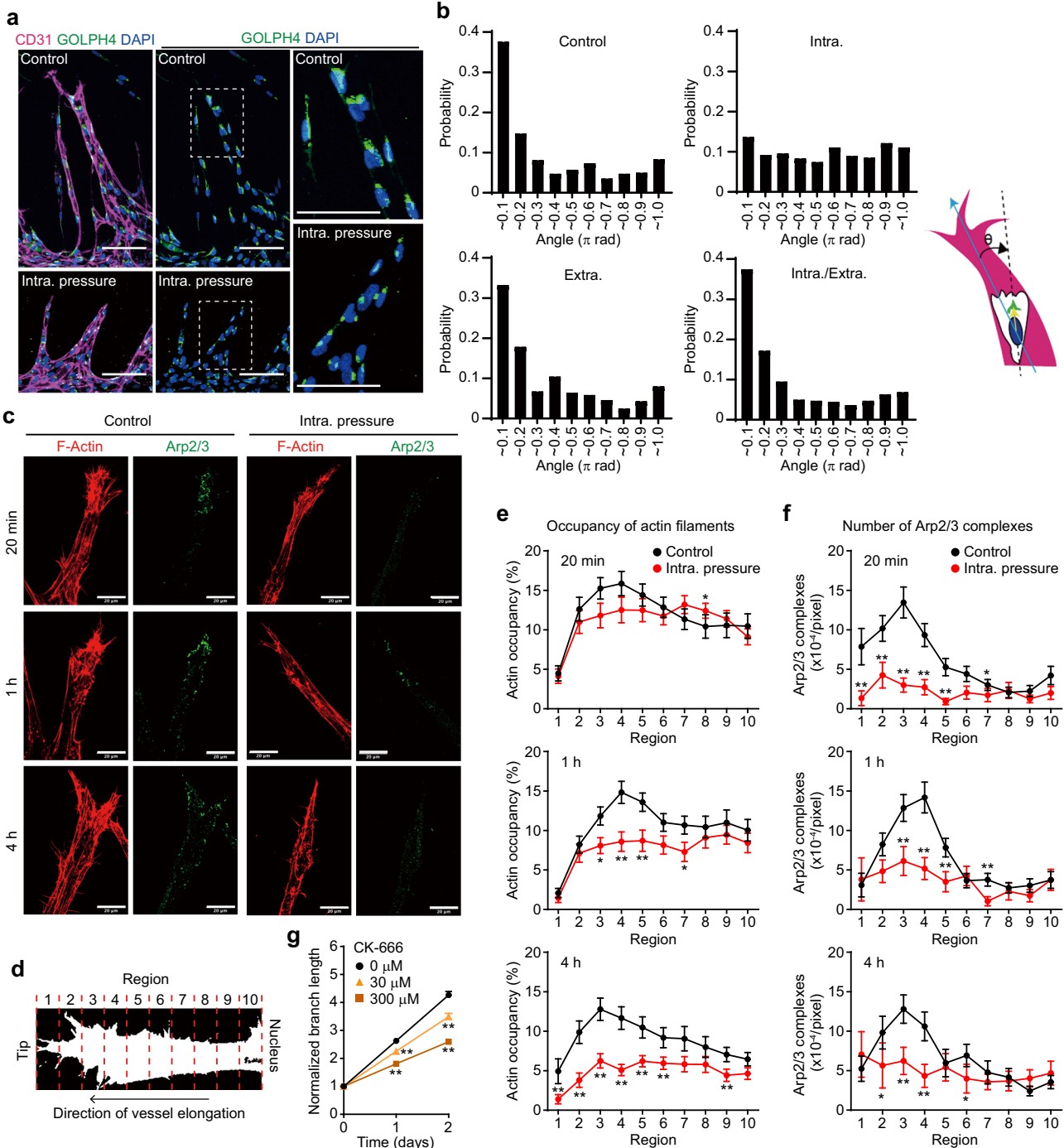

(Fig. 4g and Supplementary Fig. 11). These results suggest that IP load-induced stretching of ECs in angiogenic branches prevents leading edge localization of Arp2/3 complexes, thereby inhibiting actin polymerization and front–rear polarization leading to impaired blood vessel elongation.

**IP load-induced EC stretching suppresses elongation of upstream injured vessels by inhibiting Arp2/3 complex-mediated actin polymerization and front–rear polarization.** Then, we investigated whether IP load-induced EC stretching inhibits Arp2/3 complex-mediated actin polymerization at the leading edge and establishment of front–rear polarity which restricts elongation of upstream injured vessels during wound

angiogenesis. We first analyzed EC front–rear polarity in injured aISVs by using the *Tg(fli1a:EYFP-golgi);(fli1a:mCherry)* zebrafish line. Before the injury, ECs in ISVs directed their front–rear polarity against the blood flow direction (Fig. 5a–c), as previously reported[6]. Therefore, immediately after injury, the ECs in upstream and downstream injured vessels maintained their Golgi apparatus at the back and front of the nucleus toward the cutting direction, respectively (Supplementary Fig. 12a). In downstream injured vessels, ECs maintained their front–rear polarity throughout repair processes (Fig. 5a, c and Supplementary Movie 20). In contrast, the Golgi apparatus in ECs located in upstream injured vessels became randomly positioned during early-stage regeneration and its position never converged in the direction of vessel elongation (Fig. 5a, b and Supplementary Movie 20), suggesting that ECs in upstream

**Fig. 4 IP load-induced stretching of ECs impairs front–rear polarization and Arp2/3 complex-mediated actin polymerization during on-chip angiogenesis. a, b** Effect of pressure loading on EC front–rear polarity in angiogenic branches of on-chip angiogenesis. **a** Confocal z-projection images of angiogenic branches without (Control) and with (Intra. pressure) IP loading for 24 h. Left, merged images of CD31 (magenta), GOLPH4 (green), and DAPI (blue); right, merged images of GOLPH4 and DAPI. Boxed areas are enlarged on the right. **b** Histograms showing probability distributions of angles between the vessel elongation direction (blue arrow) and nucleus-Golgi vector (yellow arrow line) in ECs of angiogenic branches in response to indicated pressure load for 24 h (the number of ECs examined over 3 independent experiments: Control, $n = 503$; Intra., $n = 469$; Extra., $n = 325$; Intra./Extra., $n = 379$). **c–f** Spatiotemporal changes in distribution patterns of F-actin and Arp2/3 complexes at leading edges of angiogenic branches upon IP loading. **c** Confocal z-projection images of angiogenic branches without (Control) and with (Intra. pressure) IP loading for indicated periods. Left, F-actin; right, ARPC2 (Arp2/3). **d–f** The angiogenic branch between the leading edge and the nucleus was divided into 10 regions, as in **d**. Quantification of occupancy of F-actin (**e**) and the number of Arp2/3 complexes (**f**) in individual regions of angiogenic branches without (black) and with (red) IP loading, as in **c**. Data are means ± s.e.m. (the number of branches examined over 3 independent experiments: 20 min, $n = 27$ and 27; 1 h, $n = 36$ and 25; 4 h, $n = 30$ and 28, for Control and IP, respectively). *$p < 0.05$, **$p < 0.01$ versus Control by two-sided Mann–Whitney U test. **g** Elongation of angiogenic branches of on-chip angiogenesis in the absence (circle) and presence of 30 µM (triangle) and 300 µM (square) CK-666, an Arp2/3 complex inhibitor, was analyzed as in Fig. 2f. Data are means ± s.e.m. (the number of branches examined over 3 independent experiments: 0 µM, $n = 60$; 30 µM, $n = 60$; 300 µM, $n = 75$). **$p < 0.01$ versus 0 µM by two-way ANOVA followed by Tukey's test. For detailed statistics in **e–g**, see Supplementary Table 4. Source data are provided as a Source data file. Scale bars: 100 µm (**a**), 20 µm (**c**).

injured vessels lost front–rear polarity. However, since ECs in upstream injured vessels need to reverse their front–rear polarity to migrate forward for vessel repair, this time lag might result in inefficient elongation of upstream injured vessels. To exclude this possibility, we examined the elongation of injured aISVs in zebrafish larvae exhibiting defective blood flow-mediated EC polarization. For this purpose, we partially knocked out *aplnrb* using CRISPR/Cas9 technology (Supplementary Fig. 12b, c), since Aplnrb reportedly regulates EC polarization in response to blood flow[31]. As expected, larvae injected with low-dose *aplnrb* guide RNA exhibited mild defects in blood flow-induced EC polarization in aISVs (Supplementary Fig. 12d). Then, we injured their aISVs and analyzed the larvae in which ECs in upstream injured vessels positioned their Golgi apparatus in front or middle of the nucleus toward the vessel elongation direction immediately after injury. Even in those larvae, the injured ISVs were normally repaired, with vessel elongation being preferentially induced at sites downstream from blood flow, while injured upstream vessels did not efficiently elongate, as observed in control larvae (Supplementary Fig. 12e, f and Supplementary Movie 21). These results reveal that Aplnrb is not essential for EC polarization during repair processes of injured aISVs and further suggest that time lag to reverse front–rear polarity for vessel repair is not a cause of inefficient elongation of upstream injured vessels. Collectively, these findings indicate that IP loading disrupts EC front–rear polarity in upstream injured vessels.

Moreover, we analyzed the actin cytoskeleton in ECs constituting injured aISVs by imaging *Tg(fli1a:lifeact-mCherry); (fli1a:Myr-EGFP)* zebrafish larvae (Fig. 5d, e and Supplementary Movie 22). In downstream injured vessels, ECs actively extended actin-rich membrane protrusions toward the vessel elongation direction. However, the leading edge of ECs in upstream injured vessels was largely devoid of actin filaments, suggesting that IP load-induced EC stretching inhibits actin polymerization and front–rear polarization of ECs in upstream injured vessels.

We further investigated the role of Arp2/3 complexes in vessel elongation using *Tg(fli1a:EGFP-ARPC4);(fli1a:lifeact-mCherry)* larvae. During ISV development, Arp2/3 complexes frequently emerged at the leading edge of endothelial tip cells to promote actin polymerization (Supplementary Fig. 13a and Supplementary Movie 23). However, CK-666 significantly inhibited actin-based membrane protrusion formation and ISV elongation (Supplementary Fig. 13b-d and Supplementary Movie 24), indicating the involvement of Arp2/3 complex-mediated actin polymerization in angiogenesis. We also analyzed the localization of Arp2/3 complexes in injured ISVs (Fig. 5f, g and Supplementary Movie 25). In ECs of downstream injured vessels, Arp2/3

complexes frequently emerged at the leading edge and initiated actin polymerization to drive membrane protrusion. However, the Arp2/3 complex-mediated actin polymerization at the leading edge was less frequently induced in ECs of upstream injured vessels. Furthermore, CK-666 suppressed actin-rich membrane protrusion formation in downstream injured ISVs, thereby impairing elongation (Fig. 5h, i). These results indicate Arp2/3 complexes regulate elongation of downstream injured vessels by inducing actin polymerization, whereas in upstream injured vessels, IP load-induced EC stretching prevents leading edge localization of Arp2/3 complexes to inhibit the formation of actin-based protrusion and front–rear polarization, which in turn impairs vessel elongation.

**TOCA family of F-BAR proteins regulate vessel elongation during angiogenesis.** The next question to be addressed is how ECs sense IP load-induced mechanical stretch to regulate angiogenesis. Acute changes in membrane tension reportedly induce the disappearance of actin regulatory proteins from the leading edge[25,32–34]. A previous study identified the TOCA family of F-BAR proteins, consisting of TOCA-1, CIP4, and Formin-binding protein 17 (FBP17) (also known as Formin-binding protein 1 (FNBP1)), as a plasma membrane tension sensor involved in the leading-edge formation of directionally migrating cells[35]. TOCA family proteins bind to the plasma membrane through the N-terminal F-BAR domain and stimulate N-WASP-mediated Arp2/3 complex-dependent actin polymerization to promote directional cell migration. Importantly, increased plasma membrane tension results in detachment of TOCA family proteins from the plasma membrane, leading to suppression of Arp2/3 complex-mediated actin polymerization. Hence, we hypothesized that IP load-induced EC stretching might prevent leading edge localization of TOCA family proteins to inhibit N-WASP-mediated Arp2/3 complex-dependent actin polymerization and vessel elongation. To address this hypothesis, we first investigated the role of TOCA family proteins in angiogenesis. Among genes encoding TOCA family proteins, *CIP4* and *TOCA1*, but not *FBP17*, mRNAs were expressed in HUVECs (Supplementary Fig. 14a). In situ hybridization on zebrafish embryos also showed expressions of *toca1* and *cip4*, but not of *fbp17*, in blood vessels (Supplementary Fig. 14b). Consistently, publicly available scRNA-seq data of the tabula muris project showed that *Toca1* and *Cip4*, but not *Fbp17*, are preferentially expressed in ECs of various mouse organs[36]. Therefore, we investigated CIP4 and TOCA1 roles in angiogenesis by performing siRNA-mediated gene knockdown experiments (Supplementary Fig. 15a, b). In an on-chip angiogenesis model, knockdown of both *CIP4* and

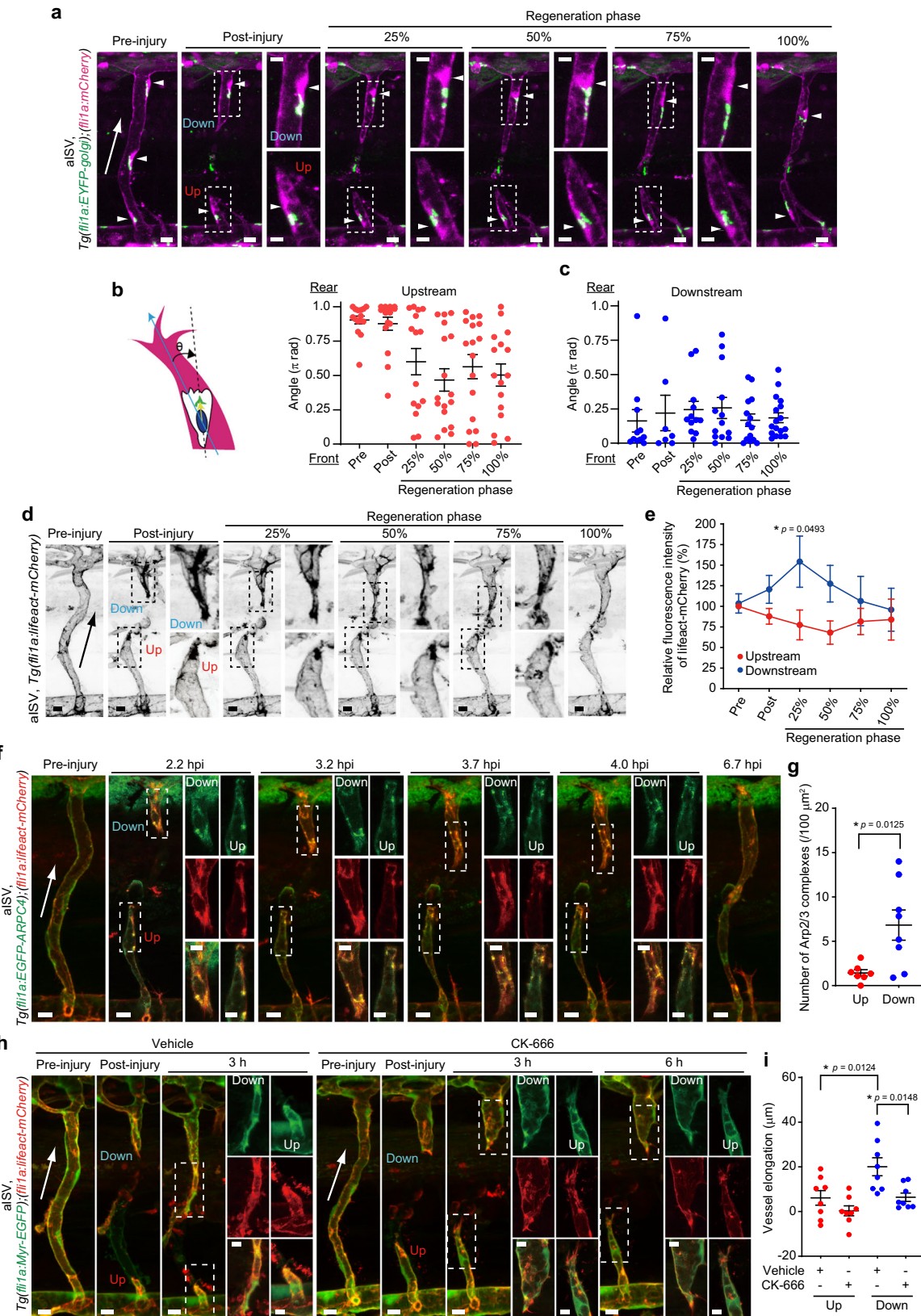

*TOCA1* significantly suppressed vessel elongation, while it was moderately inhibited by silencing of either gene (Fig. 6a, b). Impaired elongation of angiogenic branches formed by *CIP4* and *TOCA1*-double knockdown ECs was rescued by lentivirus-mediated expression of N-terminally EGFP-tagged TOCA1 (EGFP-TOCA1) in ECs (Fig. 6c, d and Supplementary Fig. 15c).

Although around 30% of ECs used in the rescue experiment were not transduced to express EGFP-TOCA1, those expressing EGFP-TOCA1 preferentially occupied a tip position in angiogenic branches constituted by *CIP4* and *TOCA1*-double knockdown ECs and even in those by control siRNA-transduced ECs (Fig. 6e, f). These results suggest that CIP4 and TOCA1 regulate blood vessel

**Fig. 5 Upstream injured vessels fail to elongate due to loss of front–rear polarity and impaired Arp2/3 complex-mediated actin polymerization in the ECs. a** Confocal z-projection images of pre- and post-injured aISVs in 3 dpf Tg(fli1a:EYFP-Golgi);(fli1a:mCherry) larva and its subsequent time-lapse images at indicated stages of repair. Images of EYFP-Golgi (green) and mCherry (magenta) are merged. Arrowheads, nuclei. Boxed areas are enlarged on the right. **b, c** Quantification of front–rear EC polarity in upstream (**b**) and downstream (**c**) injured aISVs. Front-rear polarity was analyzed by measuring the angles between the vessel elongation direction and the nucleus-Golgi vector in ECs. Each dot represents an individual cell. Error bars indicate mean ± s.e.m. [the number of ECs over 12 animals: upstream and downstream at pre ($n = 15$ and 11), post ($n = 16$ and 7), 25% ($n = 15$ and 12), 50% ($n = 17$ and 13), 75% ($n = 17$ and 14), 100% ($n = 16$ and 17)]. **d** Confocal z-projection images of pre- and post-injured aISVs in 3 dpf Tg(fli1a:lifeact-mCherry) larva and its subsequent time-lapse images are shown as in **a**. **e** Quantification of F-actin in upstream (red) and downstream (blue) injured aISVs. Relative fluorescence intensity of Lifeact-mCherry in vascular regions within 20 μm from the leading edge is expressed as a percentage of that in pre-injured vessels. Data are means ± s.e.m. [the number of animals: upstream and downstream at pre, post and 50% ($n = 5$), at 25 and 75% ($n = 4$), at 100% ($n = 3$)]. **f** Confocal z-projection images of pre-injured aISV (left) and injured aISVs at the elapsed time following the injury (hpi) in 3 dpf Tg(fli1a:EGFP-ARPC4);(fli1a:lifeact-mCherry) larva are shown as in **a**. Images of EGFP-ARPC4 (green) and Lifeact-mCherry (red) are merged. EGFP (top), mCherry (middle), and their merged (bottom) images corresponding to boxed areas are enlarged on the right. **g** Number of EGFP-ARPC4-labeled Arp2/3 complexes colocalizing with Lifeact-mCherry-marked F-actin in vascular regions within 20 μm from the leading edge at the 25% reparative phase. Each dot represents an individual injured vessel (red and blue, upstream and downstream injured vessels, respectively). Data are means ± s.e.m. (Up and Down, $n = 7$ and 8 animals). **h** Confocal z-projection images of pre- and post-injured aISVs and those 3 and 6 h after the beginning of treatment with vehicle or CK-666 in 3 dpf Tg(fli1a:lifeact-mCherry);(fli1a:Myr-EGFP) larva are shown in **f**. **i** Amounts of elongation of upstream (red) and downstream (blue) injured aISVs 3 h after the beginning of treatment with vehicle or 200 μM CK-666. Each dot represents an individual injured vessel (red and blue, upstream and downstream injured vessels, respectively). Data are means ± s.e.m. (Up and Down, $n = 8$ animals). Statistical significance was determined by two-way ANOVA followed by Sidak's test (**e**), Welch's two-sided t test (**g**), and one-way ANOVA followed by Tukey's test (**i**). *$p < 0.05$. Arrows indicate blood flow direction (**a, d, f, h**). Source data are provided as a Source data file. Scale bars: 10 μm (**a, d, f, h**), 5 μm (enlarged images in **a, f, h**).

elongation during angiogenesis. Therefore, we further investigated their roles in in vivo angiogenesis by generating toca1[nf4] and cip4[nf5] zebrafish mutants using CRISPR/Cas9 technology (Supplementary Fig. 16). toca1[nf4/nf4] mutants at 28 hpf did not show lethality and a clear morphological abnormality, and at least some mutants survived until adulthood (Fig. 6g). However, ISVs at 28 hpf were significantly shorter in toca1[nf4/nf4] mutants than in wild-type embryos (Fig. 6h, i). toca1[nf4/+] embryos also showed a tendency for the delayed formation of ISVs compared to wild type (Fig. 6h, i). However, ISVs developed normally in cip4[nf5/nf5] mutants, while tending to show mild ventral curvature phenotypes (Supplementary Fig. 17). These results suggest the TOCA family of F-BAR proteins to regulate vessel elongation during angiogenesis and that predominantly Toca1 regulates ISV angiogenesis in zebrafish embryos.

**CIP4 and TOCA1 induce N-WASP-mediated Arp2/3 complex-dependent actin polymerization at the leading edge of ECs during angiogenesis.** Next, we investigated whether CIP4 and TOCA1 induce branch elongation by stimulating N-WASP-mediated Arp2/3 complex-dependent actin polymerization. To this end, we first examined their localization in angiogenic branches using an on-chip angiogenesis model. In elongating vessels, endogenous CIP4, N-terminally EGFP-tagged CIP4 (EGFP-CIP4), and EGFP-TOCA1 largely colocalized with Arp2/3 complexes at the leading edge of endothelial tip cells, where actin polymerization actively occurred (Fig. 7a, b and Supplementary Fig. 18a–d). In addition, we generated the Tg(fli1a:EGFP-toca1) zebrafish line to analyze the localization of EGFP-Toca1 in ECs during in vivo angiogenesis and found that EGFP-Toca1 frequently emerged at the leading edge of endothelial tip cells during ISV development, similarly to Arp2/3 complexes (Fig. 7c, Supplementary Fig. 13a, and Supplementary Movies 23 and 26). Therefore, we next tested whether CIP4 and TOCA1 induce recruitment of Arp2/3 complexes to the leading edge of ECs to promote vessel elongation. Localization of Arp2/3 complexes at the leading edge of endothelial tip cells was severely inhibited by knockdown of both CIP4 and TOCA1 (Fig. 7d, e). Defective localization of Arp2/3 complexes at the leading edge of the CIP4 and TOCA1-double knockdown ECs was rescued by ectopic expression of EGFP-TOCA1, which localized at the leading edge of endothelial tip cells (Fig. 7d–f). In addition, the EGFP-TOCA1-

mediated rescue of defective vessel elongation was blocked by CK-666 treatment (Fig. 7g). These findings suggest that TOCA1 and CIP4 recruit Arp2/3 complexes to EC leading edge to promote actin polymerization and vessel elongation.

We further investigated the role of N-WASP in Arp2/3 complex-mediated actin polymerization in elongating branches. In an on-chip angiogenesis model, N-WASP colocalized with EGFP-TOCA1 and Arp2/3 complexes at the leading edge of endothelial tip cells (Fig. 7h and Supplementary Fig. 18e). Treatment with an N-WASP inhibitor, wiskostatin, suppressed branch elongation during angiogenesis (Fig. 7i, j). Consistently, wiskostatin-mediated inhibition of N-WASP prevented leading edge localization of Arp2/3 complexes, thereby suppressing actin-based protrusion formation (Fig. 7k, l). Collectively. these results indicate that CIP4 and TOCA1 regulate angiogenesis by stimulating N-WASP-mediated Arp2/3 complex-dependent actin polymerization to induce vessel elongation.

**IP load-induced EC stretching causes detachment of CIP4 and TOCA1 from the leading edge to inhibit elongation of upstream injured vessels.** Finally, we investigated whether IP load-induced EC stretching prevents leading edge localization of CIP4 and TOCA1 to suppress angiogenic branch elongation. Thus, we first examined the effect of EC stretching on the localization of Arp2/3 complexes, CIP4, and EGFP-TOCA1 in directionally migrating ECs by performing an in vitro wound-healing assay. Arp2/3 complexes, CIP4 and EGFP-TOCA1 colocalized with actin filaments at the leading edge in directionally migrating cells at the scratch edges (Fig. 8a–d and Supplementary Fig. 19a–d). Since Arp2/3 complexes are also localized at cell–cell junctions, we carefully analyzed their cellular localization by mosaically expressing either EGFP-ARPC4 or Azami-Green in ECs. EGFP-ARPC4, but not Azami-Green, localized in junctional regions at the leading edge of the follower cells (Supplementary Fig. 19e, f), suggesting localization of Arp2/3 complexes at junctional-associated intermittent lamellipodia[16]. Exposure of directionally migrating ECs to continuous stretch (10% strain) resulted in the removal of Arp2/3 complexes, CIP4, and EGFP-TOCA1 from the leading edge and reduced the amount of actin filaments, suggesting that CIP4 and TOCA1 function as a stretch sensor to regulate Arp2/3 complex-mediated actin polymerization (Fig. 8a–d and Supplementary Fig. 19a–d). Our on-chip

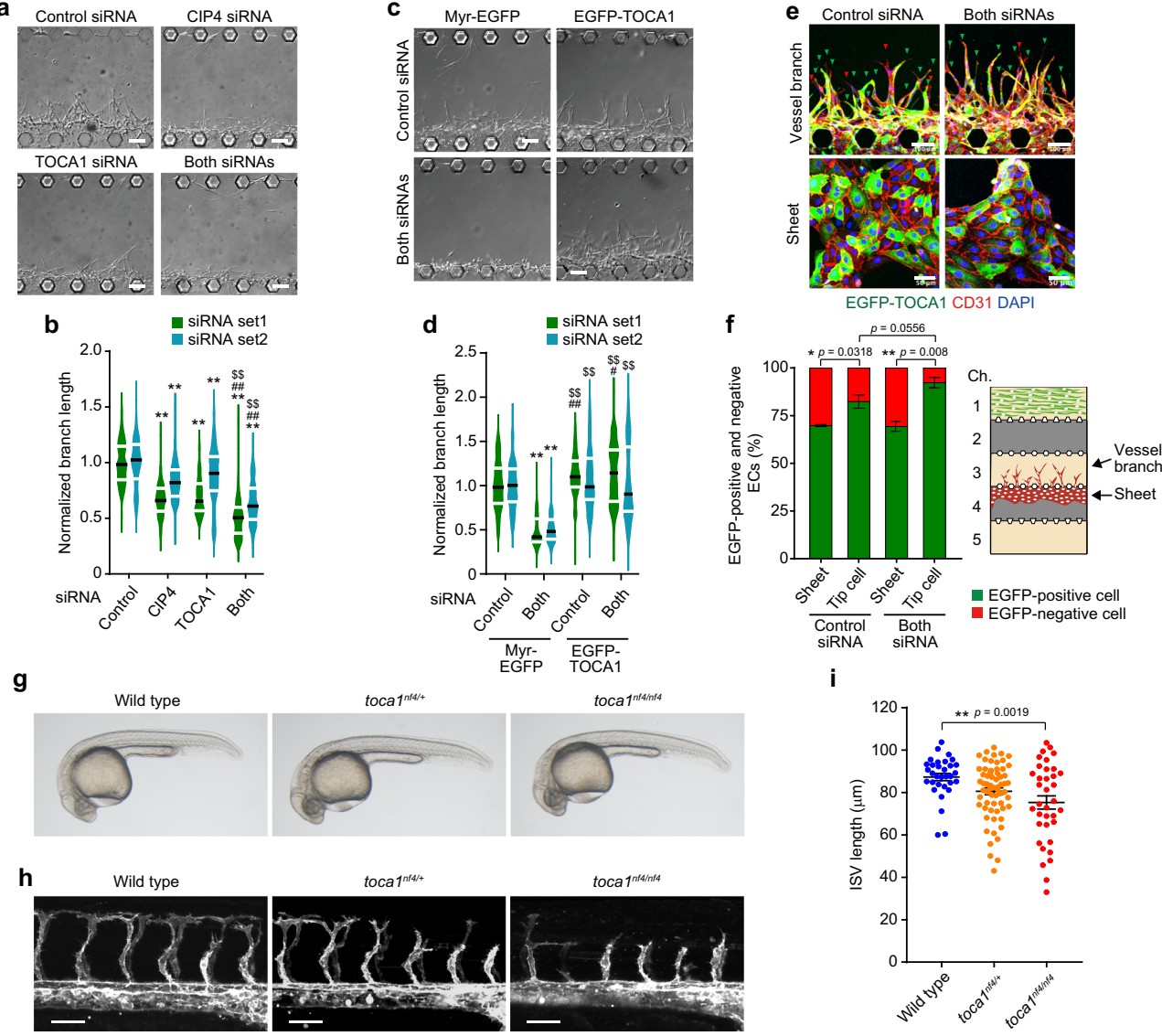

**Fig. 6 TOCA family of F-BAR proteins regulate vessel elongation during angiogenesis. a, b** Effects of silencing of *CIP4* and *TOCA1* on branch elongation in on-chip angiogenesis. **a** DIC images of angiogenic branches constituted by ECs transfected with the control siRNA, *CIP4* siRNA, *TOCA1* siRNA, or both siRNA (siRNA set1) after 3 days of culture. **b** Violin plots of branch lengths as in **a** (green). Lengths of branches constituted by ECs transfected with another set of siRNAs (siRNA set2) are also shown (blue). Black and white lines indicate the median and quartiles, respectively (each group, *n* = 75 branches examined over 3 independent experiments). **\*\****p* < 0.01 versus Control, ##*p* < 0.01 versus CIP4, $$*p* < 0.01 versus TOCA1 by two-way ANOVA followed by Tukey's test. For detailed statistics, see Supplementary Table 4. **c, d** Rescue of impaired elongation of angiogenic branches constituted by *CIP4* and *TOCA1*-double knockdown ECs by exogenous expression of EGFP-TOCA1. **c** DIC images of angiogenic branches constituted by siRNA set1-transfected ECs infected with lentiviruses encoding either Myr-EGFP or EGFP-TOCA1 after 3 days of culture. **d** Violin plots of branch lengths as in **c** (green). Lengths of angiogenic branches constituted by siRNA set2-transfected ECs are also shown (blue). Black and white lines indicate the median and quartiles, respectively (each group, *n* = 120 branches examined over 4 independent experiments). **\*\****p* < 0.01 versus Myr-EGFP/Cont.; ##*p* < 0.01 versus Myr-EGFP/Cont.; $$*p* < 0.01 versus Myr-EGFP/Both by the Steel–Dwass test. For detailed statistics, see Supplementary Table 4. **e, f** Positions of EGFP-TOCA1-transduced ECs in on-chip angiogenic branches. **e** Confocal *z*-projection images of angiogenic branches constituted by ECs transfected with control siRNA (Control siRNA) or siRNA set1 (Both siRNAs) and infected with lentivirus encoding EGFP-TOCA1 in channel 3 of the microfluidic device (upper panel) and those of sheet-forming ECs in channel 4 (lower panel). Green, EGFP-TOCA1; red, CD31; blue, DAPI. Green and red arrowheads indicate EGFP-positive and negative tip ECs, respectively. **f** Percentages of EGFP-positive (green) and negative (red) cells localized at a tip position in angiogenic branches (Tip cell) and percentages of EGFP-positive (green) and negative (red) cells in ECs forming the sheet (Sheet), as in **e**. Data are means ± s.e.m. (each 5 independent experimental samples). *\*p* < 0.05, *\*\*p* < 0.01 by two-sided Mann–Whitney *U* test. **g** Representative bright-field images of wild-type (left), *toca1^{nf4/+}* (middle), *and toca1^{nf4/nf4}* (right) zebrafish embryos at 28 hpf. **h** Confocal *z*-projection images of trunk vasculature in 28 hpf *Tg(fli1a:lifeact-mCherry)* embryos with the indicated genotypes. Lateral views with anterior to the left. **i** Quantification of ISV length, as in **h**. Each dot represents an individual embryo. Data are means ± s.e.m. (wild type, *n* = 32; *toca1^{nf4/+}*, *n* = 63; *toca1^{nf4/nf4}*, *n* = 36). **\*\****p* < 0.01 by one-way ANOVA followed by Tukey's test. Source data are provided as a Source data file. Scale bars: 100 μm (**a, c, e**), 50 μm (**h**).

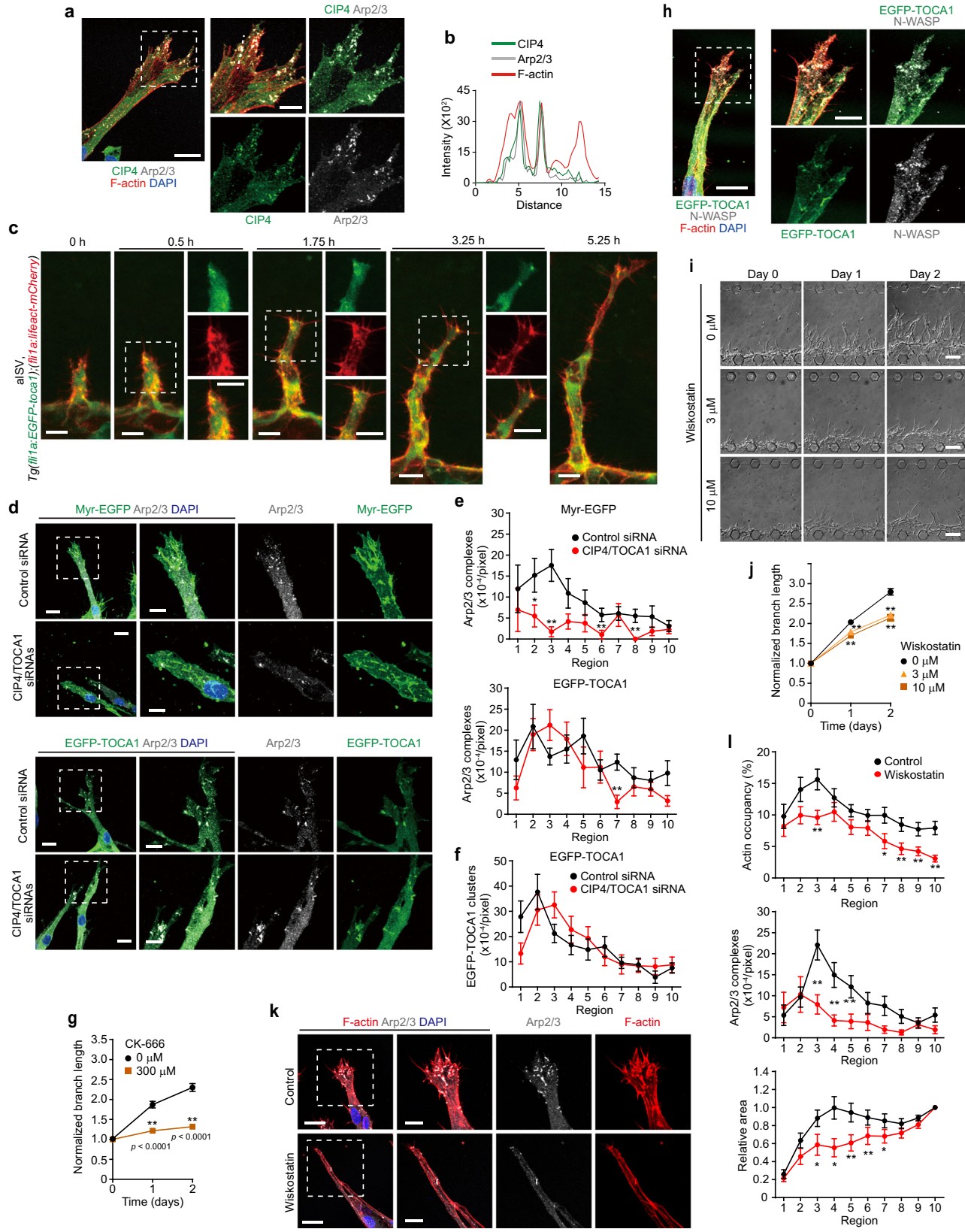

angiogenesis assay also revealed that IP loading for 20 min, which induced EC stretching in angiogenic branches (Fig. 3k–n and Supplementary Fig. 7f, g), resulted in the disappearance of both CIP4 and Arp2/3 complexes from the leading edge (Fig. 8e, f). Similarly, EGFP-TOCA1 and EGFP-CIP4 colocalized with Arp2/3 complexes immediately disappeared from the EC leading edge

upon IP loading (Supplementary Fig. 20). To confirm whether an IP load-induced increase in plasma membrane tension affects the localization of CIP4 and Arp2/3 complexes, we treated angiogenic branches with a hypotonic medium to induce cell swelling-mediated plasma membrane stretching. The addition of a hypotonic medium resulted in the disappearance of CIP4 and Arp2/3

**Fig. 7 CIP4 and TOCA1 induce recruitment of Arp2/3 complexes to the leading edge of ECs during angiogenesis. a** Confocal *z*-projection image of on-chip angiogenic branch. Green, CIP4; gray, ARPC2 (Arp2/3); red, F-actin; blue, DAPI. **b** Fluorescence intensity profiles along the lines indicated in merged image in **a**. **c** Confocal *z*-projection images of ISVs in 23 hpf *Tg(fli1a:EGFP-toca1);(fli1a:lifeact-mCherry)* zebrafish embryos and subsequent time-lapse images at the elapsed time at the top are as in Fig. 5f. Green, EGFP-Toca1; red, Lifeact-mCherry. **d** Confocal *z*-projection images of angiogenic branches constituted by ECs transfected with control siRNA (Control siRNA) or siRNA set1 (CIP4/TOCA1 siRNAs) and infected with lentiviruses encoding either Myr-EGFP (upper) or EGFP-TOCA1 (lower). Green, EGFP; gray, ARPC2 (Arp2/3); blue, DAPI. **e, f** Number of Arp2/3 complexes (**e**) and that of EGFP-TOCA1 clusters (**f**) in angiogenic branches, as in **d**, are shown as in Fig. 4d–f. Data are means ± s.e.m. (the number of branches examined over 3 independent experiments: Control and CIP4/TOCA1, *n* = 18 and 18 in Myr-EGFP and *n* = 19 and 20 in EGFP-TOCA1, respectively). *\*p* < 0.05, *\*\*p* < 0.01 versus control siRNAs. **g** Elongation of angiogenic branches constituted by siRNA set1-transfected ECs infected with lentivirus encoding EGFP-TOCA1 in the absence (square) and presence of 300 μM CK-666 (circle) is shown as in Fig. 4g. Data are means ± s.e.m. (each group, *n* = 75 branches examined over 4 independent experiments). *\*\*p* < 0.01 versus 0 μM. **h** Colocalization of EGFP-TOCA1 (green), N-WASP (gray), and F-actin (red) at the leading edge of angiogenic branch is shown as in **a**. **i** Serial DIC images of angiogenic branches in the presence of 0 (circle), 3 (triangle), and 10 μM (square) wiskostatin are shown as in Fig. 2e. **j** Quantification of branch lengths as in **i** is shown as in Fig. 4g. Data are means ± s.e.m. (each group, *n* = 75 branches examined over 4 independent experiments). *\*\*p* < 0.01 versus 0 μM. **k** Confocal *z*-projection images of angiogenic branches after 3 days of culture with and without 10 μM wiskostatin. Red, F-actin; gray, ARPC2 (Arp2/3); blue, DAPI. **l** Quantification of F-actin occupancy (top), Arp2/3 complex number (middle), and leading-edge areas (bottom) in angiogenic branches without (Control) and with (Wiskostatin) wiskostatin treatment as in **k** are shown as in Fig. 4d–f (Control and Wiskostatin, *n* = 19 and 18 branches examined over 3 independent experiments). The leading-edge area in each region is shown relative to that observed in Region 10. Data are means ± s.e.m. *\*p* < 0.05, *\*\*p* < 0.01 versus Control. Boxed areas are enlarged on the right (**a**, **d**, **h**, **k**). Statistical significance was determined by two-sided Mann–Whitney *U* test (**e**, **f**, **l**), two-way ANOVA followed by Sidak's multiple comparison test (**g**), and two-way ANOVA followed by Tukey's test (**j**). For detailed statistics in **e**, **f**, **j**, **l**, see Supplementary Table 4. Source data are provided as a Source data file. Scale bars: 20 μm (**a**, **d**, **h**, **k**), 10 μm (**c** and enlarged images in **a**, **d**, **h**, **k**), 5 μm (enlarged images in **c**), 100 μm (**i**).

---

complexes from the leading edge of angiogenic branches (Fig. 8g, h). These results suggest that IP load-induced EC stretching prevents leading edge localization of CIP4 and TOCA1 to inhibit Arp2/3 complex-mediated actin polymerization and vessel elongation during angiogenesis.

We further investigated whether detachment of TOCA family proteins from the EC leading edge causes impaired elongation of upstream injured vessels during wound angiogenesis. For this purpose, we analyzed the localization of EGFP-Toca1 in the injured ISVs (Fig. 9a, b and Supplementary Movie 27). In downstream injured ISVs, EGFP-Toca1 was frequently observed at the leading edge of endothelial tip cells where actin polymerization actively occurred. In clear contrast, EGFP-Toca1 rarely emerged at the leading edge of ECs in upstream injured ISVs. We also analyzed the repair processes of injured ISVs in *toca1^{nf4}* mutants (Fig. 9c, d). Elongation of downstream injured vessels was significantly slower in *toca1^{nf4/nf4}* and *toca1^{nf4/+}* than in wild-type larvae. However, upstream injured vessels only marginally elongated irrespective of the *toca1* genotypes. Collectively, these results suggest that TOCA family proteins not only promote wound angiogenesis but also act as sensors for IP load-induced EC stretching to inhibit elongation of upstream injured vessels.

## Discussion

Herein, we uncovered a novel role of EC stretching caused by the loading of blood flow-driven IP in directed EC migration and vascular morphogenesis during wound angiogenesis (Fig. 10). During wound angiogenesis, new blood vessels are formed via elongation of severed vessels as well as sprouting from uninjured vessels. Surprisingly, by utilizing a recently developed live-imaging system for adult zebrafish[20], we discovered that elongation of injured vessels is preferentially induced downstream from blood flow, whereas blood flow-driven IP loading suppresses elongation of upstream injured vessels by stretching ECs. By exploiting an originally and newly developed in vitro angiogenesis system in a microfluidic device, we further demonstrated that IP load-induced EC stretching inhibits actin polymerization and front–rear polarization to restrict vessel elongation. Furthermore, our in vitro and in vivo experiments identified F-BAR proteins, TOCA1 and CIP4, as key regulators of the actin cytoskeleton required for directional EC migration during angiogenesis and

revealed that they also act as sensors for IP load-induced stretching of ECs to inhibit elongation of upstream injured vessels during wound angiogenesis.

TOCA1 and CIP4 play important roles in the directional migration of ECs during angiogenesis. TOCA family proteins regulate not only endocytosis but also actin-based membrane protrusion and cell migration[37–42]. Consistently, our data showed that TOCA1 and CIP4 bind to plasma membranes and recruit Arp2/3 complexes to induce actin-based protrusion and vessel elongation. TOCA family proteins bind to N-WASP through their C-terminal SH3 domain to induce Arp2/3 complex-mediated actin polymerization[43]. They also interact with activated Cdc42, an activator of N-WASP, via their G protein-binding homology region 1 domains[43]. Therefore, TOCA1/CIP4, N-WASP, and Cdc42 may cooperatively induce Arp2/3 complex-mediated actin polymerization at the leading edge of directionally migrating ECs to promote angiogenesis. Consistently, EC-specific deletion of *Cdc42* caused defective EC migration during retinal angiogenesis in mice[44].

In addition to TOCA family proteins, other BAR domain proteins may regulate angiogenesis by inducing actin polymerization and EC migration. The F-BAR protein NOSTRIN, originally identified as a modulator of endothelial nitric oxide synthase, reportedly regulates angiogenesis by facilitating fibroblast growth factor-mediated activation of Rac1[45]. Since NOSTRIN shares a common domain architecture with TOCA family proteins and interacts with N-WASP[46], it may regulate angiogenesis cooperatively with TOCA family proteins. Furthermore, some BAR domain proteins such as IRSp53, srGAP, and Pacsin reportedly induce actin-based membrane protrusion through nucleation-promoting factors such as N-WASP and WAVE[19], although their roles in angiogenesis have not as yet been studied extensively. Therefore, EC migration and vessel elongation during angiogenesis might be regulated by multiple BAR domain proteins.

TOCA1 and CIP4 sense IP load-induced EC stretching to become detached from the leading-edge membrane, thereby suppressing actin-based protrusion to inhibit elongation of upstream injured vessels during wound angiogenesis. Plasma membrane tension has been suggested to play a crucial role in front–rear polarization of directionally migrating cells by inhibiting actin polymerization[24,25,33,47]. A previous study showed

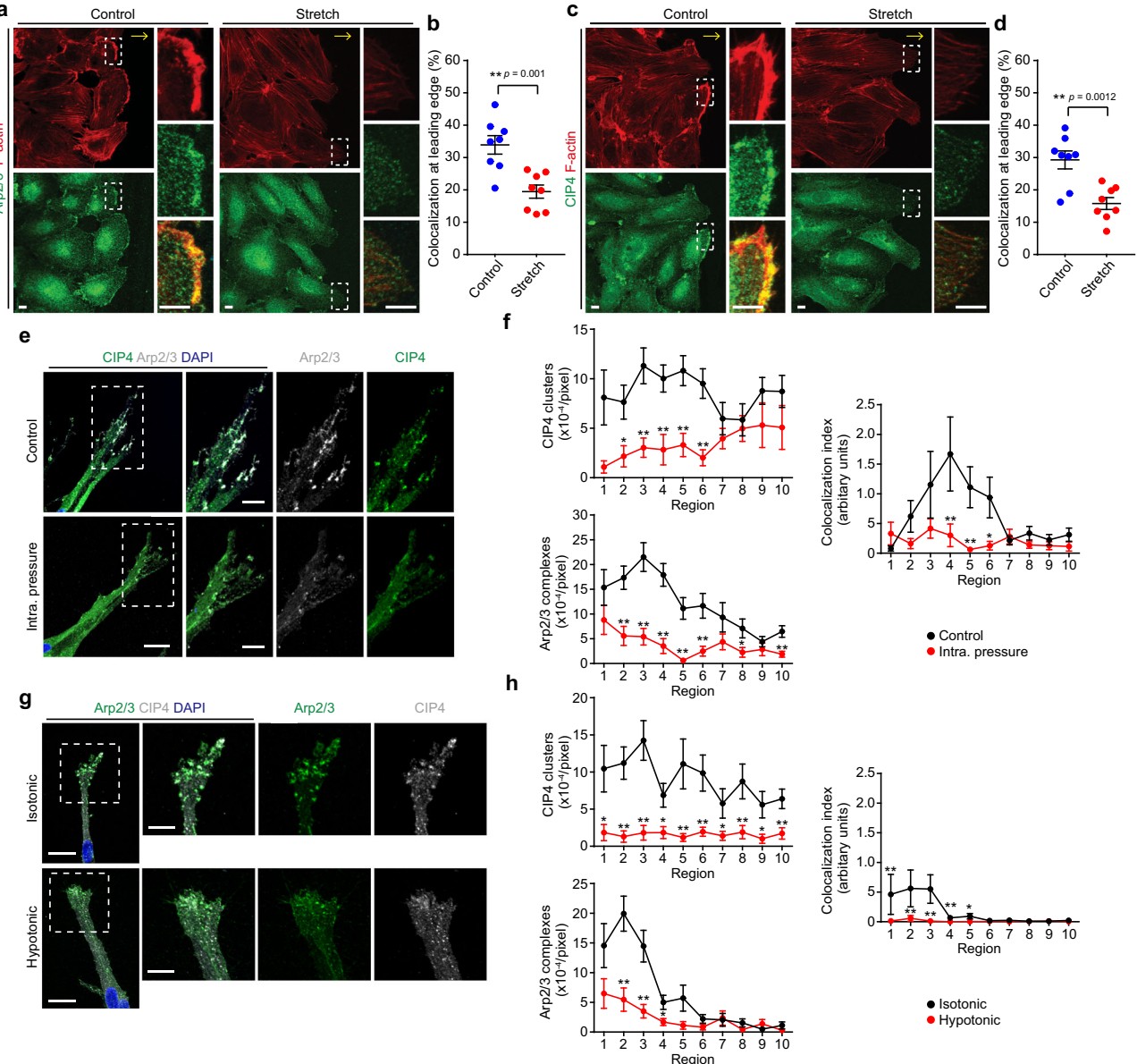

**Fig. 8 IP load-induced EC stretching causes detachment of CIP4 and TOCA1 from the leading edge of ECs to inhibit branch elongation. a–d** Effects of biaxial stretching on colocalization of Arp2/3 complexes (**a**, **b**) and CIP4 (**c**, **d**) with F-actin at leading edges of HUVECs directionally migrating on stretching chambers. Confocal fluorescence images of HUVECs exposed to continuous biaxial stretch for 3 min after being stretched to 10% over 8 min (Stretch) or kept under static conditions (Control). Left upper, F-actin (red); left lower, ARPC2 (Arp2/3, green) (**a**) or CIP4 (green) (**c**). F-actin (upper), ARPC2 or CIP4 (middle), and merged (lower) images of boxed areas are enlarged on the right. In **a**, **c**, yellow arrows indicate the direction of cell migration. **b**, **d** Quantification of Arp2/3 complexes (**b**) and CIP4 (**d**) colocalized with F-actin at leading edges of HUVECs, as in **a**, **c**, respectively. Each dot represents an individual confocal image (blue, Control; red Stretch). Data are means ± s.e.m ($n = 8$ regions examined over 2 independent experiments for each). \*\*$p < 0.01$ by two-sided $t$ test. **e**, **f** Effects of IP loading on localization of CIP4 and Arp2/3 complexes at the leading edge of on-chip angiogenic branch. **e** Confocal $z$-projection images of angiogenic sprouts loaded without (upper) and with IP (lower) for 20 min. Boxed areas are enlarged on the right. Green, CIP4; gray, ARPC2 (Arp2/3); blue, DAPI. **f** Quantification of the number of CIP4 clusters (upper left), that of Arp2/3 complexes (lower left), and the degree of their colocalization (right) in individual regions of angiogenic branches, as in **e**. The number of CIP4 clusters and that of Arp2/3 complexes are shown as in Fig. 4d–f. The colocalization index of CIP4 clusters and Arp2/3 complexes is given as described in "Methods." Data are means ± s.e.m (Control and IP, $n = 23$ and 19 branches examined over 3 independent experiments). \*$p < 0.05$, \*\*$p < 0.01$ versus Control. **g**, **h** Effects of hypotonic stimulation on localization of CIP4 and Arp2/3 complexes at the leading edge of an on-chip angiogenic branch. **g** Confocal $z$-projection images of angiogenic sprouts treated with isotonic medium (upper) and hypotonic medium (lower) for 5 min are as in **e**. **h** Quantitative data of **g** are as in **f**. Data are means ± s.e.m. (Isotonic and Hypotonic, $n = 20$ and 22 branches examined over 3 independent experiments). \*$p < 0.05$, \*\*$p < 0.01$ versus Control. Statistical significance was determined by two-sided Mann–Whitney $U$ test (**f**, **h**). For detailed statistics in **f**, **h**, see Supplementary Table 4. Source data are provided as a Source data file.

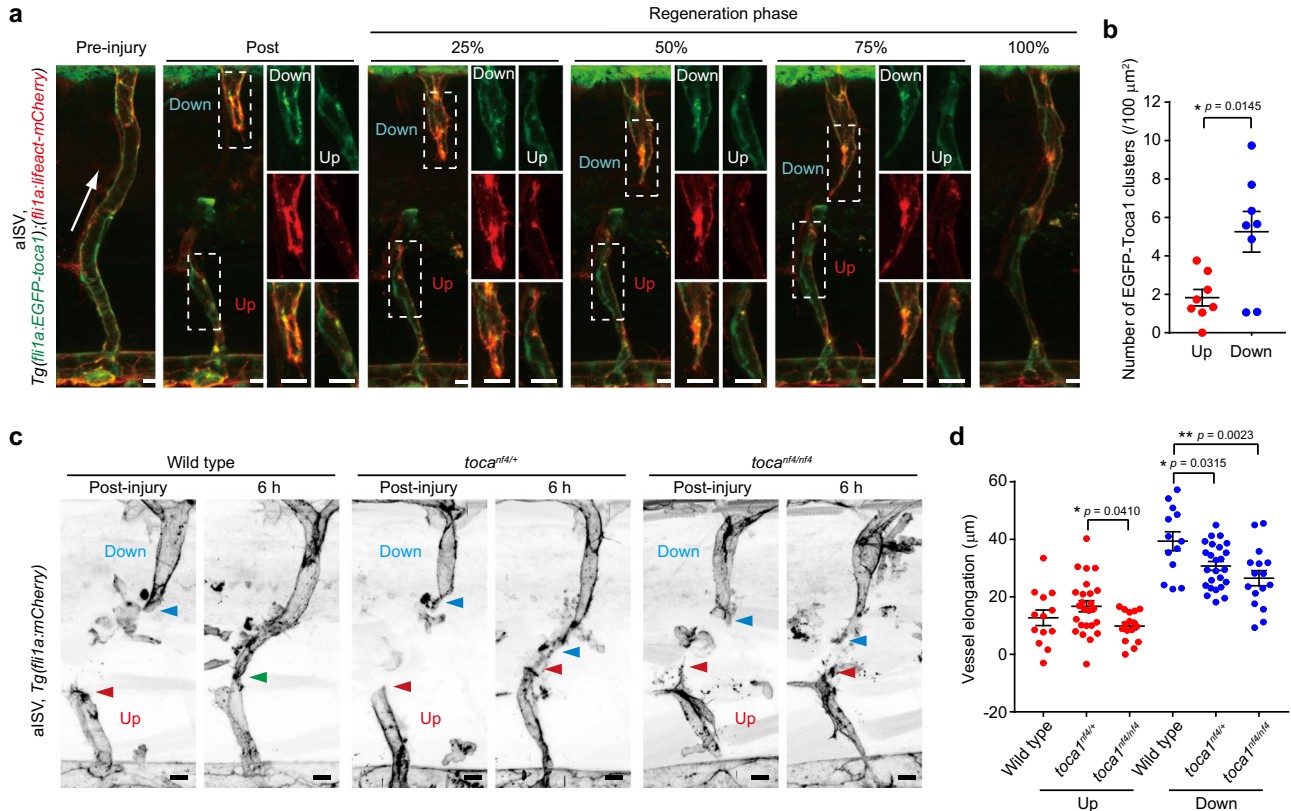

**Fig. 9 IP load-induced EC stretching inhibits leading edge localization of Toca1 to suppress elongation of upstream injured vessels. a, b** Localization of EGFP-Toca1 in injured aISVs. **a** Confocal z-projection images of pre- and post-injured aISVs in the *Tg(fli1a:EGFP-toca1);(fli1a:lifeact-mCherry)* larva at 3 dpf and its subsequent time-lapse images at the indicated stages of repair. Lateral view, anterior to the left. Images of EGFP-Toca1 (green) and Lifeact-mCherry (red) fluorescence are merged. EGFP (top), mCherry (middle), and their merged (bottom) images corresponding to boxed areas are enlarged on the right. Arrow indicates blood flow direction. **b** Number of EGFP-Toca1 clusters colocalizing with Lifeact-mCherry-marked F-actin in vascular regions within 20 μm from the leading edge at approximately 25% reparative phase. Each dot represents an individual injured vessel (red and blue, upstream and downstream injured vessels, respectively). Data are means ± s.e.m. (*n* = 8 animals for each). *$p < 0.05$ by Welch's two-sided *t* test. **c, d** Elongation of injured aISVs in *toca1* mutant zebrafish. **c** Confocal z-projection images of injured aISVs and those after 6 h in wild-type (left), *toca1^{nf4/+}* (middle), and *toca1^{nf4/nf4}* (right) larval zebrafish at 3 dpf with *Tg(fli1a:lifeact-mCherry)* background are as in **a**. Red and blue arrowheads, leading edges of upstream and downstream injured vessels, respectively; green arrowhead, anastomotic site of the injured vessel. **d** Quantification of elongation lengths of upstream and downstream injured aISVs in wild type, *toca1^{nf4/+}*, and *toca1^{nf4/nf4}* embryos, as in **c**. Each dot represents an individual larva (red and blue, upstream and downstream injured vessels, respectively). Data are means ± s.e.m. (wild type, *n* = 13; *toca1^{nf4/+}*, *n* = 25; *toca1^{nf4/nf4}*, *n* = 16). *$p < 0.05$, **$p < 0.01$ by one-way ANOVA followed by Tukey's test among the upstream or downstream group. Source data are provided as a Source data file. Scale bars: 10 μm (**a, c**), 5 μm (enlarged images in **a**).

that TOCA family proteins act as a sensor for plasma membrane tension to regulate the polarized actin polymerization required for directional cell migration[35]. In migrating cells, TOCA family proteins stimulate N-WASP-mediated actin polymerization to induce membrane protrusion at the leading edge. Subsequently, leading edge protrusion increases membrane tension at the cell rear, thereby inhibiting actin polymerization and the formation of a secondary leading edge to maintain front–rear polarity. Therefore, IP load-induced EC stretching may ectopically increase plasma membrane tension to induce the disappearance of TOCA1 and CIP4 from the leading edge, which in turn inhibits leading edge protrusion and disrupts front–rear polarity, leading to defective elongation of upstream injured vessels. Thus, this study will contribute to understanding molecular and regulatory mechanisms underlying multicellular morphogenetic processes by cell stretch-induced membrane tension.

How is blood flow-driven IP loading involved in different types of angiogenesis? We observed preferential elongation of upstream injured vessels after injury of cutaneous capillaries and ISVs. Previous research showed that ECs mainly sprouted from veins,

not arteries, to repair injured blood vessels during zebrafish fin regeneration[48]. Since blood is expected to be pumped mainly to the arteries compared to the veins, blood flow-driven IP loading might also restrict EC sprouting from arteries during fin regeneration. Thus, IP loading appears to suppress elongation of upstream injured vessels during wound angiogenesis. In contrast, it is thought that IP loading does not inhibit the elongation of vessel sprouts during developmental angiogenesis. Indeed, blood flow reportedly drives lumen expansion of elongating vessels during sprouting angiogenesis[7]. Why do blood vessel sprouts elongate even in the presence of IP loading during developmental angiogenesis? IP applied to vascular sprouts during developmental angiogenesis might be lower than that loaded onto upstream injured vessels during wound angiogenesis, since blood flow is fully established in pre-injured vessels. Another possibility is that EC states may differ between developmental and wound angiogenesis. Since ECs in mature blood vessels are maintained in a quiescent state, they are likely to remain quiescent immediately after injury. In contrast, activated ECs exist in elongating vessels during developmental angiogenesis. Therefore, quiescent ECs

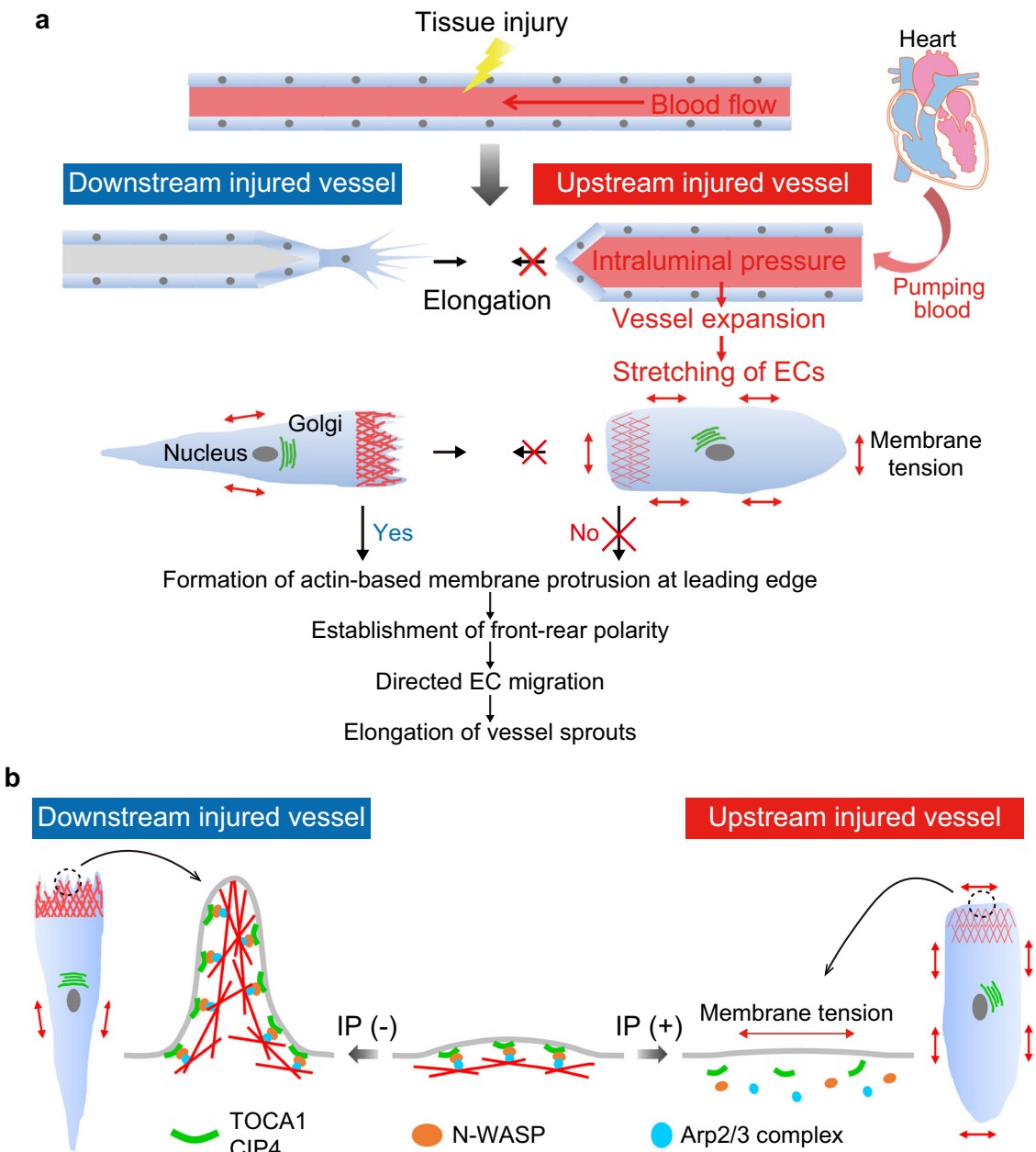

**Fig. 10 Schematic representation of proposed model accounting for how blood flow-driven intraluminal pressure regulates angiogenesis during wound healing. a** When blood vessels are damaged by tissue injury, elongation of severed blood vessels is actively induced at sites downstream from blood flow, whereas vessels located upstream do not elongate efficiently. ECs in downstream injured vessels extend leading-edge protrusions and establish front–rear polarity to directionally migrate and elongate vessel sprouts. In clear contrast, in upstream injured vessels, IP load induces vessel expansion and EC stretching, inhibiting actin-based protrusion and front–rear polarization, thereby impairing vessel elongation. **b** In downstream injured vessels, TOCA1 and CIP4 bind to plasma membranes at the leading edge of ECs and recruit N-WASP and Arp2/3 complexes to induce actin-based membrane protrusion, promoting directional EC migration and vessel elongation. However, in upstream injured vessels, IP load-induced EC stretching prevents leading edge localization of TOCA1 and CIP4, thereby inhibiting actin-based protrusion and front–rear polarization, thus impairing vessel elongation.

might be more sensitive to IP load-induced cell stretching than activated ECs. However, further investigation is required to resolve this issue.

Extravascular environments and mural cell coverage may also impact the inhibitory effects of IP loading on vessel elongation. Vascular wall stretching is thought to depend on several parameters including the pressure gap between IP and EP, vascular wall visco-elasticities, and extravascular interstitial tissue[49,50]. Indeed, our in vitro studies revealed that IP loading did not inhibit vessel elongation in the presence of high interstitial

pressure (Fig. 3h, i). In addition, IP loading is expected not to efficiently induce vessel expansion and EC stretching if the stiffness of surrounding tissues is high. Therefore, blood flow-driven IP loading and the extravascular tissue environments might regulate vessel elongation, in a coordinated fashion, during angiogenesis. In addition, mural cells such as pericytes and smooth muscle cells surround ECs in vivo and regulate blood vessel diameter. Hence, mural cell coverage might affect IP load-induced suppression of vessel elongation. In this regard, in zebrafish larvae at 3 dpf, most of aISVs were wrapped by mural cells,

while many venous ISVs lacked mural cell coverage (Supplementary Fig. 21). Nevertheless, differential elongation of injured blood vessels was similarly induced when either arterial or venous ISVs were severed by laser ablation. These results indicate that blood flow-driven IP loading suppresses elongation of upstream injured vessels regardless of mural cell coverage, at least, in ISVs. Thus, careful examination is necessary to elucidate the role of mural cells in regulating IP load-induced suppression of vessel elongation.

What is the biological significance of IP load-mediated inhibition of vessel elongation during wound angiogenesis? IP loading might inhibit elongation of upstream injured vessels to suppress blood leakage because the heart continuously pumps blood into upstream injured vessels. Alternatively, IP load-mediated inhibition of vessel elongation may prevent excessive blood vessel formation and thereby allow well-organized vascular networks to be established. During tumor angiogenesis, disorganized and immature blood vessels form in tumor tissues, leading to tumor progression and metastasis[51]. Such abnormal vessels are highly permeable, raising interstitial pressure possibly over IP on tumor vessels[52]. Therefore, such a tumor microenvironment may prevent IP load-induced stretching of ECs, thereby inducing excessive sprouting and elongation of tumor vessels to generate the characteristically disorganized and immature tumor blood vessels. Indeed, interstitial hypertension is reportedly associated with tumor neovascular development[53]. However, further studies are needed to clarify the biological significance of IP load-mediated inhibition of vessel elongation, as seen in angiogenesis during wound healing.

The discovery of an unexpected role of IP in regulating angiogenesis might contribute to developing novel effective therapies for non-healing wounds and ischemic diseases. Because IP loading restricts elongation of blood vessels, lowering IP on blood vessels might promote angiogenesis, allowing the generation of functional vascular networks in non-healing wound tissues. In addition, therapeutic angiogenesis is widely applied to ischemic diseases. Thus, therapeutic angiogenesis might be improved by controlling IP on blood vessels. On the other hand, tumor vasculature normalization, a new therapeutic strategy for cancer, is expected to enhance cancer immunotherapy and improve the effectiveness of anticancer treatments[54]. Because low IP loading on tumor vessels and high interstitial pressure may lead to abnormal tumor vascular formation, as discussed above, increasing the IP load acting on tumor vessels may promote tumor vessel normalization.

## Methods

**Zebrafish husbandry**. Zebrafish of the AB strain (Danio rerio) were bred and maintained under standard conditions. Embryos and larvae were staged by hours post-fertilization (hpf) and days post-fertilization (dpf) at 28.5 °C[55]. For live-imaging of transgenic zebrafish larvae, they were maintained from 24 hpf in E3 embryo medium (5 mM NaCl, 0.17 mM KCl, 0.33 mM CaCl$_2$, 0.33 mM MgSO$_4$) containing 0.03% N-phenylthiourea to prevent pigment formation. Animal experiments were approved by the animal committees of the National Cerebral and Cardiovascular Center (#15010) and the Nippon Medical School (#28-010, #2020-054) and performed by following the guidelines of the National Cerebral and Cardiovascular Center and the Nippon Medical School.

**Plasmids**. A cDNA encoding EYFP-Golgi, a fusion protein consisting of the EYFP coupled to a sequence encoding the N-terminal 81 amino acids of human beta-1,4-galactosyltransferase, was amplified by PCR using pEYFP-Golgi (Clontech, Takara Bio Inc.) as a template. To construct the pTol2-fli1:EYFP-Golgi plasmid, the EYFP-Golgi cDNA was subcloned into the pTolfli1 vector[56] derived from the pTol-fli1epEGFPDest plasmid, a gift from N. Lawson (University of Massachusetts Medical School)[57]. The Tol2 vector system was kindly provided by K. Kawakami (National Institute of Genetics, Japan)[58]. The EYFP-Golgi cDNA was also subcloned into the CSII-CMV-MCS-IRES2-Bsd to construct the CSII-CMV-EYFP-Golgi-IRES2-Bsd plasmid. The CSII-CMV-MCS-IRES2-Bsd lentiviral expression

vector and the packaging plasmids (pCAG-HIVgp, pCMV-VSV-G) were kindly provided by Dr. H. Miyoshi (BioResource Center, RIKEN).

A cDNA fragment encoding human ARPC4 was amplified from a cDNA library derived from HUVECs by PCR and cloned into a pEGFP-C1 vector (Clontech, Takara Bio Inc.) to construct the pEGFP-C1-ARPC4 that encodes N-terminally EGFP-tagged ARPC4 (EGFP-ARPC4). Then, the pTol2-fli1:EGFP-ARPC4 plasmid was constructed by subcloning the EGFP-ARPC4 cDNA into the pTolfli1 vector. The EGFP-ARPC4 cDNA was also subcloned into the CSII-CMV-MCS-IRES2-Bsd to construct the CSII-CMV-EGFP-ARPC4-IRES2-Bsd plasmid. A lentiviral expression vector encoding Azami-Green was kindly provided by Dr. S. Higashiyama (Ehime University).

A cDNA fragment encoding the coding sequence of human CIP4 (NM_004240) was amplified from a cDNA library derived from HUVECs by PCR and cloned into a pEGFP-C1 vector to construct the pEGFP-C1-CIP4 that encoded N-terminally EGFP-tagged CIP4 (EGFP-CIP4). A human TOCA1 cDNA was amplified by PCR using a pCS2+MT-hToca1-wt plasmid (Addgene plasmid #33030) as a template and cloned into a pEGFP-C1 vector to construct the pEGFP-C1-TOCA1 that encoded N-terminally EGFP-tagged TOCA1 (EGFP-TOCA1). The EGFP-CIP4 and EGFP-TOCA1 cDNAs were also subcloned into the CSII-CMV-MCS-IRES2-Bsd to construct the CSII-CMV-EGFP-CIP4-IRES2-Bsd and CSII-CMV-EGFP-TOCA1-IRES2-Bsd plasmids. A Myr-EGFP cDNA encoding myristoylation signal-tagged EGFP was amplified by PCR using pEGFP-N1-Myr vector[56] and subcloned into the CSII-CMV-MCS-IRES2-Bsd to construct the CSII-CMV-Myr-EGFP-CIP4-IRES2-Bsd vector.

cDNA fragments encoding zebrafish *toca1*, *cip4*, and *fbp17* were amplified from a cDNA library derived from zebrafish embryos by PCR using the primers listed in Supplementary Table 2, and cloned into a pCR4 Blunt TOPO vector (Invitrogen). A cDNA fragment encoding the coding sequence of zebrafish *toca1* (NM_001003634) was amplified from a cDNA library derived from zebrafish embryos by PCR and cloned into a pEGFP-C1 vector to construct the pEGFP-C1-toca1 that encodes N-terminally GFP-tagged Toca1. Then, the pTol2-fli1:EGFP-toca1 plasmid was constructed by subcloning the EGFP-toca1 cDNA into the pTolfli1 vector.

**Transgenic zebrafish lines**. The Tol2 transposon system was used to generate transgenic zebrafish lines[58]. Tol2 transposase mRNAs were in vitro transcribed with SP6 RNA polymerase from NotI-linearized pCS-TP vector using the mMESSAGE mMACHINE kit. To generate the *Tg(fli1a:EYFP-Golgi)*[ncv510Tg], *Tg(fli1a:EGFP-ARPC4)*[nf5Tg], and *Tg(fli1a:EGFP-toca1)*[nf10Tg] zebrafish lines, the pTol2-fli1:EYFP-Golgi, the pTol2-fli1:EGFP-ARPC4, and pTol2-fli1:EGFP-toca1 plasmids (each 25 pg) were, respectively, microinjected along with Tol2 transposase RNA (25 pg) into one-cell stage embryos of the wild type strain, AB. The embryos showing transient expression of GFP in the vasculature were selected, raised to adulthood, and crossed with wild type AB to identify germline transmitting founder fishes.

The *Tg(kdrl:EGFP)*[s843] and *Tg(fli1a:EGFP)*[y1] zebrafish lines were kindly provided by D. Y. Stainier (Max Planck Institute for Heart and Lung Research) and N. Lawson (University of Massachusetts Medical School), respectively. The *Tg(fli1a:Myr-EGFP)*[ncv2Tg56], *Tg(flt1enh:mCherry)*[ncv30Tg59], *Tg(fli1a:Lifeact-mCherry)*[ncv7Tg14], *Tg(fli1a:mCherry)*[ncv50120], *TgBAC(pdgfrb:eGFP)*[ncv23Tg59], and *Tg(fli1a:h2b-mCherry)*[ncv31Tg60] fish lines were previously described in detail. Throughout the text, all transgenic lines used in this study are simply described without their allele name.

**Generation of *toca1* and *cip4* mutant zebrafish lines**. The *toca1* and *cip4* zebrafish mutants were generated by the Alt-R CRISPR/Cas9 technology according to the method reported by Hoshijima et al.[61]. Target-specific AltR crRNAs and common Alt-R tracrRNA were purchased from IDT and dissolved in the duplex buffer to prepare 100 μM stock solutions. Equal volumes of stock solutions were mixed and annealed by heating at 95 °C for 5 min, followed by gradual cooling to 25 °C at 0.1 °C/s, incubation at 25 °C for 5 min, and rapid cooling to 4 °C to prepare the dgRNAs. The *toca1* and *cip4* dgRNAs target the exon 2 of the *toca1* and the exon 4 of the *cip4* as described in Supplementary Fig. 16a, e, respectively. Cas9 protein (Alt-R S.p. Cas9 nuclease, v.3, IDT) was dissolved in phosphate-buffered saline (PBS) to prepare 25 μM stock solution. One μL of 25 μM dgRNA was mixed with 1 μL of 25 μM Cas9 solution, 0.5 μL H$_2$O, and 2.5 μL of 2× fish injection buffer (0.5% phenol red, 240 mM KCl, 40 mM HEPES, pH7.4). The ribonucleoprotein (RPN) complex solution was incubated at 37 °C for 5 min, cooled down to room temperature (R.T.), and then used for microinjection.

Zebrafish embryos at the one-cell stage were injected with approximately 1 nl of 5 μM *toca1* dgRNA:Cas9 and *cip4* dgRNA:Cas9 RNP complex solutions and raised to adulthood. F0 founder zebrafish carrying the indel mutations were identified by performing the T7 endonuclease-I (T7EI) cleavage assay using the Alt-R Genome Editing Detection Kit (IDT) or T7 Endonuclease I reaction Mix (Nippon Gene). Each embryo obtained by crossing the F0 founder with wild-type fish was lysed in 50 mM NaOH at 98 °C for 10 min, cooled at 4 °C, and neutralized with 1 M Tris-HCl (pH 8). After centrifugation at 15,000 × g for 5 min, the supernatant was subjected to genomic PCR using the primer sets that anneal to the sequence flanking the target region, as shown in Supplementary Fig. 16a, e (F-toca1 and R-toca1 for *toca1* mutant; F-cip4 and R-cip4 for *cip4* mutant). Then, the PCR

products were denatured, annealed, digested with T7EI, and subsequently analyzed by agarose gel electrophoresis according to the manufacturer's instructions. The identified F0 founders were crossed with wild-type zebrafish to obtain the F1 fish carrying the mutations. Genomic DNA extracted from the tail fins of adult F1 zebrafish was subjected to genomic PCR using the same primer sets. Then, the PCR products were sequenced to identify frameshift mutations. Finally, we established the $toca1^{nf4}$ mutant, which carries a 6-bp deletion and a 2-bp insertion, and the $cip4^{nf5}$ mutant, which carries a 5-bp deletion (Supplementary Fig. 16a, e). Mutated alleles in $toca1^{nf4}$ and $cip4^{nf5}$ encode 41 mutated amino acids from Lys33 and 29 mutated amino acids from Arg106, followed by premature stop codons, respectively (Supplementary Fig. 16d, h).

For the genotyping of the $toca1^{nf4}$ mutant, PCR products amplified from genomic DNA using a primer set comprising F-toca1 and R-toca1 were digested with Hpy188I to yield 133-bp and 474-bp fragments for the wild type allele and 133-bp, 290-bp, and 184-bp fragments for the mutant allele (Supplementary Fig. 16b, c). For the genotyping of the $cip4^{nf5}$ mutants, PstI digestion of PCR products amplified from genomic DNA using a primer set comprising F-cip4 and R-cip4 yields 118-, 164-, and 366-bp fragments for the wild-type allele and 118- and 530-bp fragments for the mutant allele (Supplementary Fig. 16f, g).

**Microscopes used for in vivo imaging.** Confocal images were obtained using the following microscopes; a FLUOVIEW FV1000 confocal upright microscope equipped with GaAsP photomultiplier tubes and operated with FLUOVIEW FV10ASW software (Olympus), a FLUOVIEW FV1200 confocal upright microscope equipped with multi-alkali tubes and operated with FLUOVIEW FV10ASW software (Olympus), a FLUOVIEW FV3000 confocal upright microscope equipped with GaAsP photomultiplier tubes and operated with FLUOVIEW FV31S-SW software (Olympus), and an A1R MP+ multiphoton confocal upright microscope (Nikon) equipped with GaAsP photomultiplier tubes (Nikon) and a piezo nano-positioning stage (Nano-F450, Mad City Lab) and operated with NIS-Elements software (Nikon). The FV1000, FV1200, and FV3000 microscopes were equipped with a water-immersion ×20 objective lens (XLUMPlanFL N, 1.00 NA, Olympus) for many imaging cases, and a ×20 water-immersion objective lens (UMPlanFL N, 0.50 NA, Olympus), a water-immersion ×10 (UMPlanFL N, 0.30 NA, Olympus), ×40 (LUMPlanFL N, 0.80 NA, Olympus), or ×60 objective lens (LUMFL N, 1.10 NA, Olympus), when indicated. A water-immersion ×25 objective lens (CFI75 Apochromat 25XC W 1300, 1.10 NA, Nikon) was used in the A1R MP+ microscope. In some experiments, two-photon imaging was performed using an FVMPE-RS multiphoton upright microscope equipped with a water-immersion ×25 objective lens (XLPLN25XWMP2, 1.05 NA, Olympus) and operated with FLUOVIEW FV31S-SW software (Olympus).

Excitation wavelengths of lasers applied for the FV1000 and FV1200 microscopes were 473 nm for EGFP, EYFP, and Alexa Fluor 488, and 559 nm for mCherry and Alexa Fluor 546. Those for the FV3000 and A1R MP+ (when used for confocal imaging mode) microscopes were 405 nm for DAPI, Qdot 655, Qdot 705, 488 nm for EGFP, EYFP, Alexa Fluor 488, and Azami-Green, 561 nm for mCherry and rhodamine, and 640 nm for Alexa Fluor 633. For the use of the FVMPE-RS microscope, the two-photon excitation wavelength of the laser for EGFP was 920 nm. Fluorescence images were acquired sequentially to avoid cross-detection of the fluorescent signals when multiple lasers were employed.

The images obtained by confocal and two-photon imaging were processed using the FLUOVIEW FV10ASW software, FLUOVIEW FV31S-SW software (Olympus), NIS-Elements software, Volocity 6.3 3D Imaging Analysis software (Quorum Technologies), Fiji image processing software package (http://fiji.sc.), MetaMorph 7.8 software (Molecular Devices), and/or Photoshop CS4 and 2021 software (Adobe Systems Corp.).

**In vivo imaging of adult zebrafish.** Male zebrafish of the AB strain at 3–18 months post-fertilization were used for in vivo imaging. Adult zebrafish were anesthetized in 0.06% 2-phenoxyethanol (Sigma-Aldrich) in fish water, placed in a fish chamber (plastic case: 100 mm × 100 mm × 29 mm, AS ONE), and immobilized by covering the trunk and tail regions, except for the pectoral fins and the injured area, with 1.5% low-melting agarose (Thermo Fisher). After insertion of the intramedic polyethylene tube (inner diameter: 0.86 mm; outer diameter: 1.27 mm, BD) into the mouths of the fish, they were orally perfused with fish water containing 0.035–0.04% 2-phenoxyethanol at a speed of 5.5–6.0 mL/min and then subjected to cutaneous wounding and live imaging. For recovery from anesthesia, the fish were perfused with fish water without 2-phenoxyethanol for 5–10 min.

To analyze the repair process of the injured skin vessels (Figs.1a–c and 2c, d, Supplementary Fig. 1b, and Supplementary Movies 2–4 and 12), the Tg(kdrl:EGFP) or Tg(kdrl:EGFP);(fli1a:h2b-mCherry) adult zebrafish mounted on a fish chamber were imaged approximately every 6 or 12 h until the times indicated in the figures. To perform long-term confocal time-lapse imaging of wound angiogenesis (Supplementary Fig. 1a and Supplementary Movie 1), the adult zebrafish with skin injury were imaged every 30 min for approximately 14–21 h. After resting in the fish tank for 3–10 h, they were re-mounted on a fish chamber and subjected to the next round of live imaging. This imaging cycle was repeated eight times.

To morphologically analyze the injured blood vessels in the skin of adult zebrafish (Fig. 3b, c), the Tg(kdrl:EGFP) or Tg(flt1^{enh}:mCherry) fish were anesthetized at 22–48 h post-injury (hpi) and fixed in 4% paraformaldehyde in PBS

(Nacalai Tesque) at 4 °C overnight. After washing with PBS, the fish were mounted in 3% low-melting agarose dissolved in PBS on a 60 mm plastic petri dish, submerged in PBS, and subjected to confocal imaging using the FV1200 microscope equipped with either ×20 or ×40 objective lens.

**In vivo imaging of zebrafish embryos and larvae.** Zebrafish embryos and larvae of the AB strain at the stages indicated in the figure legends were used for in vivo imaging. Dechorionated embryos and larvae were anesthetized in 0.032–0.064% tricaine (Sigma-Aldrich) in E3 medium. Then, they were mounted in 1% low-melting agarose (Nacalai Tesque) dissolved in E3 medium poured on a 35 mm glass-base dish (Iwaki, ASAHI GLASS Company, Ltd.) for imaging with the Olympus confocal and multiphoton microscopes or on a 60 mm plastic petri dish for imaging with the Nikon multiphoton confocal microscopes, and sub-merged in E3 medium supplemented with 0.016% tricaine (E3 imaging medium). To prevent pigmentation in the embryos and larvae, 0.03% N-phenylthiourea was included in the fish water and E3 imaging medium during rearing and imaging, respectively, when required.

To analyze the elongation of injured arterial and venous intersegmental vessels (ISVs) (Fig. 1d, e, Supplementary Fig. 1c, d, and Supplementary Movies 5 and 6) and the morphology of the injured aISVs (Fig. 3d, e), two-photon fluorescence images of the pre- and post-injured ISVs in the Tg(fli1a:EGFP) or Tg(fli1a:Myr-EGFP) larvae at 2.5–3.5 dpf were taken with the FVMPE-RS microscope. Subsequently, the repair processes of injured ISVs were imaged every 15–20 min for 8–12 h using the FV1200 microscope. The morphological analysis of injured ISVs was performed at the timing of approximately 50% of the regeneration phase (Fig. 3d, e).

To analyze the front–rear polarity of ECs in the injured blood vessels (Fig. 5a–c, Supplementary Fig. 12, and Supplementary Movies 20 and 21), the pre- and post-injured aISVs in the Tg(fli1a:EYFP-golgi);(fli1a:mCherry) larvae at 2.5–3.5 dpf were imaged by the A1R MP+ microscope. Subsequently, confocal fluorescence images of their repair processes were acquired every 20 min for 14 h. Similarly, confocal fluorescence images of the pre- and post-injured aISVs and their subsequent time-lapse images in the Tg(fli1a:lifeact-mCherry);(fli1a:Myr-EGFP) larvae at 2.5–3.5 dpf (every 50 min for 12 h), in the Tg(fli1a:EGFP-ARPC4);(fli1a:lifeact-mCherry) larvae at 2.5–3.5 dpf (every 15 min for 12 h), and in the Tg(fli1a:EGFP-toca1);(fli1a:lifeact-mCherry) larvae at 2.5–3.5 dpf (every 15 min for 12 h) were acquired to quantify the actin cytoskeleton in the injured blood vessels (Fig. 5d, e and Supplementary Movie 22) and to analyze the localization of the EGFP-ARPC4 (Fig. 5f, g and Supplementary Movie 25) and EGFP-Toca1 (Fig. 9a, b and Supplementary Movie 27) in the injured blood vessels, respectively.

To analyze the localization of EGFP-ARPC4 (Supplementary Fig. 13a and Supplementary Movie 23) and EGFP-Toca1 (Fig. 7c and Supplementary Movie 26) in ECs during ISV development, confocal fluorescence images of the ISVs in the Tg(fli1a:EGFP-ARPC4);(fli1a:lifeact-mCherry) and Tg(fli1a:EGFP-toca1);(fli1a:lifeact-mCherry) embryos at 23 hpf were acquired every 10 and 15 min, respectively, for 6.5 h by the FV3000 microscope.

To investigate the inhibitory effect of an Arp2/3 complex inhibitor, CK-666 (Merck), on the elongation of injured blood vessels (Fig. 5h, i), confocal fluorescence images of the pre- and post-injured aISVs in the Tg(fli1a:Myr-EGFP);(fli1a:lifeact-mCherry) larvae at 3 dpf were acquired by the A1R MP+ microscope. Treatment with vehicle or 200 μM CK-666 was started just after imaging the post-injured ISVs. Then, 3 and 6 h later, the larvae were imaged to analyze elongation of the injured ISVs. In some experiments, the larvae at 3 h after the beginning of CK-666 treatment were fixed with 4% paraformaldehyde in PBS at 4 °C overnight, re-soaked in PBS, and imaged by the FV1000 or FV3000 microscope. To investigate the inhibitory effect of CK-666 on vessel elongation during ISV angiogenesis (Supplementary Fig. 13b-d and Supplementary Movie 24), the Tg(fli1a:Myr-EGFP);(fli1a:lifeact-mCherry) embryos at 24 hpf were treated with vehicle or 200 μM CK-666. Then, confocal fluorescence images of the ISVs were acquired every 10 min for 3–4 h by the FV1000 or FV3000 microscope.

**Injury of blood vessels in zebrafish.** To injure the cutaneous tissues in adult zebrafish, we removed 3–4 scales and introduced wounds onto the flank of adult zebrafish using fine forceps (Dumont No. 5) under the FV1200 microscope equipped with a ×20 water-immersion objective lens (UMPlanFL N, 0.50 NA), which resulted in the loss of most of the epidermal and dermal cells, but rarely damaged either the subcutaneous adipocytes or the underlying muscle tissue. To injure a single capillary in the skin, we removed a scale and severed the capillary using microsurgical needles included in ETHILON Nylon Suture 8-0 or 11-0 (Ethicon Inc.) under a confocal microscope. To remove IP from the upstream injured vessels as shown in Fig. 2c, the blood vessels located upstream from and supplying blood to the injured vessels were additionally severed, leading to the formation of upstream injured vessels without IP loading. In some cases, IP-unloaded injured upstream vessels had side branches lacking a blood supply. Upstream vessels without IP loading were used for the subsequent experiments when the vessels including side branches contained more than 35 ECs.

To injure ISVs in zebrafish larvae, anesthetized larvae were mounted on a 35 mm glass-base dish or a 60 mm plastic dish as described above, and subjected to laser ablation of a single ISV located in the middle of the trunk. To achieve this, the middle region of the ISV was ablated by irradiation employing a laser with a

wavelength of 720 nm for 1 s 2–5 times using the A1R MP+ or FVMPE-RS microscope. This ablation procedure usually resulted in a loss of a single EC.

**Analysis of hypoxia in the skin of adult zebrafish**. Hypoxic states of injured skin of adult zebrafish were determined by performing pimonidazole staining with a Hypoxyprobe-1 Kit (Hypoxyprobe, Inc.) (Supplementary Fig. 2). Adult *Tg(kdrl:EGFP)* zebrafish, with and without skin injury, were kept in a fish tank for 24–70 h. Under anesthesia, they were intraperitoneally injected with 15 μL of 15 mg/mL pimonidazole solution using a microsyringe (Ito Corp.) twice (2 and 24 h before fixation).

To stain the hypoxic tissues, the pimonidazole-injected zebrafish were anesthetized and fixed in 4% paraformaldehyde in PBS (Nacalai Tesque) at 4 °C overnight and then washed with PBS. To decalcify the scales, the fixed zebrafish were immersed in 0.5 M EDTA/2Na (pH = 7.4) in distilled water and continuously inverted at RT for 4 days with one EDTA solution change after 2 days. After washing with distilled water 5 times for 3 h, uninjured skin tissues and those included in the wounded region approximately 7 mm × 7 mm × 2 mm in size were excised from the fish with a razor and then washed with PBS. Next, the tissue blocks were permeabilized with PBS containing 0.25% trypsin and 1 mM EDTA/2Na at RT for 20 min, washed with PBS-T (PBS containing 0.1% Tween20) 3 times for 5 min, and blocked in PBS-T containing 10% fetal bovine serum (FBS), 1% bovine serum albumin (BSA) and 3% glycine with rocking at RT for 60 min. After washing with PBS-T, the tissue blocks were stained with anti-GFP rabbit polyclonal antibody (Thermo Fisher) at a dilution of 1:500 and anti-pimonidazole mouse monoclonal antibody at a dilution of 1:50 in the antibody dilution buffer (PBS-T containing 2% FBS, 1% BSA and 3% glycine) with gentle rocking at 4 °C for 3 days to detect *kdrl:EGFP*-positive blood vessels and pimonidazole-bound hypoxic tissues, respectively. After washing with PBS-T 7 times during a 1 h period, the tissue blocks were reacted with Alexa Fluor 488-labeled goat anti-rabbit and Alexa Fluor 546-labeled goat anti-mouse antibodies (Thermo Fisher) at a dilution of 1:500 in the antibody dilution buffer with gentle rocking at 4 °C for 3 days. After washing with PBS-T 7 times during a 1 h period and with PBS 3 times during a 10 min period, the tissue blocks were mounted in 3% agarose dissolved in PBS on a 60 mm plastic dish, washed with PBS, and finally submerged with PBS. The stained skin tissues were then imaged with the FV1200 microscope as described above.

**Synthesis of PEGylated microspheres**. Polyethylene glycol-coated fluorescent microspheres (PEGylated FM) were synthesized to prevent non-specific binding to the lumen of blood vessels as follows. Carboxylate-modified microspheres [particle size: 0.48 μm, red fluorescent (580/605), Thermo Fisher] (250 μL, 20 mg/mL, 0.1575 meq/g of carboxylic groups) were dispersed in 1 mL of 100 mM MES buffer (pH 6) and centrifuged at $10,000 \times g$ for 10 min. The supernatant was removed, and the microspheres were resuspended in 900 μL MES buffer. This solution was mixed with 100 μL of MES buffer containing 43 mg/mL *N*-hydroxysulfosuccinimide (Sulfo-NHS, Thermo Fisher) and 15 mg/mL EDC HCl. After stirring for 30 min at RT, 40 mg of methoxy polyethylene glycol amine (PEG10K-NH₂, Sunbright ME-100EA, Mw 10,000, NOF Corporation) and 111 μL of 1 M phosphate solution (pH 9.4) were added. After stirring for 12 h at RT, the solution was centrifuged at $15,000 \times g$ for 10 min and the supernatant was removed. The microspheres were resuspended in 900 μL MES buffer and mixed with 100 μL of MES buffer containing 43 mg/mL Sulfo-NHS and 15 mg/mL EDC HCl. After stirring for 30 min at RT, 2.2 mg of methoxy polyethylene glycol amine (PEG550-NH2, Mw 550, Chem-Impox) and 111 μL of 1 M phosphate solution (pH 9.4) were added. After stirring for 12 h at RT, the solution was centrifuged at $15,000 \times g$ for 10 min and the supernatant was removed. The microspheres were washed three times by resuspension in 1 mL Dulbecco's phosphate-buffered saline (D-PBS), centrifuging the solution at $15,000 \times g$ for 10 min, and discarding the supernatant. After washing with D-PBS, the microspheres were resuspended in 1 mL D-PBS and sonicated for 5 min using a Taitec VP-050 ultrasonic homogenizer (1 s intervals, PWM 50%). The diameter of the microspheres was measured by dynamic light scattering (DLS) using a Malvern Zetasizer Nano (Supplementary Fig. 3 and Supplementary Table 1). Zeta potential of the microspheres was measured using a Brookhaven Nano-Brook Omni zeta potential analyzer (Supplementary Table 1).

**Analysis of hemodynamics in injured blood vessels**. To analyze vascular hemodynamics, PEGylated FM was injected into the common cardinal vein in *Tg(fli1a:Myr-EGFP)* larvae at 2.5–3.5 dpf using an IM 300 Microinjector (Narishige) (Fig. 2b, Supplementary Fig. 4d, and Supplementary Movies 9 and 10). To analyze hemodynamics in injured ISVs, PEGylated FM and/or fluorescent quantum dots (Qdot 705 or Qdot 655, Thermo Fisher) were injected into the common cardinal vein in 2.5–3.5 dpf *Tg(fli1a:Myr-EGFP)* larvae approximately 1~2.5 h after laser-ablation of ISVs (Fig. 2a, Supplementary Fig. 4a–c, and Supplementary Movies 7, 8). After injecting PEGylated FM and/or Qdots, confocal imaging was performed at the time indicated in the figure legends. For the experiments shown in Supplementary Fig. 4e, a single aISV in the *Tg(fli1a:Myr-EGFP)* larva was injured by laser ablation. A few hours after injury, the larva was treated with E3 medium containing high-concentration tricaine (0.12–0.13%) to stop blood flow, followed by Qdot probe injection as described above. Soon after injection, the larva was washed with E3 medium and soaked in E3 imaging medium to restore the heartbeat. Confocal imaging was started before blood circulation began.

Confocal imaging was performed using the A1R MP+ microscope as described above, except that resonant scanning coupled with a fast piezoelectric Z driver was employed for high-speed detection of fluorescence. Confocal images were acquired at a rate of approximately 75 frames per min for 1 min or every 10–20 s for 3–10 min. The timelapse imaging was repeated several times in some experiments.

**Arrest and restart of blood flow in zebrafish larvae**. To analyze the effects of changes in blood flow on the morphology of injured blood vessels, pre-injured and post-injured aISVs in *Tg(fli1a:Myr-EGFP)* zebrafish larvae were imaged employing the FVMPE-RS or A1R MP+ microscope (Fig. 3f, g, and Supplementary Fig. 6). At 2.5–3 h after injury, the morphology of injured ISVs was imaged. Then, the larvae were treated with E3 medium containing 1.04% tricaine to induce cardiac arrest, which resulted in the arrest of blood circulation 15–75 min later. After imaging at 1.5–2 h following blood flow arrest, the larvae were washed with E3 medium and kept in E3 imaging medium for 1.5–2.5 h to completely restart blood flow. At 2–2.5 h after restarting blood flow, images of injured ISVs were acquired. Similar experiments were performed using BDM, an inhibitor of skeletal muscle myosin-II[62], instead of tricaine with some exceptions. To analyze morphological changes of injured ISVs by lowering blood flow, confocal images were acquired every 30 min from starting treatment with 15 mM BDM until blood flow completely stopped. The data obtained from larvae showing no blood flow arrest even when treated with tricaine or BDM for more than 75 min or no flow recovery within 2.5 h after washing with E3 medium to remove the drug were excluded from the analysis.

**Analysis of EC division in injured blood vessels**. *Tg(kdrl:EGFP);(fli1a:h2b-mCherry)* adult zebrafish, in which the nucleus of each EC was visualized, were used to analyze EC division during the repair processes of injured blood vessels (Supplementary Fig. 8). After cutaneous wounding, the injured tissues were imaged every 6 or 12 h until the completion of anastomosis of injured vessels. By analyzing serial images, we counted the number of EC divisions in upstream and downstream injured vessels from post-injury to anastomosis. The number of EC divisions in upstream and downstream injured vessels was normalized to their elongation length.

We also counted the number of EC divisions in upstream and downstream injured aISVs during their repair processes by analyzing the timelapse imaging data obtained in this study.

**Suppression of blood flow-dependent EC polarization by CRISPR/Cas9-mediated knockout of *aplnrb* in zebrafish**. To suppress blood flow-dependent establishment of front–rear polarity in zebrafish, *aplnrb* was partially knocked out employing Alt-R CRISPR/Cas9 technology. The *aplnrb* dgRNA targeting exon 1 of the *aplnrb* was purchased from IDT (Supplementary Fig. 12b). The *aplnrb* dgRNA:Cas9 RNP complex solution was prepared as described for generating *toca1* and *cip4* mutants. *Tg(fli1a:mCherry);(fli1a:EYFP-Golgi)* embryos at the one-cell stage were injected with approximately 1 nL of low-dose *aplnrb* dgRNA:Cas9 RNP complex solution (1.7 μM) and raised until 3 dpf. To estimate the knockout efficiency of *aplnrb*, each larva was collected and subjected to T7EI cleavage assay using T7 Endonuclease I reaction Mix as described above (Supplementary Fig. 12c). In addition, trunk regions of the injected larvae were imaged by the A1R MP+ microscope to investigate whether *aplnrb* partial knockout results in defective EC polarization in aISVs (Supplementary Fig. 12d). These experiments revealed that larvae injected with low-dose *aplnrb* dgRNA:Cas9 RNP complex exhibit mild defects in blood flow-induced EC polarization in aISVs.

To investigate the effect of inhibiting blood flow-induced EC polarization on the elongation of upstream and downstream injured vessels, we injured aISVs in *Tg(fli1a:mCherry);(fli1a:EYFP-Golgi)* larvae injected with low-dose *aplnrb* dgRNA:Cas9 RNP complex. Among them, the larvae in which ECs in upstream injured vessels positioned their Golgi apparatus in front or middle of the nucleus toward the vessel elongation direction immediately after injury were selected and subjected to confocal imaging of repair processes of injured aISVs (Supplementary Fig. 12e, f).

**Whole-mount in situ hybridization for zebrafish embryos**. Whole-mount in situ hybridization for zebrafish embryos was performed by following the protocol described[63]. In detail, antisense digoxigenin (DIG)-labeled RNA probes for *toca1*, *cip4*, and *fbp17* were synthesized from the linearized corresponding plasmid DNAs by using the RNA labeling kit (Sigma-Aldrich). Embryos were fixed in 4% paraformaldehyde (PFA) at 4 °C overnight and dehydrated by serial incubations with PBS-T and MeOH. After incubation in 100% MeOH at −20 °C overnight, the embryos were rehydrated by serial incubations with PBS-T and MeOH, incubated with proteinase K (QIAGEN), and re-fixed in 4% PFA at RT. After washing with PBS-T, the embryos were pre-hybridized at 65 °C for 3 h in hybridization buffer (5× SSC, 50% formamide, 5 mmol/L EDTA, 0.1% Tween 20, 50 μg/mL heparin, 1 mg/mL torula yeast RNA) and then hybridized with antisense RNA probes at 65 °C for 3 days in the hybridization buffer. After hybridization, the embryos were washed twice at 65 °C for 30 min with washing buffer I (50% formamide, 2× SSC, 0.1% Tween 20), once at 65 °C for 15 min with washing buffer II (2× SSC, 0.1% Tween 20), and four times at 65 °C for 15 min with washing buffer III (0.2× SSC, 0.1% Tween 20). After incubation with PBS-T at RT for 5 min, the embryos were

pre-incubated with blocking buffer [0.1 M maleic acid, pH 7.5, 5% sheep serum, 2% blocking reagent (Sigma-Aldrich)] at RT for 3 h, incubated with anti-DIG antibody conjugated with alkaline phosphatase (1:1,000, Roche) in blocking buffer at 4 °C overnight, and washed six times with PBS-T for 2 h. Colorimetric reaction was performed using BM purple (Sigma-Aldrich) as the substrate and was stopped by washing the embryos with PBS-T. The stained embryos were then fixed in 4% PFA at 4 °C overnight, washed three times with PBS-T, and cleared by incubations of glycerol and PBS-T (80% glycerol/20% PBS). Images of the stained embryos were acquired with a SZX16 stereo microscope (Olympus).

**Cell culture and lentivirus production and infection.** HUVECs and human lung fibroblasts (hLFs) used for on-chip angiogenesis assay were obtained from Lonza. HUVECs expressing RFP (RFP-HUVECs) were purchased from Anigio-Proteomie. HUVECs and RFP-HUVECs were cultured in EGM-2 (Lonza) and used at passages 4–5. hLFs were cultured in FGM-2 and used at passages 4 to 5. HUVECs used for in vitro wound healing assay were purchased from KURABO, cultured in HuMedia-EG2 (KURABO), and used at passages 4–7.

To produce recombinant lentiviruses encoding EYFP-Golgi, EGFP-ARPC4, Myr-EGFP, EGFP-TOCA1, EGFP-CIP4, and Azami-Green, the corresponding lentiviral vectors were co-transfected with the packaging plasmids (pCAG-HIVgp and pCMV-VSV-G) into 293T cells using Lipofectamine 3000 reagents according to the manufacturer's protocol (Thermo Fischer Scientific). Five hours after transfection, transfection media were replaced with growth media (Dulbecco's modified Eagle's medium with 10% FBS). After 48 h, the conditioned media were collected and centrifuged at $1500 \times g$ for 5 min. Then, the supernatants were filtered through a 0.45-μm filter to remove floating cells and debris, and stored at −80 °C until use. The lentiviruses encoding EGFP-ARPC4, EGFP-TOCA1, and EGFP-CIP4 were further concentrated using PEG-it™ Virus Precipitation Solution (System Biosciences) according to the manufacturer's instructions and stored at −80 °C until use. HUVECs at the third or fourth passage were infected with lentiviruses at the appropriate multiplicities of infection and used at passages 5–7.

**siRNA transduction and quantitative PCR.** For gene knockdown experiments, Stealth RNAi™ siRNAs against human *CIP4* (*CIP4* siRNA-1 and *CIP4* siRNA-2) and human *TOCA1* (*TOCA1* siRNA-1 and TOCA1 siRNA-2) and negative control siRNAs (Lo GC Duplex #2 and Med GC Duplex #2) were purchased from Thermo Fischer Scientific (Supplementary Table 3). For double knockdown of *CIP4* and *TOCA1*, HUVECs were transfected with two different sets of siRNA mixtures; siRNA set1 contains 10 nM *CIP4* siRNA-1 and 10 nM *TOCA1* siRNA-1, while siRNA set2 contains10 nM *CIP4* siRNA-2 and 10 nM *TOCA1* siRNA-2. For the transfection of siRNA targeting either *CIP4* or *TOCA1*, the concentration of total siRNAs was adjusted to 20 nM by including GC content-matched control siRNA. siRNA transfection was carried out using Lipofectamine™ RNAiMAX Transfection Reagent (Thermo Fischer Scientific) according to the manufacturer's instructions. 4 h after transfection, transfection media were replaced with fresh EGM-2 medium. After an appropriate period of culture, siRNA transduced HUVECs were used for the experiments.

For quantitative PCR (qPCR), total RNA was extracted from sub-confluent HUVECs using an RNA isolation kit, NucleoSpin RNA Plus (MACHEREY-NAGEL), and reverse transcribed to cDNA using PrimerScript RT Master Mix (Takara). qPCR was performed with the Roche Lightcycler 96 system (Roche) and THUNDERBIRD SYBR qPCR Mix (TOYOBO). Sequences of the primers used are shown in Supplementary Table 2. Each value was normalized to that of *GAPDH*.

**Western blot analysis.** To analyze CIP4 expression, total proteins were extracted from cell pellets by being suspended in lysis buffer (M-PER™ Mammalian Protein Extraction Reagent, Thermo Fisher Scientific) and centrifuged at $22,140 \times g$ for 30 min at 4 °C. Lysates were analyzed by Western blotting. After sodium dodecyl sulfate–polyacrylamide gel electrophoresis (SDS-PAGE), the gel (Bio-Rad, TGX Stain-Free™ FastCast™ Acrylamide Solutions) was activated by ultraviolet exposure for 1 min and transferred to PVDF membrane using the Trans-Blot Turbo Transfer System (Bio-Rad), and then total protein was detected by the ChemiDoc™ Imaging System (Bio-Rad). Subsequently, the membrane was immersed in 5% skim milk in TBS-T and reacted with rabbit anti-human CIP4 (1:1000, Proteintech) overnight at 4 °C. Goat anti-rabbit IgG IRDye 800CW (1:10,000, LI-COR Bioscience Systems) was used as the secondary antibody and fluorescence signals were obtained by the Image analyzer ODYSSEY Fc Imaging System (LI-COR Bioscience Systems). The membrane was immersed in a stripping solution (Nacalai Tesque, 05364-55) for 30 min at 37 °C and re-blotted by rabbit anti-GAPDH (1:2000, Cell Signaling 2118S) overnight at 4 °C. Goat anti-rabbit IgG, horseradish peroxidase (HRP)-linked antibody (1:5,000, Cell signaling 7074S) was used as the secondary antibody and the chemiluminescence signal was obtained by ChemiDoc™ Imaging System (Bio-Rad).

Similar western blot analysis was carried out to detect the expression of Myr-EGFP, EGFP-TOCA1, and EGFP-CIP4. The HUVECs infected without or with the lentiviruses were lysed at 4 °C in RIPA buffer (FUJIFILM) containing a protease inhibitor cocktail (cOmplete Mini, Roche) and centrifuged at $11,000 \times g$ for 15 min at 4 °C. The supernatants were subjected to SDS-PAGE and transferred to PVDF membranes (Millipore). The membranes were blocked by 2% skim milk in TBS-T,

incubated with rabbit anti-GFP antibody (1:2,000, Thermo Fisher) or mouse anti-actin antibody (1:3000, BD Biosciences) at 4 °C overnight, and then reacted with donkey HRP-linked anti-rabbit IgG (1:3000, GE Healthcare) or sheep HRP-linked anti-mouse IgG (1:3000, GE Healthcare), respectively, at RT for 2 h. Immunocomplexes reacted with Immobilon Forte Western HRP substrate (Millipore) were visualized under a chemiluminescence detection system, LAS4000 (Cytiva).

**On-chip angiogenesis assay.** On-chip angiogenesis was induced in 3D fibrin-collagen matrices using a microfluidic device with a design modified from a previously reported version[64], allowing optimization for the following assays of hydrostatic pressure loads. The microfluidic device has five parallel channels with smaller wells (1.5 mm diameter) in channels 2 and 4 for hydrostatic pressure loads while the wells (4 mm diameter) in channels 1 and 5 are larger, thereby functioning as reservoirs for the culture media (Supplementary Fig. 5a).

The microfluidic device was fabricated out of polydimethylsiloxane (PDMS) through conventional soft lithography and replica molding, basically as previously reported[65]. In detail, a 100 μm thick SU-8 3050 (MicronChem, Westborough, MA) resist was photolithographically patterned on a silicon wafer and used as a master mold after treatment with trichloro-(1H, 1H, 2H-perfluorooctyl) silane (Sigma-Aldrich). A PDMS prepolymer [PDMS base: curing agent = 10: 1 (w/w)] (SILPOT184, Dow Corning Toray) cast on the mold was degassed in a vacuum chamber for 1 h. After separation of the PDMS slab from the mold, the inlets and outlets were punched with a 1.5 mm- or 4 mm-diameter biopsy punch (Sterile Dermal Biopsy Punch, Kai Industries). The PDMS slab and glass-bottom dish (3.5 cm diameter, Matsunami Glass, Osaka Japan) were cleared with adhesive tape and treated with oxygen plasma for 40 s for irreversible bonding to achieve covalent bonding between them, followed by curing at 80 °C overnight (Supplementary Fig. 5a). The microfluidic device was sterilized by UV irradiation before each experiment.

For angiogenesis assays, hLFs were suspended in a fibrin-collagen gel [2.5 mg/mL fibrinogen (Sigma-Aldrich), 0.15 U/mL aprotinin (Sigma-Aldrich), 0.2 mg/mL collagen type 1 (Corning), and 0.5 U/mL thrombin (Sigma-Aldrich)] at a concentration of $1 \times 10^7$ cells/mL on ice. Immediately after being mixed with thrombin (0.5 U/mL), hLFs were introduced into channel 1 (Supplementary Fig. 5a). Subsequently, the fibrin-collagen gel without any cells was injected into channels 3 and 5, and the microfluidic device was then left at RT for 15 min to achieve polymerization of the gel. Next, channels 2 and 4 and the reservoirs were filled with angiogenesis medium [EGM-2 including 20 ng/mL of human recombinant VEGF (R&D)], and the device was incubated at 37 °C overnight to allow bubbles at the gel-medium interface to disappear. Next, HUVECs suspended in an angiogenesis medium at a concentration of $5.0 \times 10^6$ cells/mL were introduced into channel 4. The device was tilted 90° for 5 min to allow HUVECs to adhere to the gel surface in channel 4. To induce angiogenesis, the device was incubated at 37 °C in a 5% $CO_2$ incubator for 1 or 3 days with no replacements of culture medium, during which ECs migrated into the gel of channel 3 to form vessel sprouts, which were subsequently used for the following experiments (Supplementary Fig. 5b, c). For all colocalization analyses, the device samples obtained 3 days after the induction of angiogenesis were subjected to whole-mount fluorescence staining.

To confirm the presence of a perfusable lumen structure of induced angiogenic sprouts, we introduced FITC-dextran (20 kDa, Sigma-Aldrich) angiogenesis medium at a concentration of 50 μg/mL into channel 4 using hydrostatic pressure differences between the outlet and the inlet. Then, the diffusion process of FITC-dextran into the lumens of angiogenic sprouts was observed using an Olympus IX83 fluorescent inverted microscope equipped with a ×20 (UPLSAPO, 0.75 NA) objective lens (Olympus) at appropriate time points (Supplementary Fig. 5g).

**Loading system of hydrostatic pressure.** To mimic the vascular IP load in the on-chip angiogenesis assay, hydrostatic pressure was loaded onto the vascular wall from the lumen side in two different ways, fixed and variable types. For the fixed type, capillary class tubes (Hirshmann) filled with angiogenesis medium (approximately 2.5 cm height from the bottom of channel 4) were placed at the outlet and the inlet of channel 4, and other outlets and inlets and culture reservoirs were plugged using silicone (REPLISIL 22 N, dent-e-con) to keep the channels a closed circuit (Supplementary Fig. 5a, h). In this case, approximately 0.6 mmHg of negative pressure was presumed to be experimentally generated in channel 4 by the capillary phenomenon (Supplementary Fig. 5h, i, k). Therefore, approximately 1.2 mmHg of vascular IP was considered to have been loaded on the vascular walls of angiogenic branches. For the variable type, pressure-resistant tubes were inserted into the inlets of channel 2 (syringe 1) and channel 4 (syringe 2), and the tubes were connected to syringes filled with angiogenesis medium at the opposite site. Other outlets and inlets and culture reservoirs were similarly plugged using silicone. The syringes were fixed to universal laboratory stands. The water surfaces in syringes 1 and 2 were first set at the same position as the bottom of channel 2, and by moving the water surface up in syringe 2, vascular IP was dynamically loaded on the vascular walls of angiogenic sprouts (Supplementary Fig. 5j). For validation of the usefulness of the variable type, actual hydrostatic pressure in pressure-resistant tubes was measured at just the distal portion of the inlet of channel 4 using a differential pressure gauge (Supplementary Fig. 5l). Furthermore, in some experiments, hydrostatic pressure in the extravascular extracellular matrix space was

similarly increased by loading hydrostatic pressure on channel 2 (1.2 mmHg, Extraluminal, Supplementary Fig. 7a). Also, hydrostatic pressure was loaded in both the intraluminal and the extraluminal spaces (Both: 1.2 mmHg, Supplementary Fig. 7a). For both loads, extraluminal pressure was loaded first, followed by IP. Either the fixed type or the variable type was selected according to the experimental objective.

**Analyses of impacts of hydrostatic pressure loads on on-chip angiogenesis**. The angiogenesis medium was replaced with fresh medium 3 days after the induction of angiogenesis. Various types of hydrostatic pressure were loaded on angiogenic branches for the indicated times, using the fixed system. Then, differential interference contrast (DIC) images were obtained for analysis of branch elongation using an IX83 inverted microscope (Olympus), and, in some experiments, the device samples were fixed and subjected to whole-mount immunofluorescence analysis. For analyses of acute changes in vascular morphology in response to the pressure loads, and experimentally by the release of pressure, continuous DIC images were obtained before and after the loads or release of pressure with the variable type using an IX83 inverted microscope. After the release of pressure, time-lapse imaging was performed for 55 h. For morphological analysis of ECs, angiogenesis was induced using HUVECs mixed with 50% RFP-HUVECs (Mosaic HUVECs). ECs at the tips of angiogenic branches (Tip ECs) and those just below the tips (stalk ECs), which express RFP, were selected, and their 3D images were obtained before and just after the pressure loads with the variable type, using a TCS SP8 laser confocal microscope (Leica). Nuclei were detected with Hoechst 33342 (Dojin). In the analysis of positional changes of the Golgi apparatus, time-lapse imaging was performed for ECs expressing EYFP-Golgi in the angiogenic sprouts composed of mosaic HUVECs with those expressing EYFP-Golgi before and after vascular IP loading with the variable type, using an IX83 inverted microscope. Nuclei were detected with Hoechst 33342. In the analysis of changes in EGFP-ARPC4 localization, vascular IP was loaded on angiogenic sprouts composed of mosaic HUVECs with those expressing EGFP-ARPC4 using the fixed type.

**Whole-mount immunofluorescence staining**. Angiogenic sprouts in the gel of the microfluidic device were fixed with 4% PFA at 4 °C for 1 h and were washed with PBS. The primary antibodies in permeabilization/blocking buffer [PBS including 0.1% TritonX-100 and 1% BSA (Sigma-Aldrich)] were added to all channels, replaced with a new medium 30 min later, and then incubated at 4 °C overnight. If necessary, phalloidin-conjugated with TRITC (50 μg/mL, Sigma-Aldrich) was added to the primary antibody solution to detect F-actin. The samples were washed with permeabilization/blocking buffer overnight, followed by replacement with new buffer, followed by incubation with appropriate secondary antibodies (1:1000) in permeabilization/blocking buffer at 4 °C overnight after replacement with new medium once. After extensive washing with PBS including 0.1% TritonX-100, nuclei were stained with DAPI (Dojin) in PBS at 4 °C for 2 h. The first and secondary antibodies used were as follows: rabbit anti-human VE-cadherin (D87F2) (1:500, Cell Signaling), mouse anti-human CD31 (WM59) (1:500, Biolegend), rabbit anti-GOLPH4 (ab28049) (1:500, Abcam), rabbit anti-human ARPC2 (07–727) (1:500, Merck Millipore), goat anti-human ARPC2 (1:500, Novus Biologicals), rabbit anti-human CIP4 (1:500, Proteintech), rabbit anti-human N-WASP (30D10) (1:100, Cell Signaling), Alexa Fluor 488-, Cy3- and Alexa Fluor 633-conjugated goat anti-rabbit IgG (Thermo Fischer Scientific), Alexa Fluor 555- and Alexa Fluor 633-conjugated goat anti-mouse IgG (Thermo Fischer Scientific) and Alexa Fluor 555 and Alexa Fluor 488 donkey anti-goat IgG (Thermo Fischer Scientific). For multiple staining, first and secondary antibodies were used in appropriate combinations. Specifically, when co-immunostained for either CIP4 or N-WASP and ARPC2, the samples were incubated with either rabbit anti-CIP4 or anti-N-WASP antibody (1:500, Cell Signaling) and goat anti-ARPC2 antibody, extensively washed, and then treated with fluorescence-conjugated donkey anti-goat IgG antibody to detect the ARPC2 signal first. After extensive rewashing, the CIP4 or N-WASP signal was detected with fluorescence-conjugated goat anti-rabbit IgG antibody.

**Imaging of on-chip angiogenesis sample**. All DIC images were obtained using a fluorescent inverted microscope (IX83, Olympus, 10x objective lens, UPLSAPO, 0.40 NA, or ×20 objective lens, UPLSAPO, 0.75 NA, Olympus), equipped with a CMOS camera (ORCA-Flash4.0, Hamamatsu Photonics). For analysis of acute changes in vascular morphology after hydrostatic pressure loads, continuous DIC images were taken using the Stream Acquisition function provided by the Meta-Morph software operating software (10 frames per second). Time-lapse images were taken every 15 min over 10 to 55 h, and for analysis of the dynamics of EYFP-Golgi (mosaic analysis), fluorescent images were simultaneously obtained together with DIC images using appropriate fluorescence filters. All images were processed employing the Fiji (http://fiji.sc.) image processing package, and the processed images were subjected to quantitative analysis, if necessary.

Fluorescent images were also obtained at 0.5- to 2.5-μm intervals along the z-axis as necessary using a confocal laser-scanning microscope (FluoView FV1200, Olympus, 20x objective lens, UPLSAPO, 0.75 NA, Olympus or oil-immersion 60x objective lens, UPLSAPO, 1.35 NA, Olympus), equipped with a GaAsP detector

(Olympus). For analysis of acute changes in cell morphology in response to the hydrostatic pressure loads, fluorescent and bright-field images were taken at 0.3 μm intervals along the z-axis using a confocal laser-scanning microscope (TCS-SP8, Leica, oil-immersion 40x objective lens, HC PL APO, 1.30 NA, Leica), equipped with a Hybrid Detector (Leica HyD, Leica), operated by LASX software version 2.0.1.14392 (Leica). All obtained images were processed employing Fiji, and the processed images were subjected to quantitative analysis, if necessary. To analyze the colocalization of F-actin, Arp2/3, and endogenous CIP4 or EGFP-TOCA1 or EGFP-CIP4, fluorescence intensity profiles along indicated lines in confocal images were measured using the Plot Profile function of Fiji.

**Analysis of impacts of CK-666 and wiskostatin on on-chip angiogenesis**. To examine the functional involvement of the Arp2/3 complexes in angiogenesis, treatment of the on-chip angiogenic sprouts was started with an angiogenesis medium including 30 μM or 300 μM of CK-666 (Calbiochem) or DMSO 1 day after the induction of angiogenesis. These treatments were continued for the following 2 days. Also, to examine the functional involvement of the Arp2/3 complexes in rescue effects of exogenous EGFP-TOCA1 expression on impaired elongation of angiogenic branches constituted by double *CIP4* and *TOCA1* knockdown, angiogenic sprouts were treated with 0 or 300 μM CK-666 for 2 days. Similarly, to analyze whether N-WASP mediates the leading-edge formation of angiogenic branches with the localized formation of Arp2/3 complexes and the resultant branch elongation, treatment of the on-chip angiogenic sprouts was started with an angiogenesis medium including 3 μM or 10 μM of wiskostatin (Sigma-Aldrich), to inhibit N-WASP, or DMSO at 1 day after induction of angiogenesis, and continued for the following 2 days. For all of the inhibitor experiments, DIC images of the total area of channel 3 were obtained for quantitative analysis of branch elongation at the indicated time after starting the treatments. In some experiments with wiskostatin, the device samples after 2-day treatment were subjected to whole-mount fluorescence staining for ARPC2 (Arp2/3 complexes) and F-actin.

**Analysis of impacts of CIP4 and/or TOCA1 knockdown on on-chip angiogenesis**. HUVECs transfected with either CIP4 or TOCA1 siRNA, both siRNAs or control siRNA were used for on-chip angiogenesis assay 16 to 20 h after the transfections. To perform rescue experiments for the effect of the depletion of CIP4 and TOCA1 on on-chip angiogenesis, HUVECs infected with either EGFP-TOCA1 or Myr-EGFP (as control) were transfected with siRNAs targeting both *CIP4* and *TOCA1* (siRNA set1). DIC images of the total area of channel 3 were taken for quantitative analysis of branch elongation 3 days after induction of angiogenesis. In some rescue experiments, the device samples obtained 3 days after angiogenesis induction were subjected to the following whole-mount immunofluorescence staining: 1) immunostaining for CD31 to quantitate percentages of EGFP-TOCA1-transduced ECs to total ECs two-dimensionally cultured in channel 4 and to tip ECs of on-chip angiogenic branches in channel 3 and 2) immunostaining for ARPC2 to quantitatively analyze the localization pattern of Arp2/3 complexes at the leading edges of angiogenic sprouts.

**Analysis of impacts of hypotonic stimulation on localization of CIP4 clusters and Arp2/3 complexes in on-chip angiogenesis**. To examine whether increased plasma membrane tension causes detachment of CIP4 and Arp2/3 complexes from the leading edge of an angiogenic sprout, the hypotonic shock was applied to the on-chip angiogenic branches. Three days after the induction of angiogenesis, the angiogenesis medium was replaced twice at a 15-min interval with a hypotonic medium, in which an equal volume of hypotonic buffer ($H_2O$ + 1 mM $CaCl_2$ + 1 mM $MgCl_2$) was added to the angiogenesis medium. The device samples were fixed 5 min after the last replacement with hypotonic medium and then subjected to whole-mount immunofluorescence staining for CIP4 and Arp2/3 complexes.

**Mechanical stretching of directionally migrating ECs during in vitro wound healing**. An autoclaved PDMS stretch chamber (#STB-CH-04-XY, STREX) was set on the biaxial cell stretching system (100-04XY, STREX), coated with 0.05 mg/mL fibronectin (FUJIFILM), and washed with sterile water. Then, $1 \times 10^6$ HUVECs infected without or with lentivirus encoding EGFP-TOCA1 were seeded on the stretch chamber (cell culture area 2 cm × 2 cm) and incubated overnight to obtain confluent cell cultures. Linear scratches with a width of 1–2 mm were made in the cell sheets on the stretch chamber with a swab. After washing twice with Opti-MEMI (Thermo Fisher), the cells were cultured in HuMedia-EG2 for 6 h. Then, HUVECs on the chamber were manually stretched to 10 % over 8 min during which 4 stretching operations (each approximately 45 s) were performed at 75 s intervals. For control experiments, the HUVECs on the chambers were subjected to the same operation except for being stretched. After the final operation, the HUVECs on the chambers were incubated at 37 °C for 3 min and fixed in 4% formaldehyde on a rocker at RT for 5 min. Immunocytochemistry of the HUVECs on the chamber set on the biaxial cell stretching system was performed according to the instructions provided by STREX. The cells were washed three times, for 5 min each time, with PBS, permeabilized with PBS containing 0.1% Triton X-100 on a rocker at RT for 5 min, washed three more times for 5 min each with PBS, and then blocked with PBS containing 2% BSA (Sigma-Aldrich) on a rocker at RT for 30 min. Subsequently, the lentivirus-uninfected cells were incubated with rabbit

anti-ARPC2 antibody (1:500, MERCK) or rabbit anti-CIP4 antibody (1:750, Proteintech) in PBS-T on a rocker at 4 °C overnight. After washing four times with PBS-T for 7.5 min each, the cells were reacted with Alexa Fluor 488 goat anti-rabbit IgG antibody (1:500, Thermo Fisher), rhodamine phalloidin (1:1000, Thermo Fisher), and DAPI (333 nM, Thermo Fisher) in PBS-T with light shielding on a rocker at RT for 2 h, washed three times with PBS-T for 7.5 min each, and washed three times with PBS for a total of 5 min. To analyze the localization of EGFP-TOCA1, the HUVECs infected with lentivirus expressing EGFP-TOCA1 were similarly permeabilized, blocked, and stained with rhodamine phalloidin and DAPI in PBS-T with light shielding on a rocker at RT for 30 min. The stained cells were imaged employing the FV3000 microscope equipped with a ×60 objective lens (N 1.10, LUMFL N). At least six fields in the wounded area were imaged for each chamber. The experiment to analyze the localization of Arp2/3 complexes, CIP4, or EGFP-TOCA1 was performed at least twice. The obtained data were processed for preparing representative images by Volocity 6.3 software and subjected to quantitative analysis using MetaMorph 7.8 software (see below).

**Analysis of localization of EGFP-ARPC4 in ECs during in vitro wound healing**. To analyze the cellular localization of Arp2/3 complexes in directionally migrating ECs, an in vitro wound-healing assay was performed using HUVECs mosaically expressing EGFP-ARPC4. HUVECs and those infected with lentivirus encoding either EGFP-ARPC4 or Azami-Green were mixed at a 4:1 ratio, plated on a 35 mm glass-base dish (Iwaki, ASAHI GLASS Company, Ltd.) coated with type I collagen at a density of $4 \times 10^4$ cells per dish, and cultured overnight to obtain a confluent monolayer of HUVECs, 20% of which expressed EGFP-ARPC4 or Azami-Green. Linear scratches were made in the cell sheets on the dish bottom with a 200 µL pipette tip. After washing twice with OptiMEMI, the cells were cultured in HuMedia-EG2 for 6 h. Then, HUVECs were fixed in 2% formaldehyde at RT for 15 min, permeabilized with PBS containing 0.05% Triton X-100 at 4 °C for 30 min, and blocked with PBS containing 4% BSA at RT for 1 h. Subsequently, the cells were incubated with mouse anti-VE-cadherin antibody (1:300, BD Biosciences) at 4 °C overnight. After three washings with PBS, the cells were stained with Alexa Fluor 633 goat anti-mouse IgG (1:500, Thermo Fisher) and rhodamine phalloidin (1:500) in PBS with light shielding at RT for 2 h. The stained cells were washed with PBS. Then, confocal fluorescent images of EGFP or Azami-Green, rhodamine, and Alexa 633 were acquired by the FV3000 microscope equipped with a ×60 objective lens (N 1.10, LUMFL N).

**Quantification of the lengths of elongated injured blood vessels**. To quantify the amounts of elongation of injured blood vessels downstream and upstream from the blood flow, the distances from the tip positions of the post-injured downstream and upstream vessels relative to the anastomotic sites after the repair were determined by analyzing the imaging data using Volocity software or Fiji software (Figs. 1b–e and 2c, d and Supplementary Figs. 1c, d and 12e, f). Injured blood vessels located downstream and upstream from the blood flow were identified by checking the direction of blood flow in the pre-injured vessels. The amounts of elongation of downstream and upstream injured vessels were expressed as a percentage relative to the total lengths of the elongated vessels. In some experiments, the amounts of elongation of injured blood vessels during the time periods indicated in figure legends were measured (Figs. 5h, i and 9c, d).

**Quantification of outer diameters of injured blood vessels**. To morphologically analyze the injured blood vessels downstream and upstream from the blood flow, the outer diameters of the downstream and upstream injured vessels at positions 5, 10, 20, 30, 40, and 50 µm from the leading edge were determined by analyzing the imaging data using Volocity or Fiji software (Fig. 3b–e). The thin projections (diameter less than 2 µm) extending from the injured vessels were excluded from these measurements. To examine the effects of changes in blood flow by tricaine and BDM on the morphology of the injured aISVs, the outer diameters of the downstream and upstream injured vessels at the position 10 µm from the leading edge were measured 2.5–3 h after injury, 1.5–2.6 h after blood flow arrest, and 1.5–2.5 h after subsequent restart of blood flow (Fig. 3f, g and Supplementary Fig. 6). In the case of the experiments with BDM, the outer diameters were also measured when blood flow was decreased but not completely arrested (0.3–0.9 h after starting the BDM treatment). Diameters of the injured vessels were expressed as a percentage relative to that of the pre-injured vessels.

**Quantification of hypoxic states in the wounded skin of adult zebrafish**. To quantify the hypoxic states in the areas surrounding the injured upstream and downstream vessels, fluorescence intensities of pimonidazole (Alexa Fluor 546) within a circle with a diameter of 30 µm drawn at the tip of each injured vessel were measured based on the confocal stack fluorescence images using Volocity software (Supplementary Fig. 2b, c). The fluorescence intensity of pimonidazole in the uninjured regions was also determined. Hypoxic states in the areas surrounding the injured blood vessels were expressed as fold increases in the mean fluorescence intensity per pixel relative to that in the uninjured skin tissues.

**Quantification of front–rear polarity of ECs in aISVs**. To quantify EC front–rear polarity in injured aISVs downstream and upstream from the blood flow, confocal stack fluorescence images in *Tg(fli1a:EYFP-golgi);(fli1a:mCherry)* larvae, in which

the Golgi apparatus and nucleus/cytosol in ECs were visualized by EYFP and mCherry fluorescence, respectively, were acquired as described above (Fig. 5a–c). The vessel elongation direction and the nucleus-Golgi vector (originating from the approximate center of mass of the nucleus and directed toward the center of mass of the Golgi apparatus) in ECs of injured ISVs were determined using Volocity software. The front–rear polarity of ECs was defined as the angle between the vessel elongation direction and nucleus-Golgi vector, as shown in Fig. 5b.

Blood flow-dependent front–rear EC polarity in aISVs was also analyzed by following the protocol described[31]. In detail, polarization patterns of ECs were classified into three groups (Upstream, Middle and Downstream groups), when the Golgi apparatus was located at the front, middle, and behind of the nucleus against the direction of blood flow, respectively (Supplementary Fig. 12d). ECs exhibiting the Upstream polarization pattern were regarded as those acquiring blood flow-dependent front–rear polarity.

**Quantification of actin filaments in the injured ISVs**. Confocal stack fluorescence images of Lifeact-mCherry and Myr-EGFP in the *Tg(fli1a:lifeact-mCherry);(fli1a:Myr-EGFP)* larvae were used to analyze actin polymerization at the leading edges of injured aISVs (Fig. 5d, e). The Myr-EGFP fluorescence-marked vascular regions within 20 µm from the leading edge (excluding <2 µm thin projections) were manually cropped using Volocity software. Then, the areas of the cropped regions and the total Lifeact-mCherry fluorescence intensity within the areas were calculated. The amounts of actin filaments in the leading edge of injured ISVs were determined by dividing the total mCherry fluorescence intensity by the vascular area.

**Quantification of Arp2/3 complexes and EGFP-Toca1 clusters in the injured ISVs**. Confocal stack fluorescence images of EGFP-ARPC4 and Lifeact-mCherry in the *Tg(fli1a:EGFP-ARPC4);(fli1a:lifeact-mCherry)* larvae or those of EGFP-Toca1 and Lifeact-mCherry in the *Tg(fli1a:EGFP-toca1);(flia1:lifeact-mCherry)* larvae were used to quantify the numbers of Arp2/3 complexes and EGFP-Toca1 clusters in the leading edges of injured aISV downstream and upstream from the blood flow. Numbers of EGFP-ARPC4-labeled Arp2/3 complexes or EGFP-Toca1 clusters that remained colocalized with Lifeact-mCherry-marked actin filaments in the vascular region within 20 µm from the leading edge (excluding <2 µm thin projections) at two consecutive time points (15-min interval) in the approximately 25% reparative phase were quantified using Volocity software. Circular regions within a 2-µm diameter whose average pixel intensity of EGFP-ARPC4 or EGFP-Toca1 fluorescence was more than twice the strength of that in the surrounding area were regarded as being an Arp2/3 complex or a Toca1 cluster, respectively.

**Quantification of vessel elongation and actin-based protrusion formation during ISV angiogenesis in the presence or absence of CK-666**. To analyze the effects of CK-666 on actin-based membrane protrusion formation during ISV angiogenesis, the *Tg(fli1a:lifeact-mCherry);(fli1a:Myr-EGFP)* embryos at 24 hpf were treated with vehicle or CK-666, and time-lapse imaged as described above. The amounts of elongation of ISVs for 3 h from the beginning of CK-666 treatment were measured using Volocity software (Supplementary Fig. 13b, c). Vascular regions within 20 µm from the leading edge (excluding <2 µm thin projections) at 7 sequential time-points from 1 h after the beginning of image recording (elapsed time; 60, 70, 80, 90, 100, 110, 120 min) were manually cropped based on the Myr-EGFP fluorescence image using Volocity software. By comparing two consecutive images, the membrane region protruding from the cropped vascular area for 10 min was defined for each time period. Then, the Lifeact-mCherry-positive area within the region of protrusion was measured using Volocity software and defined as the actin-based membrane protrusion area (Supplementary Fig. 13d).

**Quantification of ISV length in toca1[nf4] and cip4[nf5] mutants**. Confocal fluorescence images of ISVs in the *Tg(fli1a:lifeact-mCherry)* embryos at approximately 28 hpf were used to analyze ISV development in wild-type, *toca1[nf4]*, and *cip4[nf5]* embryos. The vertical length of five ISVs at the posterior end of the yolk sac was measured using Fiji and the average ISV length was calculated (Fig. 6h, i and Supplementary Fig. 17).

**Quantification of branch elongation in on-chip angiogenesis**. For evaluation of the effects of all interventions on branch elongation, branch lengths of the angiogenic sprouts in the DIC images obtained were estimated by measuring the shortest distance between the tip of a branch and the gel surface (between channel 3 and channel 4, Supplementary Fig. 5a) using Fiji. For this quantification, 15 branches were randomly selected from among all branches in the total area of channel 3. All data were averaged for each sampling point of each group, and the normalized branch length was finally calculated as a value relative to that on day 0 or a value relative to that of the control in each group.

**Quantification for front–rear polarity of ECs in on-chip angiogenesis**. The front–rear polarity status of ECs during on-chip angiogenesis was evaluated by quantifying the position of the Golgi apparatus against the nuclear position using

$z$-projection images. For this quantification, a vector from the center of the nucleus to the center of mass of the Golgi-apparatus (nucleus-Golgi vector) and a vector from the root of the branch to the tip of the branch (branch elongation vector) were set in each EC and in each branch, respectively, using Fiji. Then, the angle between the nucleus-Golgi vector and the branch elongation vector ($0 \leq \theta \leq 1\pi$ rad) was calculated in each EC, where if the angle is 0 or $1\pi$, the front–rear polarity of the EC is well established and it moves toward or against the direction of branch elongation, and if the angle is around $0.5\pi$, the polarity is lost (Fig. 4b). All ECs of all branches observed in channel 3 were evaluated.

**Quantification for vascular tip morphology and localization patterns of F-actin and clusters of Arp2/3 complexes, CIP4, and ectopically expressed EGFP-TOCA1 and EGFP-CIP4 in on-chip angiogenesis.** Vascular morphology and localization patterns of F-actin and clusters of Arp2/3 complexes, endogenous CIP4, and ectopically expressed EGFP-TOCA1 and EGFP-CIP4 between the tip and the nearest edge of the nucleus were quantitatively evaluated. Confocal $z$-projection images for F-actin or CD31were binarized after being automatically thresholded by using the MaxEntropy or Huang mode in Fiji, respectively (Area mask). Blanks in the vascular area were filled using the Fill Holes function and, if necessary, manually. Then, the vascular area was divided into 10 equal sections (Fig. 4d), and the area of each region was estimated as the numbers of pixels present therein. For analysis of F-actin, confocal $z$-projection images for F-actin were converted into binarized images based on the automatic threshold mode with the Intermode function (F-actin mask). To mainly evaluate F-actin bundles in the cytoplasmic area, a mask created by removal of the margin of the Area mask that comes from the F-actin confocal $z$-projection image was further established by repeating the Erode function 4 times in Fiji (Cytoplasmic area mask). Then, the cytoplasmic F-actin area was estimated by counting the number of F-actin-positive pixels in the cytoplasmic area of each vascular region, and % F-actin occupancy was calculated by dividing the F-actin area by the cytoplasmic area. For analysis of Arp2/3 complexes, confocal $z$-projection images were binarized after automatic thresholding with the Intermode function in Fiji. In the case of endogenous CIP4, confocal $z$-projection images were similarly binarized after Top-hat filtering with MorphoLibJ, a plug-in software in Fiji. Clusters of Arp2/3 complexes or CIP4 exceeding 5 pixels in the cytoplasmic area were counted in each region using the Analyze Particles function in Fiji, with data normalization by each cytoplasmic area. For localization analysis of EGFP-TOCA1 and EGFP-CIP4, the vascular area was defined as the area occupied by the ECs comprising the EGFP-positive tip, and thereby the area mask was similarly devised from the confocal $z$-projection image for EGFP-TOCA1 or EGFP-CIP4.

For colocalization analysis of Arp2/3 complexes with endogenous CIP4, EGFP-TOCA1, or EGFP-CIP4, individual confocal images along the z-axis were converted into binarized images based on the automatic threshold mode with the Intermode function after appropriate brightness adjustment. Both Arp2/3-positive and CIP4 (or EGFP-TOCA1 or EGFP-CIP4)-positive pixels in the vascular region were then calculated for each confocal image, and the double-positive pixels in all $x$-$y$ planes along the $z$-axis were accumulated (colocalization volume). The colocalization index was defined as a normalized value of the colocalization volume divided by the vascular area which was estimated from the area mask. Finally, the colocalization index for each vascular region was calculated as described above.

All calculations, including counting pixel numbers, except for those specifically mentioned, were performed using the Mathematica software (Wolfram).

**Quantification for acute changes in EC morphology after hydrostatic pressure loads in on-chip angiogenesis.** EC morphology in on-chip angiogenesis was quantified by 3 indices, long-axis length, short-axis length, and cell thickness (Fig. 3k–n and Supplementary Fig. 7f, g). For calculation of the long-axis length, a confocal $z$-stack image for the nucleus (Hoechst 33342, Dojin) and the cell (RFP) was filtered with Gaussian Blur 3D and binarized based on Default thresholding in Fiji. Subsequently, the binarized z-stack image for the nucleus was subjected to determination of the 3D centroid ($x_c$, $y_c$, $z_c$) using the 3D Objective Counter function in Fiji. The nucleus in the $x$-$y$ plane image ($z = z_c$) was approximated into an ellipsoid, with the long-axis being determined using the Analyze Particles (Fit Ellipse) function in Fiji. Herein, the long-axis of EC was defined as that of the ellipsoid of the nucleus. The long-axis length of EC was then manually measured in the binarized z-stack image for the cell, using Volume Viewer, a plug-in software of Fiji (Fig. 3m and Supplementary Fig. 7f). For the short-axis length and cell thickness, cross-sectional images perpendicular to the long axis at the nucleus centroid were obtained from binarized z-stack images for the cell, using Volume viewer, and then the short-axis length and cell thickness were, respectively, obtained by estimating the lengths of the long axis and the short axis of the cell in the cross-sectional image after ellipse approximation using the Analyze Particles function in Fiji (Fit Ellipse) (Fig. 3n and Supplementary Fig. 7f).

**Quantification for localization of EGFP-ARPC4 in stalk ECs of on-chip angiogenesis.** In angiogenic sprouts composed of mosaic HUVECs, numbers of stalk HUVECs, in which EGFP-ARPC4 aggregates had accumulated at the leading edge, were counted, and their percentage to total stalk HUVECs expressing EGFP-ARPC4 was calculated.

**Quantification of the effect of mechanical stretch on the localization of Arp2/3 complexes, CIP4, EGFP-TOCA1, and F-actin in directionally migrating ECs during in vitro wound healing.** MetaMorph 7.8 Software was used to quantify the colocalization of Arp2/3 complexes, CIP4, and EGFP-TOCA1 with F-actin at the leading edge of directionally migrating cells at the scratch edges (Fig. 8a–d and Supplementary Fig. 19a–d). Three consecutive z-slices that captured the cells were extracted from a series of z-stack confocal images and used to make the projection images. After subtracting the background, the projection images for F-actin were binarized after being thresholded by using the Threshold Image function to define the regions of the migrating cell sheets. The areas localized within 5 μm and those localized between 5 and 10 μm from the edge of the cell sheets were defined as the leading edge and cytoplasmic regions of the cells, respectively. The average fluorescence intensity of Arp2/3 complexes, CIP4, EGFP-TOCA1, and F-actin in the whole cytoplasmic regions was measured, but the cytoplasmic regions that overlapped with the nuclear areas were excluded from the analysis. In addition, the average fluorescence intensity for all of the pixels along the leading edges that face the direction perpendicular to the scratched wound orientation was also calculated. The pixels along the leading edges whose average fluorescence intensity of Arp2/3 complexes, CIP4, EGFP-TOCA1, and F-actin was higher than that in the whole cytoplasmic regions were regarded as positive areas. Localization of Arp2/3 complexes, CIP4, EGFP-TOCA1, and F-actin at the leading edge was expressed as a percentage of positive areas relative to total areas. The pixels exhibiting positive for both Arp2/3 complexes, CIP4, or EGFP-TOCA1 and F-actin were defined as colocalized areas. Colocalization of either Arp2/3 complexes, CIP4, or EGFP-TOCA1 with F-actin at the leading edges was expressed as a percentage of colocalized areas relative to total areas. For the analysis of EGFP-TOCA1, cells exhibiting weak or saturated fluorescence signal were excluded from the quantification.

**Statistical analysis for in vivo experiments and for in vitro wound healing assay.** All statistical analyses were performed using GraphPad Prism 8 software (GraphPad Software Inc.) or KyPlot 6.0 (KyensLab) software. Values were expressed as the mean ± s.e.m. Two groups were compared using the unpaired $t$ test, Welch's $t$ test or Mann–Whitney $U$ test (two-sided) depending on the homogeneity of variance and the normality of the distribution. Groups in one polygonal line graph were analyzed by one-way ANOVA followed by Tukey's multiple comparison test or the Kruskal-Wallis test followed by Dunn's multiple comparison test depending on the homogeneity of variance. Two polygonal line graphs were compared by two-way ANOVA followed by Sidak's multiple comparisons test. Data were considered statistically significant if the $p$ value was < 0.05. No significant difference, $p < 0.05$ and $p < 0.01$ are shown as n.s., *, and **, respectively.

**Statistical analysis for experiments using on-chip angiogenesis.** Statistical analyses were performed using GraphPad Prism 8 software (GraphPad Software Inc.) or R. Branch lengths in on-chip angiogenesis were compared between or among groups by two-way ANOVA followed by either Sidak's or Tukey's multiple comparison test. The occupancy of actin filaments, numbers of clusters of Arp2/3 complexes, endogenous CIP4, EGFP-TOCA1, and EGFP-CIP4, and blood vessel area were compared between or among groups using the Mann–Whitney $U$ test (two-sided) or the Steel–Dwass test, as appropriate. Differences in the percentage of EGFP-TOCA1 positive cells were compared between control siRNA and TOCA1/CIP4 siRNA groups and between in-sheet and in-tip ECs using the Mann–Whitney $U$ test (two-sided). A paired $t$ test (two-sided) was used for the analysis of acute morphological changes in ECs after hydrostatic pressure loads. Uniformability in the probability distribution of the angles between the direction of vessel elongation and the nucleus-Golgi vector in ECs was evaluated by applying the Kolmogorov–Smirnov goodness-of-fit test for uniform distributions. $p < 0.05$ was considered to indicate a statistically significant difference. Data are presented as means ± s.e.m.

**Reporting summary.** Further information on research design is available in the Nature Research Reporting Summary linked to this article.

## Data availability
The authors declare that all data supporting the findings of this study are available within the main text and Supplementary Materials. Raw data to generate all graphs within the figures and Supplementary Figures and uncropped scans of all blots and gels in the Supplementary Figures are provided as a Source Data File. Source data are provided with this paper.

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

## Acknowledgements

We thank N. Lawson (University of Massachusetts Medical School) for the plasmid encoding the fli1 promoter and *Tg(fli1:GFP)^y1* fish, D. Y. Stainier (Max Planck Institute for Heart and Lung Research) for *Tg(kdrl:EGFP)^s843*, K. Kawakami (National Institute of Genetics) for the Tol2 system, H. Miyoshi (BioResource Center, RIKEN) for the CSII-CMV-MCS-IRES2-Bsd lentiviral expression vector and the packaging plasmids, S. Higashiyama (Ehime University) for the lentiviral expression vector encoding Azami-Green, and N. Suzuki (Kanazawa University) for sharing the experimental protocol for immunohistochemistry of fish skin. We are grateful to the Imaging Core Laboratory, The Institute of Medical Science, The University of Tokyo, and Nikon Instruments Inc. for helping with the A1R MP+ multiphoton confocal microscope and image acquisition and to Olympus Corporation and the University of Tokyo IMCB Olympus Bioimaging Center for helping with the FluoView FVMPE-RS multiphoton microscope and image acquisition. We also thank K. Ando and H. Nakajima for advice on in vivo experiments and hemodynamic analysis, respectively, and H. Ichimiya, S. Egawa, K. Kato, K. Hiratomi, M. Sone, W. Koeda, M. Uchikawa, and C. Esumi for excellent technical assistance. This work was supported by a grant from the Japan Agency for Medical Research and Development (AMED) under Grant Number JP17gm5810010 to S.F. and by Core Research for Evolutional Science and Technology (CREST) from Japan Science and Technology (JST) under Grant Number JPMJCR14W4 to K.N., R.Y., and T.M.; by Grants-in-Aid for Scientific Research (B) to S.F. (16H05125, 21H02665) and K.N. (19H04446), for Exploratory Research to S.F. (17K19689, 19K22517, 21K19358) and K.N. (26670394), for Scientific Research for Young Scientists to S.Y. (17K15565), for Scientific Research (C) to S.Y. (19K07307) and K.N. (16KT0173) from the Japan Society for the Promotion of Science; research grants from Takeda Science Foundation to S.F., from the Naito Foundation to S.F., from Daiichi Sankyo Foundation of Life Science to S.F., from Astellas Foundation for Research on Metabolic Disorders to S.F. and K.N., from the Princess Takamatsu Cancer Research Fund to S.F., from The NOVARTIS Foundation (Japan) for the Promotion of Science to S.F. and K.N., from The Uehara Memorial Foundation to S.F., from the TERUMO LIFE SCIENCE FOUNDATION to S.F., from SENSHIN Medical Research Foundation to K.N., and from the NAKATANI FOUNDATION for advancement of measuring technologies in biomedical engineering to K.N.

## Author contributions

S.Y., K.N., and S.F. conceived and designed the research. S.Y. developed a live imaging system for adult zebrafish. S.Y. and S.F. performed in vivo experiments, with help from E.O.-N, Tomohiro Ishii, Y.W., and N.M., and analyzed the data. K.N., Y.A., Y.H., and S.H. performed the experiments using an on-chip angiogenesis model, with help from R.Y., T.M., and Kenichi Tsujita, and analyzed the data. U.H. synthesized PEGylated microspheres. E.O.-N. and S.F. generated mutant zebrafish. S.Y. and S.F. performed in vitro wound-healing assay. Kazuya Tsujita and Toshiki Itoh assisted in the design of the experiments on the role of membrane tension in the regulation of actin polymerization. S.Y., K.N., and S.F. interpreted the data and wrote the manuscript with input from all authors.

## Competing interests

The authors declare no competing interests.
