## [Peer Review File · Nature Communications]

REVIEWER COMMENTS

Reviewer #1 (Remarks to the Author):

The present manuscript addresses the timely and relevant question of how mechanical forces, in particular those that are exerted by blood and tissue pressure, influence endothelial responses to injury during wound repair. The authors use *in vivo* zebrafish studies, both in the adult and in larvae, as well as a proprietary HUVEC based on chip sprouting assay that allows intraluminal and extra cellular pressure application to monitor growth or regrowth of vessel sprouts and vessel connections after injury.

The authors first report the interesting finding that vessel downstream of previous flow patterns regrow for quicker than the proximal vessel stump, and convincingly link this phenomenon to the differences in intraluminal pressure. *In vivo* placing a second injury further upstream effectively eliminates the different growth rates, and *in vitro* pressure application significantly inhibited sprouting whilst causing lateral expansion. The authors also demonstrate significant differences in the golgi-nuclear polarization of leading tip and stalk cells during vessel regrowth, depending on up or downstream vessel segments. The then authors use loss of function and reporter expression, staining and chemical inhibition to study the role of actin nucleation by the Arp2/3 complex, and expose significant differences in the protrusive activities of the leading membrane in dependence on intraluminal pressure. In search for the sensing mechanisms, they identify significant expression of the F-BAR proteins TOCA1 and CIP4, their localization and recruitment to the membrane *in vivo* and in cell culture, but importantly their rapid disappearance upon membrane stretch or intraluminal pressure. Knockdown of both TOCA and CIP4 *in vitro* reduced Arp2/3 mediated protrusion and sprouting. Generating mutant zebrafish for both genes demonstrated a prominent role for Toca1 *in vivo*, and less so for CIP4.

Finally, the authors demonstrate that hypotonic medium causing cell swelling recapitulates the effects, suggesting that indeed endothelial membrane stretch sensed by TOCA1 and CIP4 mediate the inhibition of N-WASP/Arp2/3 interactions to drive polarized membrane protrusion and therefore vessel elongation.

The elegantly organized figures are informative and mostly clearly labelled, the main data seem to support the overall concept, and the combination of *in vitro* and *in vivo* work suitably strengthens the findings and mechanistic explanations. The flurry of video files also helps the reader to understand the dynamics, although they do not provide detailed evidence for the rapid relocalization of the proteins studied.

A number of important aspects however remain unexplored, attention to which will further improve what is already an impressive piece of work:

First, stretch induces cell proliferation in the endothelium, and in some figures, it does seem as if the upstream end of the vessel accumulates many more nuclei, suggesting potential proliferation effects. Given that proliferating cells don't migrate as much, this would provide an alternative explanation for the direct local pressure sensing driving a inhibition of membrane protrusion. Proliferation also immediately disrupts golgi-nuclear polarity patterns, and therefore would confound also the

reasoning for lack of polarity in the upstream vessel.

Secondly, the polarity of cells needs to be considered, both before and after the injury. The authors seem to ignore the fact that only the endothelial cells located in the upstream vessel end need to repolarize from their “against the flow” polarity to a sprouting polarity. The cells in the downstream end already have the “against the flow” polarity that aligns with their sprouting elongation to regrow a connection. This is another alternative explanation for why the downstream end regrows much faster.

Third, the endothelial cells exposed to membrane stretch in Figure 7 a not only lose their Arp2/3 localization from the leading edge, but also from the cell/cell junctions. Thus, although the model of stretch causing rapid removal of Arp2/3 from the membrane may be correct, this could be a much more global effect, rather than a localized effect. This would not necessarily invalidate the conclusions, but certainly deserves a comment. In particular as the proposed mechanism may also affect branching and not only elongation. This remains unexplored.

Finally, the overall concept begs the question what may be different between wound repair and developmental sprouting angiogenesis, where clearly most sprouts are lumenized and exposed to intraluminal pressure. Is it the amount of pressure that is different, or would the endothelial cells be in a different state at which the mechanism proposed here is not operating? Or is indeed the developmental timing such that sprouts are protected from intraluminal pressure due to structural differences during the sprouting process?

The authors discuss the differences in veins and arteries, but do not attempt to use their in vitro system to simulate whether vein like or artery like IP makes a real difference.

Reviewer #2 (Remarks to the Author):

The authors aimed in this manuscript to reveal the influence of hydrostatic pressure on angiogenesis using zebrafish and cell culture models. The subject is of relevance as little information is available on this stimulus in angiogenesis. Based on the obtained data it was proposed that intraluminal hydrostatic pressure induced stretching of the vessel-lining endothelium block vessel elongation. Vessel elongation in contrast was assigned to and F-BAR proteins, TOCA1 and CIP4 mediated activation of N-WASP/ARP2/3 complex to force polarized actin polymerization. Experimentations appear to me carefully done. The documentations, especially the pictures and movies are of good quality and are comprehensible in many cases. However, there are some conceptual limitations regarding different types of hemodynamics which restricts the interpretation. The following concerns are noted.

1) General:

- a. In some parts of the manuscript it appears that effects of hydrostatic pressure and the frictional force generated by blood flow are mixed up. For example: at page 8 authors state “These results indicate that blood flow-driven IP loading induces expansion of the upstream injured vessels”. Please differentiate these two stimuli through the parts of the manuscript.
- b. Please annotate, where possible, which of the vessels were investigated and name them accordingly (anatomical correct).

c. Legends are in part incomplete due to missing information. (for example: Figure 2, cell type investigated in a, how was “flow arrest” and annotation of vessel type (specify) in 2f and 2g, etc.

2) Authors state that vascular sprouting into the wound begins immediately after wounding (first result paragraph). It would be helpful to specify the time period here.

3) Authors explain that vessel sprouting is much faster "downstream" than "upstream", parameter that are not defined to the particular vessels in this particular experimental setup. In the microcirculation, flow direction and level, as well as IP, can change depending on local regulatory mechanisms and are also dependent on vessel type. Wounding, however, will most likely change all the hemodynamic parameters (flow direction and IP as well as EP) due to change in permeability, vessel contraction and edema formation. Thus, results related to hemodynamics in this model are hard to interpret. Two major problems arise:

a. To attribute the observed effects to hydrostatic pressure, I believe that measurement of the hemodynamic parameters (flow, flow direction and hydrostatic pressure) in the injured vessels is required, which is certainly a technical challenge. Also the additional attempt to set a second cut of the injured vessels (page 7) is to my taste less convincing.

b. Previous work demonstrated in the fin regeneration model in Zebrafish that artery formation relates to back migration of venous endothelial cells (Xu et al., 2014). Might it be that the different vessel growth velocities relate to different types of vessels? The authors further state that differences in up and down stream vessels (which needs to be defined) is conserved between different types of vessels. The data needs to be discussed related to work by (Xu et al 2014) and (Gebala et al., 2016) and maybe underlined by measured hemodynamic data.

4) Authors developed an in vitro system to investigate the effect of hydrostatic pressure related to angiogenesis. They showed a pressure dependent diameter increase of vessels, which is an interesting phenomenon and fits with physiological behavior. However, in vivo endothelial cells are surrounded by pericytes or smooth muscle cells which physiologically respond to increased pressure. This should be taken in consideration, when interpreting those data, and should be discussed as well.

5) Role of ARP2/3 complex in endothelial cells during vessel formation

Authors state that “.....the role of Arp2/3 complex-mediated actin polymerization in angiogenesis and the underlying regulatory mechanism have not been studied extensively“. Previous work in mouse tissue, zebrafish and cell culture models clearly demonstrated that particular small Arp2/3 complex controlled branched actin polymerization induced locally restricted membrane protrusions in angiogenesis (for example JAIL, Lateral lamellipodia, JBL), wound healing and flow which are of critical importance in all these processes (compare for example (Cao et al., 2017), (Paatero et al., 2018), (Taha et al., 2019) and many other data that relate to ARP2/3 complex mediated endothelial remodeling (for review see (Hussain and Ciulla, 2017;Belvitch et al., 2018)). Those published data should at least be discussed.

6) The data obtained about TOCA family of F-BAR proteins are well performed and interpreted.

7) Excitation wave length for EGFP is around 483 nm and not 920 nm as stated on page 33. This is probably a typo.

8) Describe in detail how flow arrest was performed.

References:

Belvitch, P., Htwe, Y.M., Brown, M.E., and Dudek, S. (2018). Cortical Actin Dynamics in Endothelial Permeability. *Curr Top Membr* 82, 141-195.

Cao, J., Ehling, M., Marz, S., Seebach, J., Tarbashevich, K., Sixta, T., Pitulescu, M.E., Werner, A.C., Flach, B., Montanez, E., Raz, E., Adams, R.H., and Schnittler, H. (2017). Polarized actin and VE-cadherin dynamics regulate junctional remodelling and cell migration during sprouting angiogenesis. *Nat Commun* 8, 2210.

Gebala, V., Collins, R., Geudens, I., Phng, L.K., and Gerhardt, H. (2016). Blood flow drives lumen formation by inverse membrane blebbing during angiogenesis in vivo. *Nat Cell Biol* 18, 443-450.

Hussain, R.M., and Ciulla, T.A. (2017). Emerging vascular endothelial growth factor antagonists to treat neovascular age-related macular degeneration. *Expert Opin Emerg Drugs* 22, 235-246.

Paatero, I., Sauteur, L., Lee, M., Lagendijk, A.K., Heutschi, D., Wiesner, C., Guzman, C., Bieli, D., Hogan, B.M., Affolter, M., and Belting, H.G. (2018). Junction-based lamellipodia drive endothelial cell rearrangements in vivo via a VE-cadherin-F-actin based oscillatory cell-cell interaction. *Nat Commun* 9, 3545.

Taha, M., Aldirawi, M., Marz, S., Seebach, J., Odenthal-Schnittler, M., Bondareva, O., Bojovic, V., Schmandra, T., Wirth, B., Mietkowska, M., Rottner, K., and Schnittler, H. (2019). EPLIN-alpha and -beta Isoforms Modulate Endothelial Cell Dynamics through a Spatiotemporally Differentiated Interaction with Actin. *Cell Rep* 29, 1010-1026 e1016.

Xu, C., Hasan, S.S., Schmidt, I., Rocha, S.F., Pitulescu, M.E., Bussmann, J., Meyen, D., Raz, E., Adams, R.H., and Siekmann, A.F. (2014). Arteries are formed by vein-derived endothelial tip cells. *Nat Commun* 5, 5758.

Reviewer #3 (Remarks to the Author):

Yuge et al. present a very interesting manuscript where they study the impact of intraluminal pressure changes in the process of wound angiogenesis. They additionally present a mechanism by which pressure affects the sub-cellular localisation of proteins involved in cell migration modulation.

Overall the data are well presented and come along in an extensive and clear experimental study. I am overall in favour for publication, as I think the study opens a new way of thinking about angiogenesis and the role of pressure variation in the process.

I only have a few points to improve the study:

- The role of pressure is very difficult to pinpoint and the approach used by the authors to modulate pressure is really basic. While it would be too time-consuming at this point to use a genetic

approach, the authors should duplicate the experiments presented in figure2 using a different drug to alter flow in order to control that the drug itself has no additional effect on angiogenesis (independent of flow). In addition, the authors should provide experimental evidence of the effect low flow (and not complete abrogation of flow which is an extreme mechanical change for the vascular network that may have non specific effects).

- the Toca mutants have angiogenic defects but this does not prove that Toca is involved in regenerative angiogenesis. the authors should assess the effect of absence of TOCA and or Cip4 in the process of regenerative wound angiogenesis to make their point and validate their model in vivo.

**** See Nature Research's author and referees' website at www.nature.com/authors for information about policies, services and author benefits.**

Replies to the reviewers' comments

Reviewer #1 (Remarks to the Author):

The present manuscript addresses the timely and relevant question of how mechanical forces, in particular those that are exerted by blood and tissue pressure, influence endothelial responses to injury during wound repair. The authors use in vivo zebrafish studies, both in the adult and in larvae, as well as a proprietary HUVEC based on chip sprouting assay that allows intraluminal and extra cellular pressure application to monitor growth or regrowth of vessel sprouts and vessel connections after injury. The authors first report the interesting finding that vessel downstream of previous flow patterns regrow for quicker than the proximal vessel stump, and convincingly link this phenomenon to the differences in intraluminal pressure. In vivo placing a second injury further upstream effectively eliminates the different growth rates, and in vitro pressure application significantly inhibited sprouting whilst causing lateral expansion. The authors also demonstrate significant differences in the golgi-nuclear polarization of leading tip and stalk cells during vessel regrowth, depending on up or downstream vessel segments. The then authors use loss of function and reporter expression, staining and chemical inhibition to study the role of actin nucleation by the Arp2/3 complex, and expose significant differences in the protrusive activities of the leading membrane in dependence on intraluminal pressure. In search for the sensing mechanisms, they identify significant expression of the F-BAR proteins TOCA1 and CIP4, their localization and recruitment to the membrane in vivo and in cell culture, but importantly their rapid disappearance upon membrane stretch or intraluminal pressure. Knockdown of both TOCA and CIP4 in vitro reduced Arp2/3 mediated protrusion and sprouting. Generating mutant zebrafish for both genes demonstrated a prominent role for Tocal in vivo, and less so for CIP4. Finally, the authors demonstrate that hypotonic medium causing cell swelling recapitulates the effects, suggesting that indeed endothelial membrane stretch sensed by TOCA1 and CIP4 mediate the inhibition of N-WASP/Arp2/3 interactions to drive polarized membrane protrusion and therefore vessel elongation.

The elegantly organized figures are informative and mostly clearly labelled, the main data seem to support the overall concept, and the combination of in vitro and in vivo work suitably strengthens the findings and mechanistic explanations. The flurry of video

files also helps the reader to understand the dynamics, although they do not provide detailed evidence for the rapid relocalization of the proteins studied.

A number of important aspects however remain unexplored, attention to which will further improve what is already an impressive piece of work:

First, we thank reviewer #1 for his/her supportive opinions on our study and insightful comments. We believe addressing the concerns raised would greatly improve our manuscript. Thus, we have performed the additional experiments and have revised the manuscript according to reviewer #1's suggestions, as described below.

First, stretch induces cell proliferation in the endothelium, and in some figures, it does seem as if the upstream end of the vessel accumulates many more nuclei, suggesting potential proliferation effects. Given that proliferating cells don't migrate as much, this would provide an alternative explanation for the direct local pressure sensing driving a inhibition of membrane protrusion. Proliferation also immediately disrupts golgi-nuclear polarity patterns, and therefore would confound also the reasoning for lack of polarity in the upstream vessel.

As the reviewer pointed out, cell stretching induces division of endothelial cells (ECs) and proliferating cells usually do not migrate efficiently, raising the possibility that intraluminal pressure (IP) load-induced cell stretching induces EC division to disrupt front-rear polarity, thereby inhibiting elongation of upstream injured vessels. To address this hypothesis, we analyzed the number of EC divisions in upstream and downstream injured blood vessels during cutaneous wound healing and found that the number of EC divisions was significantly higher in the downstream than in the upstream injured vessels (Supplementary Fig. 8). In addition, we analyzed proliferation of ECs during repair processes of injured intersegmental vessels (ISVs). However, EC division occurred in neither upstream nor downstream injured vessels (mentioned in the "Results" section). These results indicate that IP load-mediated inhibition of vessel elongation does not depend on increased EC division. These results have been included as the new Supplementary Fig. 8., and the manuscript has been revised accordingly (p.12).

Secondly, the polarity of cells needs to be considered, both before and after the injury. The authors seem to ignore the fact that only the endothelial cells located in the upstream vessel end need to repolarize from their “against the flow” polarity to a sprouting polarity. The cells in the downstream end already have the “against the flow” polarity that aligns with their sprouting elongation to regrow a connection. This is another alternative explanation for why the downstream end regrows much faster.

We fully agree with reviewer #1 that blood flow-mediated EC polarization before injury should be considered, because ECs in upstream injured vessels need to reverse their front-rear polarity to migrate forward for vessel repair.

As the reviewer pointed out, immediately after injury, the ECs in upstream injured vessels positioned their Golgi apparatus behind the nucleus toward the direction of vessel elongation (Fig. 5a, b and Supplementary Fig. 12a). Subsequently, their Golgi apparatus became randomly positioned until the early stage of regeneration (Fig. 5a, b). If the ECs in upstream injured vessels acquired front-rear polarity at this time, the Golgi apparatus should gradually turn toward the direction of vessel elongation. However, the Golgi apparatus remained randomly positioned even when the injured vessels were repaired (Fig. 5a, b). These results suggest that ECs in upstream injured vessels fail to acquire the front-rear polarity necessary for directed cell migration. Thus, inefficient elongation of upstream injured vessels was likely attributable to loss of front-rear polarity of ECs rather than time lag to reverse their front-rear polarity.

To confirm it, we further examined elongation of injured arterial ISVs (aISVs) in zebrafish larvae injected with a Cas9/guide RNA (gRNA) targeting *aplnr*b (apelin receptor b), since *Aplnr*b reportedly regulates EC polarization by blood flow (Kwon et al. Nat. Commun. 7:11805, 2016). As expected, the larvae injected with low dose of *aplnr*b gRNA exhibited mild defects in blood flow-induced EC polarization in aISVs (Supplementary Fig. 12b-d). Therefore, we injured their aISVs and analyzed the larvae in which ECs in upstream injured vessels positioned their Golgi apparatus in front or middle of the nucleus toward the vessel elongation direction immediately after injury (Supplementary Fig. 12e, f). In those larvae, the injured ISVs were normally repaired, during which vessel elongation was preferentially induced at a site downstream from blood flow, while the injured upstream vessels did not efficiently elongate, as observed in control larvae. These results reveal that *Aplnr*b is not essential for establishing the front-rear polarity required for directed EC migration during repair processes of injured ISVs and further suggest that time lag to reverse front-rear polarity for vessel repair is not a cause of inefficient elongation of upstream injured vessels. Collectively, these

findings suggest that IP loading disrupts the front-rear polarity of ECs in upstream injured vessels. These results have been included as the new Supplementary Fig. 12 and the “Results” section has been revised accordingly (p.14, line9 – p.15, line14).

Third, the endothelial cells exposed to membrane stretch in Figure 7 a not only lose their Arp2/3 localization from the leading edge, but also from the cell/cell junctions. Thus, although the model of stretch causing rapid removal of Arp2/3 from the membrane may be correct, this could be a much more global effect, rather than a localized effect. This would not necessarily invalidate the conclusions, but certainly deserves a comment. In particular as the proposed mechanism may also affect branching and not only elongation. This remains unexplored.

As the reviewer pointed out, Arp2/3 complexes localized not only at leading edges but also at cell-cell contacts and their junctional localization was also prevented by stretching of ECs (Fig. 8a). To carefully analyze cellular localization of Arp2/3 complexes in directionally migrating ECs, we performed an *in vitro* wound healing assay in which EGFP-ARPC4 was mosaically expressed in ECs and found that EGFP-ARPC4 also localized in junctional regions at the leading edges of follower cells (Supplementary Fig. 19e, f). A previous study showed that Arp2/3 complexes localized at the leading edge of follower cells and induced polarized formation of actin-driven junctional intermittent lamellipodia (JAIL) to promote directed EC migration during sprouting angiogenesis (Cao et al. Nat. Commun. 8: 2210, 2017). Consistently, we also showed, using an on-chip angiogenesis model, that EGFP-ARPC4 localized at the leading edge of stalk cells and its localization was abolished by IP loading (Supplementary Fig. 10f, g). Therefore, our data suggest that cell stretching induces removal of Arp2/3 complexes from the leading edges not only in leader ECs but also in follower ECs. These new data have been included as the new Supplementary Fig. 19e, f, and the manuscript has been revised accordingly (p.20, line10–15).

As reviewer #1 noted, cell stretch-induced removal of Arp2/3 complexes from the leading edge may affect not only vessel elongation but also vessel branching. Although this hypothesis is very intriguing and important, we believe that this issue is beyond the scope of the present investigation and should be addressed in a future study.

Finally, the overall concept begs the question what may be different between wound

repair and developmental sprouting angiogenesis, where clearly most sprouts are lumenized and exposed to intraluminal pressure. Is it the amount of pressure that is different, or would the endothelial cells be in a different state at which the mechanism proposed here is not operating? Or is indeed the developmental timing such that sprouts are protected from intraluminal pressure due to structural differences during the sprouting process?

As reviewer #1 pointed out, it is very important to ascertain why blood vessel sprouts elongate even in the presence of IP loading during developmental angiogenesis. It has been shown that the vascular lumens in angiogenic sprouts are formed by stalk cells and exposed to blood flow-driven IP during developmental angiogenesis. Regarding this question, we discussed several possibilities that may explain why vessel sprouts elongate in the presence of IP loading during developmental angiogenesis. One possibility is that IP applied to the vascular sprouts during developmental angiogenesis might be lower than that loaded onto upstream injured vessels during wound angiogenesis, because fully established blood flow is present in pre-injured vessels. Another possibility is that the states that ECs are in may differ between developmental and wound angiogenesis. Since ECs in mature blood vessels are maintained in a quiescent state, they are likely to remain quiescent immediately after injury. In contrast, activated ECs exist in elongating vessels during developmental angiogenesis. Therefore, quiescent ECs might be more sensitive to IP load-induced cell stretching than activated ECs. Alternatively, different extravascular environments might result in different effects of IP loading on vessel elongation in developmental and wound angiogenesis. Vascular wall stretching is thought to depend on several parameters including the pressure gap between IP and EP, visco-elasticities of the vascular wall, and extravascular interstitial tissue. Indeed, our *in vitro* studies revealed that IP loading did not inhibit vessel elongation in the presence of high interstitial pressure (Fig. 3h, i). In addition, IP loading is expected not to efficiently induce vessel expansion and EC stretching if stiffness of the surrounding tissues is high. Therefore, blood flow-driven IP loading and the extravascular tissue environments might regulate vessel elongation, in a coordinated fashion, during angiogenesis. These possibilities are mentioned in the Discussion section (p.24, line10 – p.25, line5) and will be addressed in greater detail in a future study.

The authors discuss the differences in veins and arteries, but do not attempt to use their

in vitro system to simulate whether vein like or artery like IP makes a real difference.

In a previous version of our manuscript, we discussed the possibility that higher IP loading might restrict sprouting from and/or elongation of arterial vessels during angiogenesis. However, as reviewer #1 pointed out, we did not examine whether vein-like or artery-like IP results in different forms of elongation of vascular sprouts using an on-chip angiogenesis model. We think that this issue is important, but should be investigated in another study, because its objective is not precisely the same as that of our current study. Therefore, we had already removed this discussion from the previous version of the manuscript.

Reviewer #2 (Remarks to the Author):

The authors aimed in this manuscript to reveal the influence of hydrostatic pressure on angiogenesis using zebrafish and cell culture models. The subject is of relevance as little information is available on this stimulus in angiogenesis. Based on the obtained data it was proposed that intraluminal hydrostatic pressure induced stretching of the vessel-lining endothelium block vessel elongation. Vessel elongation in contrast was assigned to and F-BAR proteins, TOCA1 and CIP4 mediated activation of N-WASP/ARP2/3 complex to force polarized actin polymerization. Experimentations appear to me carefully done. The documentations, especially the pictures and movies are of good quality and are comprehensible in many cases. However, there are some conceptual limitations regarding different types of hemodynamics which restricts the interpretation. The following concerns are noted.

We thank reviewer #2 for his/her insightful comments. We believe addressing the concerns raised by the reviewer would substantially improve our manuscript. Therefore, we have revised the manuscript by addressing all of the concerns raised by reviewer #2, as described in detail below.

1) General:

a. In some parts of the manuscript it appears that effects of hydrostatic pressure and the frictional force generated by blood flow are mixed up. For example: at page 8 authors state “These results indicate that blood flow-driven IP loading induces expansion of the upstream injured vessels”. Please differentiate these two stimuli through the parts of the manuscript.

We agree that the issue pointed by reviewer #2 is very important, because we did not analyze hemodynamics in injured blood vessels. Hence, we analyzed hemodynamics in injured arterial intersegmental vessels (aISVs) in zebrafish larvae (Fig. 2a, b, Supplementary Fig. 4, and Supplementary Movies 7-10). For this purpose, we injected polyethylene glycol-coated fluorescent microspheres (PEGylated FM) and quantum dots (Qdot) into blood vessels of zebrafish larvae. FM was coated with polyethylene glycol to avoid non-specific binding to the blood vessel lumen (Supplementary Fig. 3 and Supplementary Table 1). The particle size of PEGylated FM and Qdots is approximately 0.5 μm and 10 nm, respectively. Qdots are sufficiently small to diffuse freely in the vessels, and thereby visualize the blood vessel lumen. On the other hand,

diffusion of PEGylated FM within the narrow injured vessels is expected to be much slower due to the high particle/vessel size ratio compared to Qdots. Because of this, PEGylated FM can rarely enter the injured vessels without blood flow.

When aISVs were severed by laser ablation, both PEGylated FM and Qdots entered only the base of downstream injured vessels, barely reaching the tip (Fig. 2a, Supplementary Fig. 4a-c and Supplementary Movies 7, 8), indicating that blood did not flow into the lumen. Thus, markedly high IP and shear stress are not applied to downstream injured vessels. On the other hand, the entire region of upstream injured vessels was filled with Qdots (Fig. 2a, Supplementary Fig. 4b, c and Supplementary Movies 7). However, PEGylated FM rarely entered upstream injured vessels from the dorsal aorta, despite frequently going into intact aISVs before injury (Fig. 2a, b, Supplementary Fig. 4a, d and Supplementary Movies 7, 9, 10). Importantly, some PEGylated FM, which ended up in upstream injured vessels, showed Brownian motion-like movement when entering upstream injured vessels by chance (Fig. 2a, Supplementary Fig. 4a and Supplementary Movie 8), suggesting absence of laminar blood flow within upstream injured vessels. Moreover, we confirmed the hemodynamics in injured arterial ISVs by injecting Qdots into the common cardinal veins in larvae with cardiac arrest and subsequently analyzing the fluorescence dynamics in injured aISVs in response to re-starting blood flow (Supplementary Fig. 4e and Supplementary Movie 11). The dorsal aorta was quickly filled with Qdots when blood flow started. However, the Qdots moved only gradually from the dorsal aorta to the tip of the upstream injured aISV (approximately 0.3 $\mu\text{m/s}$), suggesting Qdot accumulation in upstream injured aISV via passive diffusion. Collectively, these findings indicate that blood flow in upstream injured vessels is minimal or absent. Therefore, upstream injured vessels are probably exposed mainly to IP rather than shear stress generated by blood flow.

This conclusion is also supported by the following results.

- (1) The diameter of upstream injured ISVs became smaller when blood flow was stopped by treatment with either tricaine or 2,3-butanedione monoxime (BDM) and re-expansion occurred in response to re-starting blood flow (Fig. 3f, g and Supplementary Fig. 6). In contrast, changes in blood flow did not significantly affect the morphology of downstream injured ISVs (Fig. 3f, g and Supplementary Fig. 6). Together with the results mentioned above, these data strongly suggest that blood flow-driven IP loading induces expansion of upstream injured vessels.
- (2) We investigated whether IP loading suppresses vessel elongation by mechanically stretching ECs or by applying hydrostatic pressure to ECs. For this purpose, we

examined the effects of IP or extraluminal pressure (EP) loading, alone and in combination, on the elongation of angiogenic sprouts using an on-chip angiogenesis model. IP loading immediately induced vessel expansion, while loading of EP caused shrinkage of angiogenic branches (Supplementary Fig. 7b-e and Supplementary Movies 13, 14). Furthermore, the shrunken vessels loaded with EP were only slightly expanded by additional loading of IP (Supplementary Fig. 7d, e and Supplementary Movie 15), suggesting that ECs in vessels loaded with either EP or both EP and IP were not exposed to stretching, despite hydrostatic pressure having been applied to these ECs. Thus, using our novel system, we examined whether hydrostatic pressure is involved in IP load-induced inhibition of vessel elongation, and found that loading of either EP or both EP and IP did not inhibit elongation of angiogenic branches (Fig. 3h, i). In contrast, the expanded vessels loaded with IP failed to elongate, but they showed immediate shrinkage and began to extend protrusions and re-elongate upon the release of pressure (Fig. 3h-j and Supplementary Movies 16, 17). These results suggest EC stretching to be a cause of IP load-mediated inhibition of vessel elongation.

Considering all of the data described above, we concluded that blood flow-driven IP loading, but not blood flow-generated shear stress, inhibits elongation of upstream injured vessels through vessel expansion.

b. Please annotate, where possible, which of the vessels were investigated and name them accordingly (anatomical correct).

According to reviewer #2's suggestion, information regarding which types of blood vessels were investigated have been added in panels of figures, wherever possible. In addition, upstream and downstream injured vessels shown in figures are indicated by blue and red arrowheads (Figs. 1a, 1b, 1d, 2a, 2c, 3f, 9c and Supplementary Figs. 1b, 1c, 4a, 4b, 4c, 4e, 6a) or labeled by "Down" and "Up" (Figs. 5a, 5d, 5f, 5h, 9a and Supplementary Fig. 12e), respectively. In Supplementary Fig. 1a, elongating injured blood vessels are indicated by green arrowheads.

c. Legends are in part incomplete due to missing information. (for example: Figure 2, cell type investigated in a, how was "flow arrest" and annotation of vessel type (specify) in 2f and 2g, etc.

We apologize for the information missing from the figure legends. We checked and

added the missing information to the legends of all of the figures.

2) *Authors state that vascular sprouting into the wound begins immediately after wounding (first result paragraph). It would be helpful to specify the time period here.*

In accordance with reviewer #2's suggestion, we have now specified the time period during which vascular sprouting into the wound begins in the Results section (p.7, line 6–8).

3) *Authors explain that vessel sprouting is much faster "downstream" than "upstream", parameter that are not defined to the particular vessels in this particular experimental setup. In the microcirculation, flow direction and level, as well as IP, can change depending on local regulatory mechanisms and are also dependent on vessel type. Wounding, however, will most likely change all the hemodynamic parameters (flow direction and IP as well as EP) due to change in permeability, vessel contraction and edema formation. Thus, results related to hemodynamics in this model are hard to interpret. Two major problems arise:*

a. To attribute the observed effects to hydrostatic pressure, I believe that measurement of the hemodynamic parameters (flow, flow direction and hydrostatic pressure) in the injured vessels is required, which is certainly a technical challenge. Also the additional attempt to set a second cut of the injured vessels (page 7) is to my taste less convincing.

As described in the reply to Reviewer #2's comment mentioned in 1), we analyzed hemodynamics in injured blood vessels. The data from these experiments clearly showed that blood flow in upstream injured vessels is minimal or even absent and, together with the other results, further revealed that blood flow-driven IP loading, but not blood flow-generated shear stress, restricts elongation of upstream injured vessels through vessel expansion.

b. Previous work demonstrated in the fin regeneration model in Zebrafish that artery formation relates to back migration of venous endothelial cells (Xu et al., 2014). Might it be that the different vessel growth velocities relate to different types of vessels? The authors further state that differences in up and down stream vessels (which needs to be

defined) is conserved between different types of vessels. The data needs to be discussed related to work by (Xu et al 2014) and (Gebala et al., 2016) and maybe underlined by measured hemodynamic data.

We consider the data reported by Xu et al. to not be discrepant, but rather consistent with our conclusion. Xu et al. showed that ECs mainly sprout from veins but not from arteries and contribute to artery formation to repair injured blood vessels during zebrafish fin regeneration (Xu et al. Nat. Commun. 5: 5758, 2014). Since blood is expected to be pumped mainly to the arteries comparing to the veins, these results suggest that blood flow-driven IP loading might restrict EC sprouting from the arteries during fin regeneration. Therefore, their study results also support our conclusion that blood flow-driven IP loading restricts vessel elongation during wound healing. Such discussion related to the work by Xu et al. has now been included in the Discussion section (p.24, line 10–17).

In contrast to wound angiogenesis, however, blood vessel sprouts elongate even in the presence of IP loading during developmental angiogenesis. Indeed, Gebala et al. previously reported that blood flow drives lumen expansion of elongating vessels during sprouting angiogenesis (Gebala et al. Nat. Cell Biol. 18: 443-450, 2016). At present, why blood flow-driven IP loading regulates wound angiogenesis and developmental angiogenesis differently remains unknown. However, we discussed several possibilities that might explain why vessel sprouts elongate in the presence of IP loading during developmental angiogenesis, as described in the reply to Reviewer #1's last comment. One possibility is that IP applied to the vascular sprouts during developmental angiogenesis might be lower than that loaded onto upstream injured vessels during wound angiogenesis, because fully established blood flow is present in pre-injured vessels. Another possibility is that the states that ECs are in may differ between developmental and wound angiogenesis. Since ECs in mature blood vessels are maintained in a quiescent state, they are likely to remain quiescent immediately after injury. In contrast, activated ECs exist in elongating vessels during developmental angiogenesis. Therefore, quiescent ECs might be more sensitive to IP load-induced cell stretching than activated ECs. Alternatively, different extravascular environments might result in different effects of IP loading on vessel elongation in developmental and wound angiogenesis. Vascular wall stretching is thought to depend on several parameters including the pressure gap between IP and EP, visco-elasticities of the vascular wall, and extravascular interstitial tissue. Indeed, our *in vitro* studies revealed that IP loading did not inhibit vessel elongation in the presence of high interstitial

pressure (Fig. 3h, i). In addition, IP loading is expected not to efficiently induce vessel expansion and EC stretching if stiffness of the surrounding tissues is high. Therefore, blood flow-driven IP loading and the extravascular tissue environments might regulate vessel elongation, in a coordinated fashion, during angiogenesis. These possibilities are mentioned in the Discussion section (p.24, line 17–p.25, line 5) and will be addressed in greater detail in a future study.

4) Authors developed an in vitro system to investigate the effect of hydrostatic pressure related to angiogenesis. They showed a pressure dependent diameter increase of vessels, which is an interesting phenomenon and fits with physiological behavior. However, in vivo endothelial cells are surrounded by pericytes or smooth muscle cells which physiologically respond to increased pressure. This should be taken in consideration, when interpreting those data, and should be discussed as well.

As reviewer #2 pointed out, mural cells such as pericytes and vascular smooth muscle cells cover ECs forming the inner surface of vascular tubes *in vivo* and are known to regulate the diameter of blood vessels in response to blood flow. Hence, mural cell coverage might finely control the IP load-induced suppression of vessel elongation. In this regard, in zebrafish larvae at 3 dpf, most of the aISVs were wrapped by mural cells, while many venous ISVs lacked this mural cell coverage (This observation has been included as the new Supplementary Fig. 21). Nevertheless, the difference in elongation of injured blood vessels was similar when either arterial or venous ISVs were severed by laser ablation (Fig. 1d, e and Supplementary Fig. 1c, d). These results indicate that blood flow-driven IP loading suppresses elongation of upstream injured vessels whether or not mural cell coverage is present, at least, in the ISVs. Thus, careful examination is necessary to elucidate the role of mural cells in regulating IP load-induced suppression of vessel elongation. In our view, this issue is beyond the scope of the present experiments and should be addressed in a future study. This issue is now mentioned in the Discussion section (p.25, line 6–24).

5) Role of ARP2/3 complex in endothelial cells during vessel formation
Authors state that “.....the role of Arp2/3 complex-mediated actin polymerization in angiogenesis and the underlying regulatory mechanism have not been studied extensively“. Previous work in mouse tissue, zebrafish and cell culture models clearly

demonstrated that particular small Arp2/3 complex controlled branched actin polymerization induced locally restricted membrane protrusions in angiogenesis (for example JAIL, Lateral lamellipodia, JBL), wound healing and flow which are of critical importance in all these processes (compare for example (Cao et al., 2017), (Paatero et al., 2018), (Taha et al., 2019) and many other data that relate to ARP2/3 complex mediated endothelial remodeling (for review see (Hussain and Ciulla, 2017; Belvitch et al., 2018)). Those published data should at least be discussed.

We appreciate the valuable information about the role of Arp2/3 complexes in ECs. We agree with the reviewer that Arp2/3 complex-mediated actin polymerization has been shown to regulate angiogenesis, EC migration, junctional remodeling and so on. Thus, we have revised the “Introduction” section accordingly (p.5, line 7–10).

6) The data obtained about TOCA family of F-BAR proteins are well performed and interpreted.

We thank reviewer #2 for his/her positive opinion of our data pertaining to the TOCA family of F-BAR proteins.

7) Excitation wave length for EGFP is around 483 nm and not 920 nm as stated on page 33. This is probably a typo.

Two-photon excited fluorescence of EGFP was imaged using an FVMPE-RS multiphoton upright microscope system (Olympus). Therefore, the two-photon excitation wavelength for EGFP was 920 nm.

8) Describe in detail how flow arrest was performed.

The method of arresting blood flow is now described in detail in the Methods section (p.49, line 2–18).

Reviewer #3 (Remarks to the Author):

Yuge et al. present a very interesting manuscript where they study the impact of intraluminal pressure changes in the process of wound angiogenesis. They additionally present a mechanism by which pressure affect the sub cellular localisation of proteins involved in cell migration modulation.

Overall the data are well presented and come along in an extensive and clear experimental study. I am overall in favour for publication, as I think the study opens new way of thinking angiogenesis and the role of pressure variation in the process.

I only have a few points to improve the study:

First, we thank reviewer #3 for his/her positive opinions on our study and the valuable comments provided. We believe that the new experiments described in the revised manuscript, according to the suggestions made by reviewer #3, have significantly strengthened our conclusion.

- The role of pressure is very difficult to pinpoint and the approach used by the authors to modulate pressure is really basic. While it would be too time consuming at this point to use a genetical approach, the authors should duplicate the experiments presented in figure2 using a different drug to alter flow in order to control that the drug itself has no additional effect on angiogenesis (independent of flow). In addition, the authors should provide experimental evidence of the effect low flow (and not complete abrogation of flow which is an extreme mechanical change for the vascular network that may have non specific effects).

In a previous version of our manuscript, we used only tricaine, an anesthetic agent, to control blood flow and found that the diameter of upstream injured vessels became smaller when blood flow was arrested. However, as reviewer #3 pointed out, the same experiments should be repeated using a different drug to confirm that the decrease in diameter of upstream injured vessels is caused by the arrest of blood flow rather than as a side effect. Therefore, we additionally used 2,3-butanedione monoxime (BDM) to control blood flow and obtained results similar to those with tricaine (Supplementary Fig. 6). In accordance with the reviewer's suggestion, we also examined the effect of low blood flow on the morphology of injured ISVs and found that the diameter of upstream injured vessels, but not that of downstream vessels, correlates with blood flow

velocity (Supplementary Fig. 6). These results clearly indicate that blood flow induces expansion of upstream injured vessels. These data have been included as the new Supplementary Fig. 6.

- the Toca mutants have angiogenic defects but this does not prove that Toca is involved in regenerative angiogenesis. the authors should assess the effect of absence of TOCA and or Cip4 in the process of regenerative wound angiogenesis to make their point and validate their model in vivo.

We fully agree with reviewer #3 that we should analyze regenerative wound angiogenesis in the *tocal* mutant to strengthen our conclusion. Therefore, we analyzed repair processes of injured ISVs in *tocal* mutant larvae. Elongation of the downstream injured vessels was significantly slower in the *tocal^{nf4/nf4}* and *tocal^{nf4/+}* larvae than in wild type larvae (Fig. 9c, d). However, the upstream injured vessels only marginally elongated irrespective of their *tocal* genotypes (Fig. 9c, d). These results clearly show that Toca1 not only promotes wound angiogenesis but also acts as a sensor for IP load-induced EC stretching to inhibit elongation of upstream injured vessels. These results have been included as the new Fig. 9c and 9d.

REVIEWERS' COMMENTS

Reviewer #1 (Remarks to the Author):

The authors present a substantially improved revision of their exciting work. They not only have addressed all my comments in a highly comprehensive manner, but thoroughly tested all alternative hypotheses by performing numerous additional careful experiments. Overall, this is an extremely impressive body of work, beautifully illustrated. The discussion is thorough and educating. Exciting concept, high value for the interested vascular biology community, but moreover for any cell biologists as well. In fact, the concepts might also be interesting for clinicians interested in improving wound healing, so all in all very exciting. I have no further comments.

Reviewer #2 (Remarks to the Author):

In most cases, the authors have been responsive to the criticisms and have responded appropriately. Therefore, I have no further reservations about recommending the ms for publication

Reviewer #3 (Remarks to the Author):

The authors have addressed all my comments, very nice work.

** See Nature Research's author and referees' website at www.nature.com/authors for information about policies, services and author benefits

Replies to the reviewers' comments

Reviewer #1 (Remarks to the Author):

The authors present a substantially improved revision of their exciting work. They not only have addressed all my comments in a highly comprehensive manner, but thoroughly tested all alternative hypotheses by performing numerous additional careful experiments. Overall, this is an extremely impressive body of work, beautifully illustrated. The discussion is thorough and educating. Exciting concept, high value for the interested vascular biology community, but moreover for any cell biologists as well. In fact, the concepts might also be interesting for clinicians interested in improving wound healing, so all in all very exciting. I have no further comments.

We really appreciate reviewer #1's positive opinions and encouraging words on our current study. We believe that addressing the comments from the reviewer greatly improved our manuscript.

As reviewer #1 pointed out, we think that this study will be interesting not only for vascular biologists but also for many fields of cell biologists, because this is the first report to show the significant role of regulatory mechanism of cell migration by cell stretch-induced membrane tension in tissue morphogenesis *in vivo*. In addition, our discovery of an unexpected role of intraluminal pressure in regulating angiogenesis might contribute to developing novel effective therapies for non-healing wounds and ischemic diseases, as discussed in the Discussion section. Therefore, as reviewer #1 mentioned, this study might also be interesting for clinicians.

Again, we thank reviewer #1 for enormously improving our manuscript by giving us valuable and insightful comments.

Reviewer #2 (Remarks to the Author):

In most cases, the authors have been responsive to the criticisms and have responded appropriately. Therefore, I have no further reservations about recommending the ms for publication.

We thank reviewer #2 for his/her positive opinions on our revised manuscript. We believe that our manuscript has substantially been improved by addressing the concerns raised by the reviewer.

Reviewer #3 (Remarks to the Author):

The authors have addressed all my comments, very nice work.

We really appreciate reviewer #3's positive opinions on our revised manuscript. We believe that addressing the comments raised by the reviewer greatly improved our manuscript.